# ADVANCING LLM REASONING WITH NATURAL LANGUAGE AND NUMERICAL FEEDBACK

## ABSTRACT

Recent advances in reinforcement learning (RL) with numerical feedback, such as scalar rewards, have significantly enhanced the complex reasoning capabilities of large language models (LLMs). Despite this success, we identify three key challenges encountered by RL with solely numerical feedback: performance plateaus, limited effectiveness of spontaneous self-reflection, and persistent failures. We then demonstrate that RL-finetuned models, even after exhibiting performance plateaus, can generate correct refinements on persistently failed problems by leveraging natural language feedback in the form of critiques. Building on this insight, we propose Critique-GRPO, an online RL framework that integrates both natural language and numerical feedback for effective policy optimization. Critique-GRPO enables LLMs to learn from initial responses and critique-guided self-refinements simultaneously while maintaining exploration. Additionally, we employ a shaping function to amplify learning from correct, especially unfamiliar, refinements and penalize incorrect ones. Extensive experiments show that Critique-GRPO outperforms all compared supervised and RL-based fine-tuning methods, achieving average Pass@1 improvements of approximately +15.0%, +21.6%, and +15.0% on Qwen2.5-7B-Base, Qwen2.5-Math-7B-Base, and Qwen3-8B across eight challenging reasoning tasks. Notably, Critique-GRPO facilitates effective self-improvement through self-critiquing, achieving substantial gains over GRPO, *e.g.,* a +16.7% Pass@1 improvement on AIME 2024.

## 1 INTRODUCTION

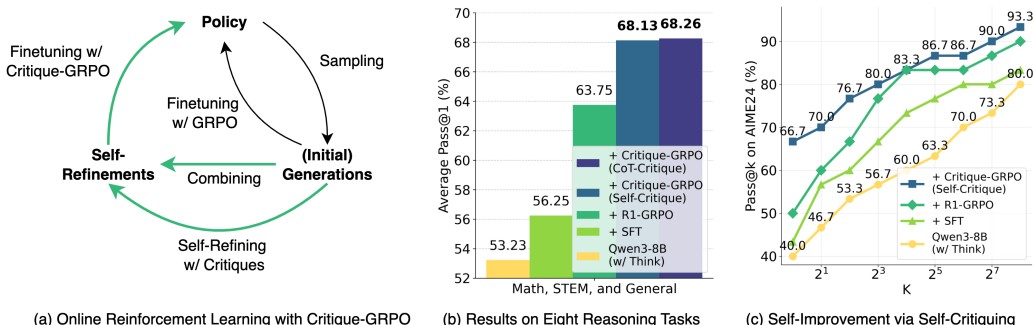

(a) Online Reinforcement Learning with Critique-GRPO    (b) Results on Eight Reasoning Tasks    (c) Self-Improvement via Self-Critiquing

Figure 1: (a) Critique-GRPO enhances online reinforcement learning by enabling the model to learn from both initial responses and self-refinements through natural language feedback (critiques), highlighted in green, instead of relying solely on numerical feedback (scalar rewards). (b) Critique-GRPO improves the average Pass@1 score on Qwen3-8B by approximately 4.5% across eight reasoning tasks compared to GRPO. (c) Critique-GRPO facilitates self-improvement via self-critiquing, obtaining 66.7% pass@1 on AIME 2024 and consistent gains across pass@k scores (k=1-256) over GRPO.

RL has been a key driver of recent advancements in improving the reasoning capabilities of LLMs (Yang et al., 2025a; DeepSeek-AI et al., 2025; OpenAI et al., 2024; OpenAI, 2025). Specifically, RL with numerical feedback—commonly in the form of scalar rewards, such as the R1-Zero

training paradigm (DeepSeek-AI et al., 2025)—enables LLMs to learn from their own generations via trial-and-error (Silver & Sutton, 2025), leading to substantial performance improvements.

Despite its success, we identify three key limitations of RL with solely numerical feedback (Section 3): (*i*) *Performance Plateaus:* Peak performance stagnates even when scaling training data by 8x. (*ii*) *Limited Effectiveness of Spontaneous Self-Reflection:* Spontaneous self-reflection during fine-tuning, often described as "Aha moments," (DeepSeek-AI et al., 2025) has limited impact on enhancing problem-solving success. (*iii*) *Persistent Failures:* Models consistently fail on certain problems despite extensive trial-and-error fine-tuning. These limitations impede efficient RL scaling. We hypothesize that these issues stem from numerical feedback's inherent constraints, which provide little insight into *why* responses succeed or fail and *how* to improve them. The limited impact of spontaneous self-reflection further compounds these challenges, highlighting the need for richer feedback mechanisms to enable more effective RL scaling up (Karpathy, 2025).

Natural language feedback (NLF), *e.g.,* textual critiques, offers a promising solution by providing detailed, targeted guidance (Saunders et al., 2022; Chen et al., 2024; McAleese et al., 2024). However, existing studies (Lightman et al., 2024; Whitehouse et al., 2025; Liu et al., 2025b) primarily use critiques for evaluation, transforming them into numerical rewards for guiding model improvement via RL algorithms such as Group Relative Policy Optimization (GRPO) (Shao et al., 2024). This transformation often discards valuable constructive information embedded within the critiques. Some studies (Chen et al., 2024; Xi et al., 2024) utilize critiques to generate refinements and fine-tune models on these refinements through supervised learning. While effective, these offline approaches are limited by their inability to support consistent exploration and real-time refinement. This raises a natural question: *Can we incorporate critiques into an online RL framework to enable LLMs to consistently and simultaneously learn from both natural language and numerical feedback?*

To address this, we first demonstrate that RL-finetuned models, even after reaching performance plateaus, can successfully refine responses to persistently failed problems when guided by critiques. This holds true with both simple indicative critiques (*e.g.,* "correct"/"incorrect") and Chain-of-Thought (CoT) critiques, which provide step-by-step evaluations (Wang et al., 2025) (Section 3). Building on this insight, we propose Critique-GRPO, an online RL framework that combines numerical and natural language feedback for effective policy optimization. As illustrated in Figure 1, Critique-GRPO enables the model to learn from both its initial sampled responses and subsequent self-refinements, guided by critiques from a reward system (either model-based or rule-based). This dual learning mechanism integrates targeted feedback while maintaining policy exploration. To enhance learning further, we incorporate a shaping function (Yan et al., 2025) that amplifies the impact of correct and unfamiliar refinements while penalizing incorrect ones. This ensures that the model not only benefits from valuable feedback but also prioritizes meaningful updates.

We evaluate Critique-GRPO on the non-reasoning models Qwen2.5-7B-Base (Qwen et al., 2025) and Qwen2.5-Math-7B-Base (Yang et al., 2024), as well as the reasoning model Qwen3-8B (Yang et al., 2025a), across five challenging mathematical reasoning tasks (in-distribution) and three scientific and general reasoning tasks (out-of-distribution). Extensive results show that Critique-GRPO significantly outperforms all compared supervised and RL-based fine-tuning methods, achieving average Pass@1 improvements of approximately 15.0%, 21.6%, and 15.0% on the three models, respectively. Additionally, Critique-GRPO demonstrates strong self-improvement capabilities through self-critiquing, *e.g.,* achieving a +16.7% Pass@1 improvement over GRPO on AIME 2024.

In summary, our contributions are three-fold: (*i*) We identify three key limitations of RL using solely numerical feedback and highlight the potential of leveraging natural language feedback to address these limitations. (*ii*) We propose Critique-GRPO, an online RL framework that enables LLMs to learn simultaneously from both initial responses and their refinements by leveraging both natural language and numerical feedback. (*iii*) We validate the efficacy of Critique-GRPO through extensive experiments, demonstrating superior performance across eight challenging reasoning tasks.

## 2 RELATED WORK

**Enhancing LLM Reasoning with Reinforcement Learning.** RL with numerical feedback has been highly effective in improving the reasoning capabilities of LLMs (OpenAI et al., 2024; DeepSeek-AI et al., 2025; Fatemi et al., 2025; Li et al., 2025; Liu et al., 2025a; Yu et al., 2025). However, numerical feedback is inherently sparse, and models often struggle to identify errors in generated responses, limiting its effectiveness in driving substantial improvements (Xi et al., 2024; Gandhi et al., 2025). To

address this limitation, recent approaches (Yan et al., 2025; Lv et al., 2025) have combined learning from high-quality expert demonstrations with online RL policy optimization. Distinct from these methods, our approach enables models to learn from diverse textual feedback during online RL. This strategy eliminates the reliance on high-quality demonstration data, offering greater flexibility and scalability. Further discussions on learning from NLF can be found in Appendix B.

## 3 LIMITATIONS OF RL WITH SOLELY NUMERICAL FEEDBACK AND THE PROMISE OF NATURAL LANGUAGE GUIDANCE

### 3.1 THREE LIMITATIONS OF LEARNING WITH SOLELY NUMERICAL FEEDBACK

In this section, we identify three key limitations of RL fine-tuning that relies solely on numerical feedback: (*i*) *performance plateaus*, (*ii*) *limited effectiveness of spontaneous self-reflection*, and (*iii*) *persistent failures*. Our investigation focuses on mathematical reasoning tasks using three models: non-reasoning models Qwen2.5-7B-Base, reasoning model Qwen3-8B, and Qwen3-8B-Base. Due to space limitations, we summarize key findings on Qwen2.5-7B-Base here, while a detailed setup and comprehensive analysis of all three models are provided in Appendix C and Appendix D.

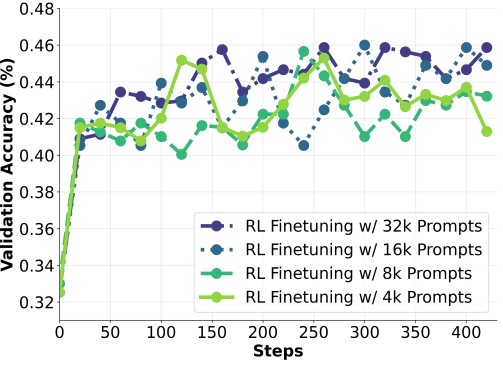 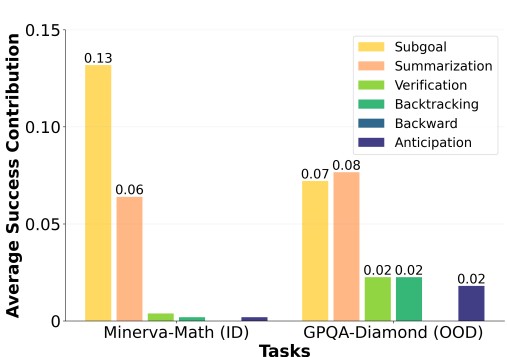

(a) RL fine-tuning dynamics of Qwen2.5-7B-Base.

(b) Reasoning behavior contributions to successful problem-solving in RL-tuned Qwen2.5-7B-Base.

Figure 2: (Left) RL fine-tuning dynamics using GRPO with numerical feedback on the validation set with varying numbers of training examples. (Right) Impact of different reasoning behaviors on solving previously failed problems after RL fine-tuning with numerical feedback on test tasks.

**RL with solely numerical feedback frequently encounters performance plateaus.** Figure 2a illustrates the RL fine-tuning dynamics of Qwen2.5-7B-Base across varying numbers of training examples. We observe that Qwen2.5-7B-Base reaches its highest performance at approximately 45-46% accuracy after 120 steps on the validation set, showing minimal improvement in peak performance, even with an 8x increase (from 4k to 32k) in the number of training prompts.

**Spontaneous self-reflection has limited impact on enhancing problem-solving success.** We characterize six key cognitive behaviors that contribute to self-improving reasoning during RL fine-tuning: subgoal setting, summarization, verification, backtracking, backward chaining (Gandhi et al., 2025), and anticipation. We categorize the first two behaviors as *planning behaviors* and the latter four as *self-reflection behaviors*. We analyze the contribution of these behaviors to solving problems that the base model was previously unable to solve. For Qwen2.5-7B-Base, we identify 87 previously unsolved problems from the Minerva-Math dataset (Lewkowycz et al., 2022) and 37 from the GPQA-Diamond dataset (Rein et al., 2024). We then examine the cognitive behaviors present in the correct responses generated by RL-finetuned models for these problems. The average success contribution of each behavior is computed as the average occurrence of each behavior across these responses, where the presence of a behavior in a single response is scored as 1 and its absence as 0. Further details are provided in Appendix C. Figure 2b presents the average success contribution of various behaviors. The results reveal that planning behaviors are the primary contributors to successful problem-solving, while spontaneous self-reflection behaviors provide minimal or significantly lower contributions across both tasks. This underscores the unreliability of relying on spontaneous self-reflection for self-correction to improve problem-solving success.

**Models exhibit persistent failures on a subset of problems despite trial-and-error fine-tuning.** In the left panel of Table 6, the best-performing RL-finetuned Qwen2.5-7B-Base, trained with 4k prompts, consistently fails on approximately 29% of training questions with pass@4 = 0.

## 3.2  PROMISE OF LEARNING FROM NATURAL LANGUAGE FEEDBACK

To overcome the limitations of purely numerical reward signals, we investigate the potential of natural language feedback to help models identify errors and refine their responses. Specifically, we examine three types of critiques: (*i*) *Indicative Critique*: A simple critique that solely indicates the correctness of the response. (*ii*) *Indicative Critique with Ground Truth (Critique w/ GT)*: A critique that provides both the binary correctness indication and the ground truth answer. (*iii*) *CoT Critique*: A model-generated critique that offers step-by-step evaluation, concluding with a binary correctness indication, but does not contain an expert demonstration. Examples of these three critique types are shown as below, detailed examples can be found in Appendix C.

---

**Three Types of Critiques**

**Indicative Critique:**  The generated solution is incorrect.

**Indicative Critique w/ GT:** The generated solution is incorrect, the ground truth is $\frac{7}{25}$.

**CoT Critique:** Let's analyze the student's solution step-by-step and identify any errors:
### Step 1: Geometry Understanding The student correctly ...
### Step 7: Precise Calculation The student attempts to ...
Conclusion: incorrect [END]

---

We utilize textual critiques to guide the refinement of the model's responses to persistently failed questions (Pass@4 = 0). Specifically, we identify failed question-response pairs from RL-finetuned models and associate each with one of the three critique types. Then we prompt the RL-finetuned models to refine their failed responses using the provided critiques. The entire process is detailed in Appendix C. Key findings for Qwen2.5-7B-Base are summarized here, while the detailed setup and comprehensive analysis are provided in Appendices C and D.

Table 1: Analysis of performance gains from critique-based self-refinement.

| Method | % Failed Questions (Pass@4=0) | Critique Type | % Valid Critiques | % Valid Refinements | % Critiqued Questions | % Questions Refined |
|---|---|---|---|---|---|---|
| RL-finetuned Qwen2.5-7B-Base | 29.07 | Indicative Critique | 100.00 | 2.09 | 100.00 | 7.05 |
| | | Indicative Critique w/ GT | 100.00 | 1.98 | 100.00 | 6.88 |
| | | CoT Critique | 60.06 | **36.47** | 95.10 | **55.37** |

**Deliberate critiques are more effective than spontaneous self-reflection for self-correction.** Incorporating all three types of critiques results in non-zero valid refinements and correctly refined questions. This demonstrates that deliberate critiques can enable both models to successfully address some persistently failed responses that cannot be resolved through spontaneous self-reflection.

**Binary correctness signals alone can provide refinement benefits.** Refinement with indicative critiques (with or without ground truth) yields some successful refinements, though at a substantially lower rate (approximately 2% valid refinements). The negligible difference between indicative critiques and critiques with ground truth indicates that providing only the ground truth answer without explanation offers little additional guidance.

**CoT Critiques facilitate effective model refinement.** Table 1 demonstrates that refinement guided by CoT critiques achieves the highest valid refinement rate (36.47%) and the largest percentage of successfully refined questions (55.37%) on Qwen2.5-7B-Base. The effectiveness of CoT critiques can be attributed to their richness, *e.g.,* providing a step-by-step explanation.

# 4 CRITIQUE-GRPO

In this section, we introduce Critique-GRPO, an online RL framework, built upon GRPO (Shao et al., 2024), that enables an LLM to learn from both natural language and numerical feedback. Before delving into the details of Critique-GRPO (Section 4.2), we briefly review GRPO (Section 4.1).

## 4.1 FROM GRPO TO CRITIQUE-GRPO

GRPO is an online RL algorithm commonly used during LLM fine-tuning. Building on PPO, it eliminates value function approximation by estimating advantages from relative performance across grouped actions. For LLM policy optimization with parameters $\theta$, GRPO operates as follows: For each question $q$ in set $Q$, responses $\{y^{(i)}\}_{i=1}^n$ are sampled from the old policy $\pi_{\text{old}}$ and scored by a reward model or rule-based system, yielding rewards $\{R^{(i)}\}_{i=1}^n$. The GRPO training objective is:

$$\mathcal{J}_{\text{GRPO}}(\theta) = \mathbb{E}_{q \sim Q, \{y^{(i)}\}_{i=1}^n \sim \pi_{\text{old}}(\cdot|q)}$$

$$\frac{1}{n} \sum_{i=1}^n \frac{1}{|y^{(i)}|} \sum_{t=1}^{|y^{(i)}|} \left\{ \min \left[ r_t^{(i)}(\theta) \hat{A}_t^{(i)}, \text{clip}(r_t^{(i)}(\theta), 1-\epsilon, 1+\epsilon) \hat{A}_t^{(i)} \right] - \beta D_{\text{KL}}[\pi_\theta || \pi_{\text{ref}}] \right\}, \tag{1}$$

where $r_t^{(i)}(\theta)$ is the probability ratio, comparing the current policy $\pi_\theta$ to the old policy $\pi_{\text{old}}$:

$$r_t^{(i)}(\theta) = \frac{\pi_\theta(y_t^{(i)}|q, y_{<t}^{(i)})}{\pi_{\text{old}}(y_t^{(i)}|q, y_{<t}^{(i)})}, \quad \hat{A}_t^{(i)} = \frac{R^{(i)} - \text{mean}(\{R^{(1)}, \dots, R^{(n)}\})}{\text{std}(\{R^{(1)}, \dots, R^{(n)}\})}. \tag{2}$$

The hyperparameter $\epsilon$ sets the clipping range for the probability ratio, preventing overly large policy updates by enforcing a pessimistic bound on performance. The hyperparameter $\beta$ regulates the KL divergence penalty, constraining the trained policy $\pi_\theta$ from deviating significantly from the reference policy $\pi_{\text{ref}}$. The advantage $\hat{A}_t^{(i)}$ for all tokens in a response is calculated by normalizing the rewards $\{R^{(i)}\}_{i=1}^n$ using the group mean and standard deviation. Following Liu et al. (2025a), we remove the terms (highlighted in gray) that may introduce biased optimization.

## 4.2 ONLINE RL WITH CRITIQUE-GRPO

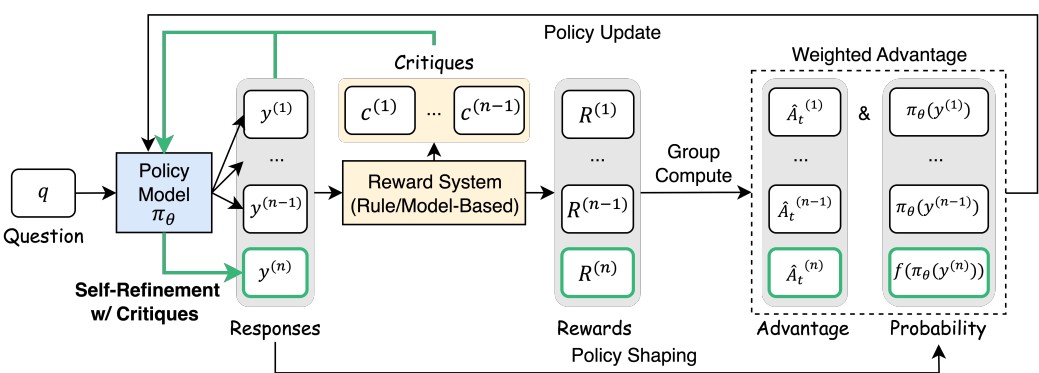

Figure 3: Overview of Critique-GRPO. Given a question, Critique-GRPO samples initial responses and then refines these responses using critiques generated by a reward system (either rule-based or model-based). These refinements are combined with the initial responses to optimize the policy within an online RL framework. A weighted advantage function, combined with policy shaping, emphasizes correct refinements while penalizing incorrect ones.

As shown in Figure 3, Critique-GRPO operates in three main steps:

**Step 1: Initial Response Sampling.** Given an LLM and a set of questions $Q = \{q\}$, we sample $n$ initial responses for each question from the old policy $\pi_{\text{old}}$: $\{y^{(i)}\}_{i=1}^n \sim \pi_{\text{old}}(\cdot \mid q)$. A reward

system evaluates these responses, generating critiques $\{c^{(i)}\}_{i=1}^n$ and scalar rewards $\{R^{(i)}\}_{i=1}^n$, with positive rewards for correct responses and zero or negative rewards for incorrect ones: $c^{(i)}, R^{(i)} \leftarrow \text{Reward}(q, y^{(i)}), \forall i$. Critique-GRPO supports two types of reward systems: (*i*) In the *rule-based* reward system, generated responses are compared to ground-truth answers using a string-matching function to compute scalar rewards: $R^{(i)} = \text{is\_equivalent}(y^{(i)}, y_{\text{GT}})$. Based on these evaluations, two indicative critiques are heuristically constructed: one without ground truth ($c_{\text{I}}^{(i)}$) and one with ground truth ($c_{\text{GT}}^{(i)}$): $c_{\text{I}}^{(i)}, c_{\text{GT}}^{(i)} \leftarrow R^{(i)}$. (*ii*) In the *model-based* reward system, a reward model $\pi_{RM}$ generates CoT critiques: $c_{\text{CoT}}^{(i)} \sim \pi_{RM}(\cdot \mid I_c, q, y^{(i)})$, where $I_c$ is the critique instruction. Binary correctness labels from these critiques are converted into scalar reward scores: $R^{(i)} \leftarrow c_{\text{CoT}}^{(i)}$.

**Step 2: Critique-Guided Self-Refinement.** Next, we prompt the LLM to generate refined responses conditioned on the question-response-critique triplet $(q, y^{(i)}, c^{(i)})$ and a refinement instruction $I_{\text{refine}}$ (detailed in Appendix S): $y_{\text{refined}}^{(i)} \sim \pi_{\text{old}}(\cdot \mid I_{\text{refine}}, q, y^{(i)}, c^{(i)})$, where $c^{(i)} \in \{c_{\text{CoT}}^{(i)}, c_{\text{GT}}^{(i)}, c_{\text{I}}^{(i)}\}$. These refinements are then evaluated by the reward system, producing corresponding rewards $\{R_{\text{refined}}^{(i)}\}_{i=1}^n$. To reduce distributional shifts caused by the refinements, a random subset of $k$ refinements is sampled from the full set $\{y_{\text{refined}}^{(i)}\}_{i=1}^n$, prioritizing correct refinements. If no correct refinements are available, incorrect refinements are sampled randomly. The sampled subset, denoted by $\{y_{\text{refined}}^{(i')}\}_{i'=1}^k$, is then combined with the initial responses to form a mixed group of responses: $\{y^{(i)}\}_{i=1}^n \cup \{y_{\text{refined}}^{(i')}\}_{i'=1}^k$.

**Step 3: Online Policy Optimization.** Finally, we fine-tune the model on the mixed set of initial and refined responses using scalar rewards. The training objective is:

$$\mathcal{J}_{\text{Critique-GRPO}}(\theta) = \mathbb{E}_{q \sim Q, \{y^{(i)}\}_{i=1}^n \sim \pi_{\text{old}}(\cdot|q), \{y_{\text{refined}}^{(i')}\}_{i'=1}^k \sim \pi_{\text{old}}(\cdot|q)} \Bigg[$$

$$\underbrace{\frac{1}{n} \sum_{i=1}^n \sum_{t=1}^{|y^{(i)}|} \min\left[ r_t^{(i)}(\theta) \hat{A}_t^{(i)}, \text{clip}(r_t^{(i)}(\theta), 1-\epsilon, 1+\epsilon) \hat{A}_t^{(i)} \right]}_{\text{Objective for Initial Responses}}$$

$$+ \underbrace{\frac{1}{k} \sum_{i'=1}^k \sum_{t=1}^{|y_{\text{refined}}^{(i')}|} \min\left[ r_{\text{refined},t}^{(i')}(\theta) A_t^{(i')}, \text{clip}(r_{\text{refined},t}^{(i')}(\theta), 1-\epsilon, 1+\epsilon) A_t^{(i')} \right]}_{\text{Objective for Refined Responses}} \Bigg].$$

$$(3)$$

where the advantages $A_t^{(i)}, A_t^{(i')}$ for all tokens in a response are defined as:

$$A_t^{(i/i')} = R^{(i/i')} - \text{mean}\left( \{R^{(i)}\}_{i=1}^n \cup \{R_{\text{refined}}^{(i')}\}_{i'=1}^k \right). \tag{4}$$

The terms $r_t^{(i)}(\theta)$ and $f(r_{\text{refined},t}^{(i')}(\theta))$ represent the token-level probability ratios:

$$r_t^{(i)}(\theta) = \frac{\pi_\theta(y_t^{(i)}|q, y_{<t}^{(i)})}{\pi_{\text{old}}(y_t^{(i)}|q, y_{<t}^{(i)})}, \quad f(r_{\text{refined},t}^{(i')}(\theta)) = \frac{\pi_\theta(y_{\text{refined},t}^{(i')}|q, y_{\text{refined},<t}^{(i')})}{\pi_\theta(y_{\text{refined},t}^{(i')}|q, y_{\text{refined},<t}^{(i')}) + \gamma}. \tag{5}$$

We employ a shaping function $f(\cdot)$ (Yan et al., 2025) ($0 < \gamma < 1$) to reweight gradients, assigning higher importance to low-probability tokens in refined responses. This mechanism encourages the model to learn effectively from unfamiliar (*i.e.,* low-probability under the current policy) yet correct refinements, while penalizing incorrect ones. To enable more substantial policy updates, we remove the KL-divergence penalty term from the original GRPO formulation. These adjustments facilitate efficient learning from refinements. A detailed summary of Critique-GRPO is provided in Appendix F. More details can be found in Appendix C.

## 5 EXPERIMENTS

### 5.1 EXPERIMENTAL SETUP

**Datasets and Evaluation Metrics.** We use randomly sampled subsets of 4k examples from a reorganized 46k subset (Yan et al., 2025) of OpenR1-Math-220k (Bakouch et al., 2025) as the training set. For validation, we utilize the curated validation set provided by (Yan et al., 2025). The model is evaluated on five established mathematical reasoning benchmarks: MATH-500 (Hendrycks et al., 2021), Minerva-Math, OlympiadBench (He et al., 2024a), AIME 2024/2025 (Li et al., 2024), and AMC 2023 (Li et al., 2024). To analyze out-of-distribution generalization, we further evaluate the model on three reasoning tasks spanning scientific and general domains: TheoremQA (Chen et al., 2023), GPQA-Diamond, and MMLU-Pro (Wang et al., 2024). For evaluation, we use greedy decoding (temperature = 0) and report pass@1 over three runs.

**Compared Methods.** We compare Critique-GRPO against the following approaches. All differences are considered significant at $p < 0.01$. During RL fine-tuning, we utilize binary scalar rewards (+1 for correct responses and 0 for incorrect ones). *1. Supervised Learning-based Finetuning:* (*i*) *Supervised Fine-tuning (SFT)*: Finetuning on high-quality annotated training data. (*ii*) *Reward Ranked Finetuning (RAFT)* (Dong et al., 2023): Finetuning on correct initial responses. (*iii*) *Refinement Finetuning (Refinement FT)* (Chen et al., 2024): Finetuning on correct refinements generated conditionally on CoT critiques. (*iv*) *Critique Finetuning (Critique FT)* (Wang et al., 2025): Finetuning on CoT critiques. (*v*) *Critique-in-the-Loop Finetuning (CITL-FT)* (Xi et al., 2024): Fine-tuning on both correct initial responses and their refinements generated conditionally on CoT critiques. *2. Reinforcement Learning-based Finetuning:* (*vi*) *R1-GRPO* (DeepSeek-AI et al., 2025): Fine-tuning on initial responses using the GRPO algorithm with binary scalar rewards. (*vii*) *R1-Dr.GRPO* (Liu et al., 2025a): Fine-tuning on initial responses using the Dr.GRPO algorithm, which removes optimization bias terms, with binary scalar rewards. We implement Critique-GRPO using asynchronous rollouts via the VERL framework (Sheng et al., 2024) for computational efficiency. Implementation details are in Appendix H.

### 5.2 MAIN RESULTS

Table 2 presents the evaluation results, highlighting the following key observations:

**Incorporating natural language feedback into online RL enhances policy optimization.** Critique-GRPO consistently outperforms both supervised learning-based and RL-based fine-tuning approaches on Qwen2.5-7B-Base and Qwen3-8B across all tasks. Specifically, Critique-GRPO surpasses R1-GRPO and R1-Dr.GRPO, improving average pass@1 scores by approximately 4.4 points (42.66% $\rightarrow$ 47.08%) on Qwen2.5-7B-Base and 3.8 points (64.46% $\rightarrow$ 68.26%) on Qwen3-8B. In addition, we provide a comprehensive qualitative analysis in Appendix P.

**Online self-refinements are more effective than offline self-refinements.** Critique-GRPO (CoT critique) substantially outperforms Refinement FT by approximately 11.9 points (47.08% *v.s.* 35.21%) and 8.81 points (68.26% *v.s.* 59.45%) in average pass@1 on Qwen2.5-7B-Base and Qwen3-8B, respectively. Furthermore, it surpasses CITL-FT by approximately 11.4 points (47.08% *v.s.* 35.66%) and 12.4 points (68.26% *v.s.* 55.84%) on Qwen2.5-7B-Base and Qwen3-8B, respectively.

**Higher-quality refinements, guided by richer critiques, result in more effective policy optimization.** Critique-GRPO utilizing CoT critiques consistently outperforms its variant with indicative critiques across all tasks, achieving average pass@1 improvements of +1.8-2.4% on Qwen2.5-7B-Base and +2.0-2.3% on Qwen3-8B. This improvement stems from the superiority of CoT critiques to guide the model in generating more effective refinements, as discussed in Section 3.

### 5.3 INVESTIGATION ON MATH-CENTRIC BACKBONE MODELS

We validate the efficacy of Critique-GRPO on the math-centric backbone model, Qwen2.5-Math-7B-Base. Specifically, we compare its performance against three representative RL fine-tuning approaches based solely on numerical feedback: *SimpleRL-Zero* (Zeng et al., 2025), *PRIME-Zero* (Cui et al., 2025a), and *Oat-Zero* (Liu et al., 2025a). Details are shown in Appendix H.

**Natural language feedback effectively overcome performance plateaus faced by solely relying on numerical feedback.** As shown in Table 3, Critique-GRPO achieves an average pass@1 improvement

Table 2: Evaluation results (Pass@1) on mathematical reasoning (ID) and scientific and general (OOD) reasoning tasks. "Expert Demo." refers to expert demonstrations. "Num. Feedback" and "Lang. Feedback" denote numerical and natural language feedback, respectively.

| Method | Supervision | | | Math (ID) | | | | | Science & General (OOD) | | | Avg. |
|---|---|---|---|---|---|---|---|---|---|---|---|---|
| | Expert Demo. | Num. FB | Lang. FB | MATH 500 | Minerva MATH | Olympiad Bench | AMC23 | AIME24 | Theorem QA | GPQA Diamond | MMLU Pro | |
| ***Non-Reasoning Model*** | | | | | | | | | | | | |
| Qwen2.5-7B-Base | - | - | - | 60.80 | 20.20 | 30.40 | 35.00 | 13.30 | 21.60 | 28.79 | 46.24 | 32.04 |
| *Supervised Learning-based Finetuning* | | | | | | | | | | | | |
| + SFT | ✓ | ✗ | ✗ | 61.60 | 24.30 | 23.40 | 40.00 | 6.70 | 26.50 | 30.30 | 51.49 | 33.04 |
| + RAFT | ✗ | ✓ | ✗ | 67.00 | 19.50 | 32.40 | 50.00 | 10.00 | 24.40 | 23.74 | 47.12 | 34.27 |
| + Refinement FT | ✗ | ✓ | ✓ | 65.80 | 21.30 | 32.10 | 47.50 | 13.30 | 24.40 | 29.80 | 47.51 | 35.21 |
| + Critique FT | ✗ | ✗ | ✓ | 66.00 | 19.10 | 29.30 | 47.50 | 13.3 | 29.60 | 28.79 | 44.46 | 34.76 |
| + CITL-FT | ✗ | ✓ | ✓ | 70.20 | 19.90 | 30.70 | 42.50 | 16.70 | 28.70 | 28.28 | 48.31 | 35.66 |
| *Reinforcement Learning-based Finetuning (w/o SFT)* | | | | | | | | | | | | |
| + R1-GRPO | ✗ | ✓ | ✗ | 74.00 | 32.00 | 38.50 | 42.50 | 16.70 | 40.60 | 33.33 | 51.81 | 41.18 |
| + R1-Dr.GRPO | ✗ | ✓ | ✗ | **78.40** | 34.90 | 39.90 | 40.00 | 13.30 | 43.10 | 38.89 | 52.83 | 42.66 |
| + Critique-GRPO (Ours) (Indicative Critique) | ✗ | ✓ | ✓ | 76.00 | 36.00 | 41.00 | 55.00 | 13.30 | 41.80 | 37.88 | **55.97** | 44.62 |
| + Critique-GRPO (Ours) (Critique w/ GT) | ✗ | ✓ | ✓ | 76.80 | 35.70 | 39.60 | 62.50 | 10.00 | 44.00 | **38.89** | 54.88 | 45.30 |
| + Critique-GRPO (Ours) (CoT Critique) | ✗ | ✓ | ✓ | 77.80 | **36.80** | **42.40** | 62.50 | **20.00** | 44.00 | 37.88 | 55.28 | **47.08** |
| ***Reasoning Model (w/ Thinking)*** | | | | | | | | | | | | |
| Qwen3-8B | - | - | - | 82.00 | 41.20 | 44.10 | 67.50 | 40.00 | 46.90 | 35.86 | 68.25 | 53.23 |
| *Supervised Learning-based Finetuning* | | | | | | | | | | | | |
| + SFT | ✓ | ✗ | ✗ | 83.20 | 43.80 | 46.40 | 82.50 | 40.00 | 48.90 | 38.38 | 66.81 | 56.25 |
| + RAFT | ✗ | ✓ | ✗ | 82.80 | 44.10 | 46.40 | 75.00 | 36.70 | 46.80 | 37.88 | 69.00 | 54.84 |
| + Refinement FT | ✗ | ✓ | ✓ | 87.40 | 46.00 | 54.50 | 80.00 | 40.00 | 55.40 | 45.45 | 66.82 | 59.45 |
| + Critique FT | ✗ | ✗ | ✓ | 84.40 | 37.10 | 49.80 | 80.00 | 36.70 | 46.40 | 35.35 | 64.10 | 54.23 |
| + CITL-FT | ✗ | ✓ | ✓ | 85.00 | 43.00 | 46.80 | 70.00 | 43.30 | 48.00 | 41.92 | 68.73 | 55.84 |
| *Reinforcement Learning-based Finetuning (w/o SFT)* | | | | | | | | | | | | |
| + R1-GRPO | ✗ | ✓ | ✗ | 91.00 | 52.60 | 65.60 | 82.50 | 50.00 | 57.90 | 40.40 | 70.00 | 63.75 |
| + R1-Dr.GRPO | ✗ | ✓ | ✗ | 91.20 | 51.10 | 63.60 | 82.50 | 53.30 | 59.00 | 44.44 | 70.51 | 64.46 |
| + Critique-GRPO (Ours) (Indicative Critique) | ✗ | ✓ | ✓ | 91.00 | 47.80 | 63.30 | 85.00 | **63.30** | **60.40** | 47.47 | 70.00 | 66.03 |
| + Critique-GRPO (Ours) (Critique w/ GT) | ✗ | ✓ | ✓ | 92.00 | 50.00 | 66.80 | 87.50 | 56.70 | 59.00 | 47.47 | **70.87** | 66.29 |
| + Critique-GRPO (Ours) (CoT Critique) | ✗ | ✓ | ✓ | **92.00** | **52.90** | **66.80** | **92.50** | **63.30** | 60.10 | **47.98** | 70.47 | **68.26** |

Table 3: Investigation of RL finetuning with Critique-GRPO on Qwen2.5-Math-7B-Base (Yang et al., 2024). Results marked with an asterisk are cited from Yan et al. (2025).

| Method | Training Data Volume | Math (ID) | | | | | Science & General (OOD) | | | Avg. |
|---|---|---|---|---|---|---|---|---|---|---|
| | | MATH 500 | Minerva MATH | Olympiad Bench | AMC23 | AIME24 | Theorem QA | GPQA Diamond | MMLU Pro | |
| Qwen2.5-Math-7B-Base | - | 51.20 | 13.20 | 17.60 | 47.50 | 13.30 | 26.40 | 26.77 | 39.70 | 29.46 |
| + SimpleRL-Zero* | 46k | 76.00 | 25.00 | 34.70 | 54.90 | 27.00 | - | 23.20 | 34.50 | - |
| + PRIME-Zero* | 46k | 81.40 | 39.00 | 40.30 | 54.00 | 17.00 | - | 18.20 | 32.70 | - |
| + Oat-Zero* | 46k | 78.00 | 34.60 | 43.40 | 61.20 | **33.40** | - | 23.70 | 41.70 | - |
| + Critique-GRPO (Ours) (CoT-Critique) | 4k | **84.20** | 40.10 | 48.60 | 70.00 | 30.00 | 51.40 | 40.40 | 43.79 | **51.06** |

of +21.6% over Qwen2.5-Math-7B-Base using only 4k RL training prompts. This significantly outperforms other approaches that rely solely on numerical feedback, which require 46k prompts.

## 5.4 SELF-IMPROVEMENT VIA SELF-CRITIQUING

We explore the potential of Critique-GRPO in enabling an LLM's self-improvement through self-critiquing, resulting in Critique-GRPO (self-critique). Further details are provided in Appendix I.

Table 4: Potential for self-improvement via self-critiquing using Critique-GRPO on Qwen3-8B.

| Method | Math (ID) | | | | | Science & General (OOD) | | | Avg. | AIME24 (Avg@32) | AIME25 (Avg@32) |
|---|---|---|---|---|---|---|---|---|---|---|---|
| | MATH 500 | Minerva MATH | Olympiad Bench | AMC23 | AIME24 | Theorem QA | GPQA Diamond | MMLU Pro | | | |
| Qwen3-8B (w/ Think) | 82.00 | 41.20 | 44.10 | 67.50 | 40.00 | 46.90 | 35.86 | 68.25 | 53.23 | 30.00 | 23.30 |
| + SFT | 83.20 | 43.80 | 46.40 | 82.50 | 40.00 | 48.90 | 38.38 | 66.81 | 56.25 | 33.30 | 40.00 |
| + R1-GRPO | 91.00 | 52.60 | 65.60 | 82.50 | 50.00 | 57.90 | 40.40 | 70.00 | 63.75 | 53.33 | 43.30 |
| + Critique-GRPO (Ours) (Self-Critique) | **92.60** | **52.60** | **66.20** | **95.00** | **60.00** | **60.60** | **47.98** | **70.03** | **68.13** | **60.00** | **46.70** |

**Critique-GRPO facilitates self-improvement through self-critiquing.** In Table 4, RL fine-tuning with Critique-GRPO (self-critique) improves pass@1 by an average of +4.5% and +12.0% compared to R1-GRPO and SFT, respectively.

**Self-critiquing aids effective exploration.** Figures 4 and 11b highlight the consistently superior performance of Critique-GRPO (self-critique) across pass@k metrics on AIME24 and AIME25, with $k$ ranging from 1 to 256, indicating genuine improvements over both R1-GRPO and SFT. Additional results are provided in Appendix I, with an exploration of weak-to-strong generalization in Appendix J, and an analysis of the impact of policy shaping on RL fine-tuning in Appendix O.

## 5.5 INVESTIGATION OF POLICY EXPLORATION DURING RL FINETUNING

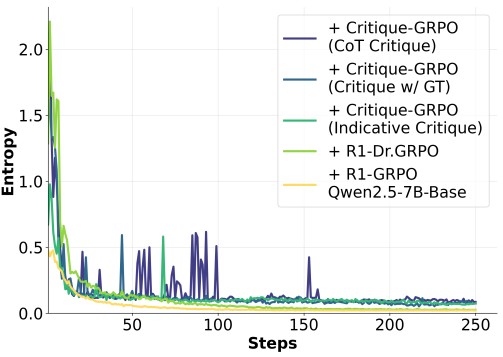

(a) Entropy dynamics on Qwen2.5-7B-Base.

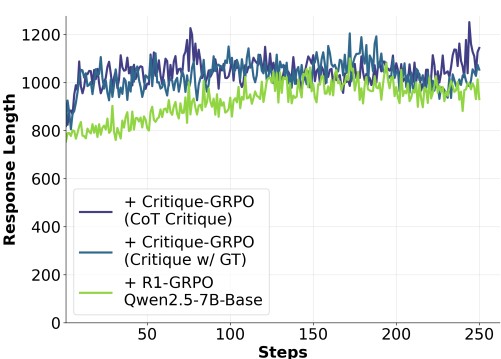

(b) Response length changes on Qwen2.5-7B-Base.

Figure 5: Entropy dynamics (left) and response length changes (right) over training steps for various RL-based fine-tuning approaches on Qwen2.5-7B-Base.

To investigate policy exploration, we analyze two primary aspects: (*i*) entropy dynamics and (*ii*) changes in response length during RL fine-tuning.

**Learning from natural language feedback helps sustain exploration.** As shown in Figure 5a, the policy entropy of Critique-GRPO generally remains higher than that of R1-GRPO and R1-Dr.GRPO, suggesting more consistent exploration. The peaks in Critique-GRPO's entropy dynamics (before step 200) likely occur when its self-generated refinements deviate significantly from the initial sampled responses, leading to increased entropy and potentially beneficial distributional shifts. The subsequent decrease in entropy indicates that the model quickly internalizes these refinements, reducing the distributional deviation. This dynamic aligns with the observation that rare actions with high advantage can increase policy entropy (*i.e.*, unfamiliar but correct responses with high rewards promote *effective exploration*), whereas

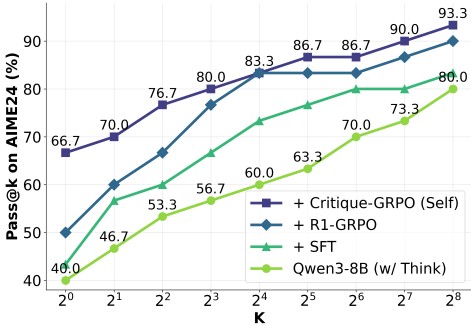

Figure 4: Comparison of Pass@k for self-improvement via Critique-GRPO (self-critique), R1-GRPO, SFT, and the base model Qwen3-8B on AIME24. Results for AIME25 are shown in Figure 11.

high-probability actions with high advantage tend to reduce entropy (Cui et al., 2025b). Combined with the results in Table 2, the superior performance of Critique-GRPO over R1-Dr.GRPO and R1-GRPO highlights the importance of maintaining a certain level of entropy for better performance.

**Critique-GRPO facilitates concise reasoning.** In Figure 5b, Critique-GRPO achieves superior performance (Table 2) while minimally increasing response length on Qwen2.5-7B-Base. This efficiency likely stems from its critique mechanism, which enables precise error identification and refinement, reducing the need for verbose reasoning. More analyses can be found in Appendix K.

## 5.6 EXPLORATION WITH DIVERSE BACKBONE AND CRITIQUE MODELS

Table 5: Evaluation of Critique-GRPO with varying policy model architectures, scales, and critique models. (The decoding budget is set to 8192 tokens for Llama-3.2-3B-Instruct and Qwen2.5-7B-Base, and 16384 tokens for Qwen3-32B.)

| Method | Critique Model | Math (ID) | | | | | Science & General (OOD) | | | Avg. |
|---|---|---|---|---|---|---|---|---|---|---|
| | | MATH 500 | Minerva MATH | Olympiad Bench | AMC23 | AIME24 | Theorem QA | GPQA Diamond | MMLU Pro | |
| *Evaluation with varying policy model architectures* | | | | | | | | | | |
| Llama-3.2-3B-Instruct | - | 46.60 | 17.30 | 15.30 | 27.50 | 13.30 | 21.80 | 22.22 | 38.16 | 22.52 |
| R1-GRPO | - | 53.60 | 19.50 | 18.80 | 27.50 | 13.30 | 24.60 | 29.29 | 40.72 | 24.79 |
| Critique-GRPO | GPT-4o | **58.80** | **25.70** | **24.10** | **35.00** | **20.00** | **31.10** | **36.87** | **43.79** | **29.86** |
| *Evaluation with varying policy model scales* | | | | | | | | | | |
| Qwen3-32B | - | 91.80 | 53.70 | 62.70 | 92.50 | 63.30 | 62.40 | 45.96 | 77.06 | 68.68 |
| R1-GRPO | - | 93.00 | 54.00 | 66.40 | 97.50 | 80.00 | 64.00 | 56.57 | 77.97 | 73.68 |
| Critique-GRPO | GPT-4o | **96.40** | **59.60** | **69.30** | **100.00** | **90.00** | **66.40** | **60.61** | **78.49** | **77.60** |
| *Evaluation with varying critique models* | | | | | | | | | | |
| Qwen2.5-7B-Base | - | 60.80 | 20.20 | 30.40 | 35.00 | 13.30 | 21.60 | 28.79 | 46.24 | 32.04 |
| R1-GRPO | - | 74.00 | 32.00 | 38.50 | 42.50 | 16.70 | 40.60 | 33.33 | 51.81 | 41.18 |
| Critique-GRPO | Llama3.1-405B | 78.40 | 37.90 | 41.00 | **62.50** | 16.70 | 42.80 | **38.89** | 56.12 | 46.79 |
| Critique-GRPO | GPT-4o | 77.80 | 36.80 | 42.40 | **62.50** | 20.00 | **44.00** | 37.88 | 55.28 | 47.08 |
| Critique-GRPO | DeepCritic-7B | **79.20** | **40.10** | **44.00** | 60.00 | **23.30** | 43.10 | 36.87 | **57.28** | **47.98** |

We validate the robustness of Critique-GRPO across three dimensions: (*i*) *Testing on a different model architecture*: Specifically, Llama-3.2-3B-Instruct (Grattafiori et al., 2024). (*ii*) *Testing on a different model scale*: Specifically, Qwen3-32B (Yang et al., 2025a). (*iii*) *Evaluating compatibility with alternative critique models for CoT critiques*: Including Llama3.1-405B (Grattafiori et al., 2024) and the fine-tuned off-the-shelf model DeepCritic-7B-RL1.5-PRM800K (Yang et al., 2025b).

The upper section of Table 5 highlights that Critique-GRPO consistently outperforms GRPO on all eight reasoning tasks when tested with Llama-3.2-3B-Instruct and Qwen3-32B, achieving average gains of +5.1 and +4.0 points, respectively. These results confirm its robustness across different model architectures and scales. The lower section demonstrates Critique-GRPO's compatibility with both proprietary and fine-tuned critique models, yielding average improvements of +5.6 to +6.8 points over GRPO. These findings underscore the effectiveness and versatility of Critique-GRPO across diverse critique models.

## 6 CONCLUSION

In this paper, we first identify three key challenges faced by RL approaches that rely solely on numerical feedback. We then propose Critique-GRPO, an online RL framework that enables LLMs to learn from both natural language and numerical feedback simultaneously for effective policy optimization. Specifically, Critique-GRPO facilitates learning from initial model responses and critique-guided self-refinements while preserving exploration. Extensive experiments demonstrate that Critique-GRPO consistently outperforms RL-based methods relying solely on numerical feedback across eight challenging reasoning tasks, paving the way for scaling RL with natural language feedback. Future work could explore leveraging Critique-GRPO to fine-tune LLMs using real-world human-machine interaction logs and real-time human feedback, enabling continuous improvement after deployment in real-world scenarios (Sutton, 2025).

ETHICS STATEMENT

Throughout our research, we have adhered to the ICLR Code of Ethics, prioritizing privacy, fairness, and individual well-being. All benchmark datasets used were strictly for research purposes and contained no personally identifiable information to safeguard privacy. Prompts were carefully designed to minimize bias or discriminatory language and reduce potential negative impacts. Additionally, model-generated responses were verified to ensure they were free from offensive content, misinformation, and personally identifiable information.

REPRODUCIBILITY STATEMENT

We provide a detailed description of the proposed framework in Appendix G, the utilized prompts in Appendix S, and comprehensive details on experimental setups, training framework, implementation, and hyperparameters in Appendix G, Appendix C, and Appendix H to facilitate reproducibility. We will release all related data, code, and models to support the research community.

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

CONTENTS

## A    THE USE OF LARGE LANGUAGE MODELS (LLMS)

In this paper, we utilized LLMs, specifically GPT-4o (Hurst et al., 2024), for two limited purposes: (1) as a reward model or judge model for academic experimentation, as described in Section 3 and Section 5; and (2) to assist in polishing the manuscript's language for clarity and fluency. LLMs played no role in research ideation, experimental design, or drafting the initial version of the paper. The authors take full responsibility for all content presented in this submission.

## B    ADDITIONAL DISCUSSION ON RELATED WORK

**Learning from Natural Language Feedback.**    Natural Language Feedback (NLF), provided as textual critiques, serves as a powerful tool for improving LLMs.  NLF delivers detailed and targeted insights into flaws in model-generated outputs, facilitating accurate evaluation and response refinement (Saunders et al., 2022; Chen et al., 2024).  Many existing methods translate NLF into numerical reward signals to fine-tune models using RL (Kim et al., 2023; Whitehouse et al., 2025; Liu et al., 2025b; Lightman et al., 2024; Ouyang et al., 2022; Casper et al., 2023; Rafailov et al., 2024). Other approaches focus on learning directly from NLF by fine-tuning models to mimic the provided feedback (Hancock et al., 2019; Wang et al., 2025). Some strategies employ dedicated refinement models (Chen et al., 2024), while others leverage the policy model itself (Xi et al., 2024) to use feedback for response refinement.  These correctly refined responses are then utilized for further optimization, typically through supervised fine-tuning. In contrast, our approach enables LLMs to learn directly from real-time NLF to refine their responses while maintaining online exploration through RL. This integration of textual feedback with online RL enhances the model's performance more effectively.

**Enhancing LLM Reasoning with Reinforcement Learning.**    Reinforcement Learning (RL) has demonstrated significant effectiveness in improving the reasoning capabilities of Large Language Models (LLMs) (OpenAI et al., 2024; DeepSeek-AI et al., 2025; Fatemi et al., 2025; Li et al., 2025). This is commonly achieved by fine-tuning models on complex reasoning tasks to encourage diverse and robust reasoning behaviors (Gandhi et al., 2025; Yue et al., 2025). Recent advancements leverage RL with numerical feedback, such as assigning positive rewards (*e.g.,* +1) for correct responses and negative rewards (*e.g.,* -1) for incorrect ones (OpenAI et al., 2024; DeepSeek-AI et al., 2025; Liu et al., 2025a; Yu et al., 2025). These methods often employ online policy optimization algorithms, including Proximal Policy Optimization (PPO) (Schulman et al., 2017), Group Relative Policy Optimization (GRPO) (Shao et al., 2024), REINFORCE (Williams, 1992), and Decoupled Clip and Dynamic Sampling Policy Optimization (DAPO) (Yu et al., 2025). However, numerical feedback tends to be sparse, and models often struggle with tasks that exceed their current knowledge boundaries, limiting the potential for substantial improvement (Xi et al., 2024; Gandhi et al., 2025). To address this limitation, recent approaches have incorporated high-quality expert demonstrations alongside online exploration (Yan et al., 2025). In contrast, our approach enables models to refine their outputs through the integration of various textual feedback, which helps identify and address potential errors. This combination of textual feedback with online exploration for policy optimization enhances flexibility and scalability.

## C  LIMITATIONS OF RL WITH NUMERICAL FEEDBACK AND THE PROMISE OF NATURAL LANGUAGE GUIDANCE (DETAILED VERSION)

### C.1  THREE LIMITATIONS OF LEARNING WITH NUMERICAL FEEDBACK

We investigate the limitations of fine-tuning with RL relying solely on numerical feedback from three key perspectives: (*i*) How best performance improves as the number of training examples increases. (*ii*) How cognitive behaviors contribute to improving successful problem-solving. (*iii*) The model's ability to solve previously failed problems through trial-and-error.

**Setup.** We conduct experiments on non-reasoning models, Qwen2.5-7B-Base (Qwen et al., 2025) and Qwen3-8B-Base (Yang et al., 2025a), and a reasoning model, Qwen3-8B (Yang et al., 2025a), for mathematical reasoning tasks. Specifically, we fine-tune the models using GRPO (Shao et al., 2024) with numerical feedback.[1]

*Datasets and Evaluation Metrics.* We utilize randomly sampled subsets of 4k, 8k, 16k, and 32k examples from a reorganized 46k subset (Yan et al., 2025) of OpenR1-Math-220k (Bakouch et al., 2025). The prompts are sourced from NuminaMath 1.5 (Li et al., 2024), while the ground truth chain-of-thought (CoT) reasoning paths are generated by Deepseek-R1 (DeepSeek-AI et al., 2025). Unless otherwise specified, experiments primarily use 4k training prompts. For validation, we randomly sample 500 examples from the validation set curated by (Yan et al., 2025), which includes examples from Olympiad Bench (He et al., 2024b), MATH (Hendrycks et al., 2021), Minerva-Math (Lewkowycz et al., 2022), AIME 2024 (Li et al., 2024), and AMC 2023 (Li et al., 2024). To provide a comprehensive evaluation, we assess performance on in-distribution (ID) tasks using Minerva-Math (Lewkowycz et al., 2022) and on out-of-distribution (OOD) tasks using GPQA-Diamond, which covers physics, chemistry, and biology (Rein et al., 2024). For evaluation, we employ greedy decoding (temperature = 0) and report accuracy as pass@1, following prior work (Zeng et al., 2025; Yang et al., 2025a; Liu et al., 2025a).[2]

*Reward Design.* We employ rule-based evaluation to provide numerical feedback (scalar rewards), using Math-Verify[3] to validate the correctness of generated answers against ground truth during fine-tuning. Binary rewards are assigned as follows: $+1$ for correct final answers and $0$ for incorrect ones. These rewards serve as a proxy for assessing the accuracy of generated responses.

*Implementation Details.* Our implementation leverages the VERL library (Sheng et al., 2024) and samples four candidate responses per prompt during fine-tuning.

**Results.  RL with solely numerical feedback frequently encounters performance plateaus.** Figure 6 illustrates the RL fine-tuning dynamics of Qwen2.5-7B-Base and Qwen3-8B across varying numbers of training examples. On-policy RL frequently stagnates, as reflected in validation set accuracy: Qwen2.5-7B-Base reaches its highest performance at approximately 45-46% accuracy after 120 steps (Figure 6a), while Qwen3-8B plateaus at 65-67% accuracy after 200 steps (Figure 6b). Notably, both models show minimal improvement in peak performance, even with an 8x increase in the number of training prompts. This highlights a significant challenge in the efficient scaling of RL.

**Spontaneous self-reflection has limited impact on enhancing problem-solving success.** Cognitive behaviors are widely recognized as key contributors to successful complex reasoning (DeepSeek-AI et al., 2025; Gandhi et al., 2025). In particular, increased self-reflection behaviors after RL fine-tuning, which mimic humans reflecting on past experiences and refining their approach to reach a solution (commonly referred to as the "Aha moment" (DeepSeek-AI et al., 2025)), have drawn significant attention. However, does spontaneous self-reflection play the most critical role in improved performance?

To address this question, we characterize six key cognitive behaviors that contribute to self-improving reasoning during RL fine-tuning:

---

[1]GRPO is used without loss of generality, as RL algorithms such as PPO and GRPO exhibit comparable performance.

[2]The pass@k metric represents the percentage of problems where the model produces a correct solution within its first $k$ attempts.

[3]https://github.com/huggingface/Math-Verify

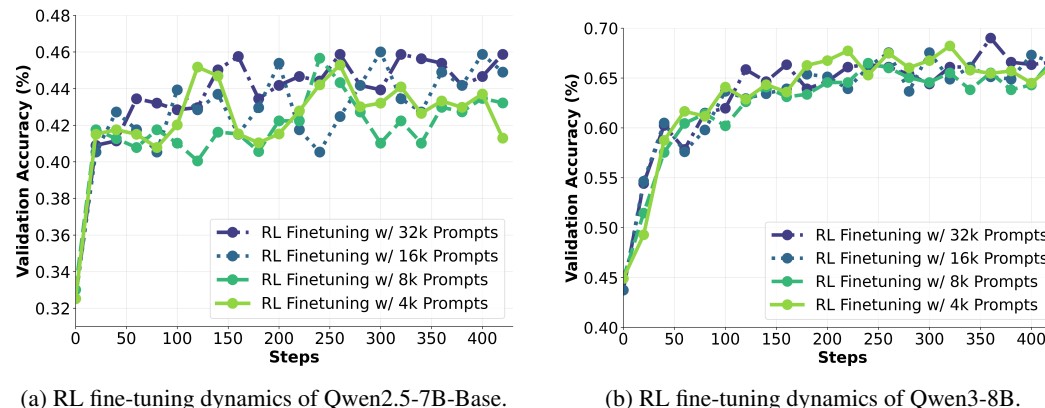

(a) RL fine-tuning dynamics of Qwen2.5-7B-Base.    (b) RL fine-tuning dynamics of Qwen3-8B.

Figure 6: RL fine-tuning dynamics of Qwen2.5-7B-Base (left) and Qwen3-8B (w/ Thinking) (right) using GRPO with numerical feedback on the validation set with varying numbers of training examples.

- **Subgoal Setting**: Decomposing complex problems into smaller, manageable subtasks.
- **Summarization**: Summarizing the current state by identifying completed subtasks and determining the next steps in reasoning.
- **Verification**: Systematically checking intermediate results or computations to ensure correctness.
- **Backtracking**: Identifying errors or dead-ends in reasoning and revising previous methods or approaches.
- **Backward Chaining**: Reasoning from desired outcomes back to the initial inputs or steps required to achieve the result. This is particularly useful for multiple-choice questions with provided answer options (Gandhi et al., 2025).
- **Anticipation**: Anticipating potential inaccuracies or exploring alternative solutions to a problem.

We categorize the first two behaviors as **planning behaviors** and the remaining four as **self-reflection behaviors**. To analyze their contributions, we evaluate problems previously unsolved by the base model. For Qwen2.5-7B-Base, we identify 87 unsolved problems from the Minerva-Math dataset and 37 from the GPQA-Diamond dataset. For Qwen3-8B, we identify 33 unsolved problems from the Minerva-Math dataset and 15 from the GPQA-Diamond dataset. We then examine the cognitive behaviors exhibited in the correct responses generated by RL-finetuned models for these problems.

To detect these behaviors, we use GPT-4o (Hurst et al., 2024) as an automatic judge with manually crafted prompts (Appendix S). Each behavior $b_i$ is scored as:

$$s_i = \begin{cases} 1 & \text{if behavior } b_i \text{ is present,} \\ 0 & \text{otherwise.} \end{cases} \tag{6}$$

The average success contribution of each behavior is computed as follows:

$$\text{Contribution}(b_i) = \frac{\sum_{j=1}^{N} s_{i,j}}{N}, \tag{7}$$

where $N$ is the total number of analyzed responses, and $s_{i,j}$ indicates whether behavior $b_i$ appears in response $j$. Further details are provided in Appendix R.

Figure 7 presents the average success contribution of various behaviors, showing that planning behaviors are the primary contributors to successful problem-solving, while self-reflection behaviors contribute less in both the mathematical (Minerva-Math) and STEM (GPQA-Diamond) domains. In Figure 7a, self-reflection behaviors barely contribute in the mathematical domain for the non-reasoning model. Thanks to extensive training on expert demonstrations with diverse reasoning

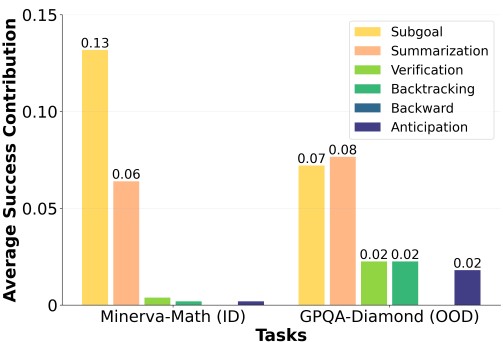 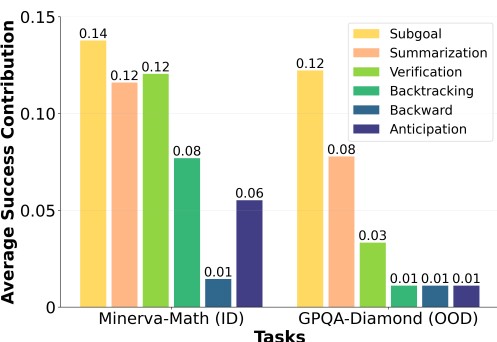

(a) Reasoning behavior contributions to successful problem-solving in RL-tuned Qwen2.5-7B-Base.

(b) Reasoning behavior contributions to successful problem-solving in RL-tuned Qwen3-8B.

Figure 7: Impact of different reasoning behaviors on solving previously failed problems after RL fine-tuning with numerical feedback for Qwen2.5-7B-Base (left) and Qwen3-8B (right) on test tasks.

behaviors in the mathematical domain (Yang et al., 2025a), the reasoning model shows that self-reflection behaviors make a noticeable contribution (Figure 7b). Nevertheless, self-correction-related behaviors, such as backtracking, backward chaining, and anticipation, still contribute considerably less. These observations suggest the limited effectiveness of spontaneous self-reflection. Further analysis on Qwen3-8B-Base is provided in Appendix D. This underscores the unreliability of relying on spontaneous self-reflection for self-correction to improve problem-solving success.

**Models exhibit persistent failures on a subset of problems despite trial-and-error fine-tuning.** We evaluated the best-performing RL-finetuned Qwen2.5-7B-Base and Qwen3-8B models on 4k training prompts. As shown in the left panel of Table 6, these models consistently failed on approximately 29% and 3.75% of problems, respectively, with pass@4 = 0. This occurred despite undergoing trial-and-error fine-tuning, where correct responses are rewarded, and incorrect responses are penalized. While the reasoning model (Qwen3-8B) exhibits evident spontaneous self-reflection and significantly better performance with fewer persistent failures, it still struggles with certain problems when relying solely on numerical feedback.

A likely cause of these performance plateaus and persistent failures is the sparse informational content of numerical feedback. Scalar rewards often fail to convey *why* a response is correct or incorrect or *how* to improve multi-step reasoning. Furthermore, the limited effectiveness of spontaneous self-reflection exacerbates these challenges. Together, these limitations highlight the necessity of richer feedback mechanisms to enable more effective learning.

### C.2 PROMISE OF LEARNING FROM NATURAL LANGUAGE FEEDBACK

To move beyond the limitations of purely numerical reward signals, we explore the potential of leveraging natural language feedback to help models identify errors and refine their responses. Specifically, we examine three types of critiques: (*i*) *Indicative Critique*: A heuristic-based critique that merely indicates the binary correctness of the generated solution. (*ii*) *Indicative Critique with Ground Truth (Critique w/ GT)*: A heuristic-based critique that provides both the binary correctness indication and the ground truth answer. (*iii*) *CoT Critique*: A model-generated critique providing step-by-step evaluation to justify correctness or incorrectness, concluding with a binary correctness indication.[4] Examples of these three critique types are shown later.

We summarize the process for leveraging textual critiques to guide the refinement of LLM-generated responses in Algorithm 1. A more detailed description is provided in Appendix E, and an illustrative example of the refinement process with a CoT critique is available in Appendix T.

> **Indicative Critique**     The generated solution is incorrect.

---

[4]The CoT critique does not contain an expert demonstration.

**Indicative Critique w/ GT**    The generated solution is incorrect, the ground truth is $\frac{7}{25}$.

---

**CoT Critique**

Let's analyze the student's solution step-by-step and identify any errors:
### Step 1: Geometry Understanding The student correctly identifies the geometry of the problem, noting that the sphere is tangent to the base and sides of the cone, and that the axial section forms an isosceles triangle.
### Step 2: Variable Definitions The student defines $r$ as the radius of the sphere, $R$ as the radius of the base of the cone, and $h$ as the height of the cone. These definitions are appropriate.
### Step 3: Relating Sphere and Cone The student attempts to relate $r$, $R$, and $h$ using similar triangles. However, the equation derived:

$$r = \frac{Rh}{h + R}$$

is not directly applicable to the problem as it does not consider the specific condition that the areas are equal.
### Step 4: Equating Areas The student correctly equates the areas of the sphere and the base of the cone:
$$4\pi r^2 = \pi R^2 \implies R = 2r$$

This step is correct.
### Step 5: Substitution and Error The student substitutes $R = 2r$ into the earlier equation, leading to:

$$r = \frac{(2r)h}{h + 2r}$$

This substitution is unnecessary and results in incorrect simplification. The correct approach would involve using the inradius formula for the triangle formed by the axial section of the cone.
### Step 6: Angle Calculation The student calculates $\tan\theta = 1$ and concludes $\theta = 45°$, leading to $\cos(90°) = 0$. This is incorrect because the angle calculation should be based on the relationship between $h$ and $R$, derived from the inradius formula.
### Step 7: Precise Calculation The student attempts to verify the angle calculation but repeats the same incorrect logic, leading to the same incorrect conclusion.
### Correct Solution Approach The correct approach involves using the inradius formula for the triangle formed by the axial section of the cone: - Calculate the inradius using the area and semi-perimeter of the triangle. - Substitute $R = 2r$ and solve for $h$ in terms of $r$. - Relate $h$ and $\theta$ using trigonometric identities. - Calculate $\cos(2\theta)$ using the double angle identity.
The correct final answer is:
$$\cos(2\theta) = \frac{7}{25}.$$

Conclusion: incorrect [END]

---

**Setup.**    As described in the previous section, we evaluate the best-performing RL-finetuned Qwen2.5-7B-Base and Qwen3-8B models, generating four responses per question for a set of 4k prompts. Persistently failed question-response pairs are identified as those with pass@4 = 0. For each such pair, we prompt GPT-4o (Hurst et al., 2024) to generate a CoT critique and then construct the two corresponding heuristic-based critiques. Examples of these critique types are shown below. Subsequently, we prompt the models to refine these failed responses.

*Evaluation Metrics.* To assess the efficacy of the critique and refinement process, we define the following metrics: (*i*) **Valid Critiques Generated**: The percentage of initially erroneous responses for which the critique model ($\pi_\phi$) successfully generated a well-formed and usable critique. This accounts for potential failures in the critique generation process itself. (*ii*) **Successful Refinements**: The percentage of initially erroneous responses (that received a valid critique) for which the LLM ($\pi_\theta$) subsequently generated a correct refined response. (*iii*) **Critiqued Questions**: The percentage of unique questions (all of whose initial $k$ responses were erroneous) for which at least one of their erroneous responses received a valid critique. (*iv*) **Questions Successfully Refined**: The percentage of unique questions (all of whose initial $k$ responses were erroneous) for which at least one of their

---

**Algorithm 1** Leveraging Textual Critiques for Refinement of LLM-Generated Responses

---

**Require:** LLM $\pi_\theta$, Reasoning-based reward model $\pi_{RM}$, evaluation function Eval, set of questions $Q = \{q\}$, predefined instructions $I_c$ and $I_{\text{refine}}$, number of samples $k$

**Ensure:** Refined responses $\{y_{\text{refined}}^{(j)}\}$ for persistently failed questions

1: **for** each question $q \in Q$ **do**           # Step 1: Initial Response Sampling
2:      Sample $k$ initial responses: $\{y_0^{(i)}\}_{i=1}^k \sim \pi_\theta(\cdot \mid q)$
3: **end for**
4: **for** each question $q \in Q$ **do**       # Step 2: Response Evaluation and Critique Generation
5:      **for** each initial response $y_0^{(i)} \in \{y_0^{(i)}\}_{i=1}^k$ **do**
6:          Generate CoT critique: $c_{\text{CoT}}^{(i)} \sim \pi_{RM}(\cdot \mid I_c, q, y_0^{(i)})$
7:          Evaluate correctness: $\text{Eval}(q, y_0^{(i)}) \in \{0, 1\}$
8:          **if** $\text{Eval}(q, y_0^{(i)}) = 0$ **then**
9:             Construct heuristic-based critiques: $c_I^{(i)}$ (indicative) and $c_{\text{GT}}^{(i)}$ (with ground truth)
10:          **end if**
11:      **end for**
12: **end for**
13: Identify persistently failed questions: $Q_{\text{failed}} \leftarrow \{q \in Q \mid \forall i : \text{Eval}(q, y_0^{(i)}) = 0\}$
14: Form triplets $(q, y_0^{(j)}, c^{(j)})$ for each $q \in Q_{\text{failed}}$ and each $y_0^{(j)}$, where $c^{(j)} \in \{c_{\text{CoT}}^{(j)}, c_{\text{GT}}^{(j)}, c_I^{(j)}\}$
15: **for** each triplet $(q, y_0^{(j)}, c^{(j)})$ **do**       # Step 3: Self-Refinement Generation
16:      Generate refined response: $y_{\text{refined}}^{(j)} \sim \pi_\theta(\cdot \mid I_{\text{refine}}, q, y_0^{(j)}, c^{(j)})$
17: **end for**
     **return** Refined responses $\{y_{\text{refined}}^{(j)}\}$ for all $(q, y_0^{(j)}, c^{(j)})$

---

erroneous responses was successfully corrected through the refinement process. We evaluate the correctness of the refinements using the rule-based method described earlier.

Table 6: Analysis of performance gains from critique-based self-refinement.

| Method | % Failed Questions (Pass@4=0) | Critique Type | % Valid Critiques | % Valid Refinements | % Critiqued Questions | % Questions Refined |
|---|---|---|---|---|---|---|
| RL-finetuned Qwen2.5-7B-Base | 29.07 | Indicative Critique | 100.00 | 2.09 | 100.00 | 7.05 |
| | | Indicative Critique w/ GT | 100.00 | 1.98 | 100.00 | 6.88 |
| | | CoT Critique | 60.06 | **36.47** | 95.10 | **55.37** |
| RL-finetuned Qwen3-8B (w/ Thinking) | 3.75 | Indicative Critique | 100.00 | 3.33 | 100.00 | 8.67 |
| | | Indicative Critique w/ GT | 100.00 | 3.67 | 100.00 | 10.67 |
| | | CoT Critique | 50.17 | **10.63** | 88.67 | **20.00** |

**Results. Deliberate critique is more effective than spontaneous self-reflection for self-correction.** Incorporating all three types of critiques yields non-zero valid refinements and correctly refined questions. This suggests that critiques can enable both models to successfully correct some persistently failed responses that cannot be resolved through spontaneous self-reflection, as discussed in Section 3.1.

**CoT Critiques facilitate effective model refinement.** Table 6 shows that refinement guided by CoT critiques achieves the highest valid refinement rate (36.47% and 10.63%) and the largest percentage of successfully refined questions (55.37% and 20.00%) on Qwen2.5-7B-Base and Qwen3-8B, respectively. This performance significantly surpasses refinement based on indicative critiques or critiques with ground truth, even though the CoT critique generation process produces valid critiques for only 60.06% and 50.17% of erroneous responses. The effectiveness of CoT critiques can be attributed to their richness: by providing a step-by-step evaluation of the reasoning (*potentially along with* targeted guidance on the correct solution approach).

**Binary correctness signals alone can provide refinement benefits.** Refinement with indicative critiques with/without ground truth also yields some successful refinements, albeit at a substantially lower rate (approximately 2%-4% valid refinements). This suggests that even simply indicating the

correctness of a response can provide a minimal benefit, indicating some promise in leveraging natural language feedback to augment learning from numerical signals. However, the lack of a substantial difference between indicative critiques and critiques with ground truth suggests that providing only the ground-truth answer, without any explanation or reasoning, provides little additional guidance to the model. Additional results on Qwen3-8B-Base are shown in Appendix D.

# D  PRELIMINARY INVESTIGATION ON QWEN3-8B-BASE

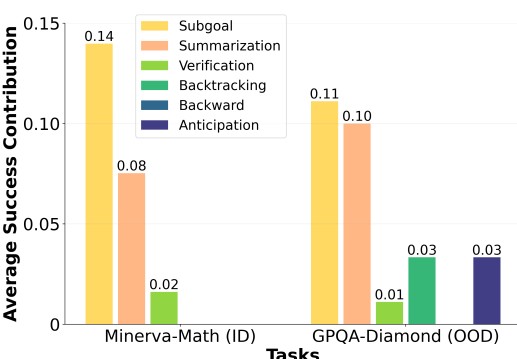

Figure 8: Average contribution of reasoning behaviors to successful completions of previously failed questions by Qwen3-8B-Base on Minerva-Math (in-distribution) and GPQA-Diamond (out-of-distribution) datasets.

We identify 50 previously unsolved problems from the Minerva-Math dataset and 15 from the GPQA-Diamond dataset for Qwen3-8B-Base. Figure 8 shows the average contribution of reasoning behaviors to successful completions of previously failed questions by Qwen3-8B-Base. Notably, self-reflection behaviors contribute minimally to successful problem-solving.

Table 7: Analysis of performance gains from critique-based self-refinement on Qwen3-8B-Base.

| Method | % Failed Questions (Pass@4=0) | Critique Type | % Valid Critiques | % Valid Refinements | % Critiqued Questions | % Questions Refined |
|---|---|---|---|---|---|---|
| RL-finetuned Qwen3-8B-Base | 17.18 | Indicative Critique | 100.00 | 3.57 | 100.00 | 11.21 |
| | | Indicative Critique w/ GT | 100.00 | 3.93 | 100.00 | 12.23 |
| | | CoT Critique | 66.08 | **44.71** | 98.25 | **66.96** |

Table 7 reveals that the best-performing RL-finetuned Qwen3-8B-Base persistently failed on 17.18% of training problems. In addition, all three types of critiques facilitate the LLM's self-refinements. These findings are consistent with the observations in Section 3.

# E   LEVERAGING TEXTUAL CRITIQUES FOR REFINING LLM RESPONSES

We describe the process for leveraging these textual critiques to guide the refinement of LLM-generated responses:

1. **Initial Response Sampling**: Given an LLM $\pi_\theta$ parameterized by $\theta$ and a set of questions $\{q\}$, we sample multiple initial responses for each question $\{y_0^{(i)}\}_{i=1}^k \sim \pi_\theta(\cdot \mid q)$, where $k$ is the number of samples.

2. **Response Evaluation and Critique Generation**: We use an evaluation function $\text{Eval}(q, y_0)$ to assess the correctness of each response $y_0$. The function outputs 1 if $y_0$ is correct and 0 otherwise. Specifically, we adopt a model-based evaluation with a reasoning-based reward model $\pi_{RM}$. The reasoning-based reward model generates a CoT critique $c_{\text{CoT}}^{(i)} \sim \pi_{RM}(\cdot \mid I_c, q, y_0^{(i)})$, where $I_c$ is a predefined instruction (detailed in Appendix S). Based on the binary correctness label within $c_{\text{CoT}}^{(i)}$, we construct the corresponding heuristic-based critiques: an indicative critique $c_{\text{I}}^{(i)}$ (containing only the correctness label) and a critique with ground truth $c_{\text{GT}}^{(i)}$ (correctness label plus the known ground truth answer for $q$).

To focus on the model's ability to learn from critiques for initially incorrect solutions and to control for spontaneous self-correction, we identify persistently failed questions. A question $q$ is classified as persistently failed if all $k$ of its initial responses $\{y_0^{(i)}\}_{i=1}^k$ are deemed incorrect based on the labels from their respective CoT critiques. For each such incorrect response $y_0^{(j)}$ from a persistently failed question, we form a triplet $(q, y_0^{(j)}, c^{(j)})$, where $c^{(j)}$ is one of the three critique types: $c_{\text{CoT}}^{(j)}$, $c_{\text{GT}}^{(j)}$, or $c_{\text{I}}^{(j)}$.

3. **Self-Refinement Generation**: For each selected triplet $(q, y_0^{(j)}, c^{(j)})$ corresponding to an initial incorrect response, we prompt the original LLM $\pi_\theta$ to generate a refined response $y_{\text{refined}}^{(j)} \sim \pi_\theta(\cdot \mid I_{\text{refine}}, q, y_0^{(j)}, c^{(j)})$. This generation is conditioned on a specific refinement instruction $I_{\text{refine}}$ (detailed in Appendix S), the original question $q$, the initial failed response $y_0^{(j)}$, and its associated critique $c^{(j)}$.

The full process is summarized in Algorithm 1. An example illustrating the self-refinement process, including the application of a CoT critique, is provided in Appendix T.

# F  THE CRITIQUE-GRPO ALGORITHM

The Critique-GRPO algorithm is summarized in Algorithm 2.

---

**Algorithm 2 Critique-GRPO: Online Policy Optimization Framework with Critiques**

---

1: **Input:** Pretrained LLM policy $\pi_{\text{old}}$ parameterized by $\theta$, reward model $\pi_{RM}$, set of questions $Q = \{q\}$, refinement instruction $I_{\text{refine}}$, critique instruction $I_c$
2: **Goal:** Improve LLM policy by learning from initial responses and their refinements
3: **Step 1: Initial Response Sampling**
4: **for** each question $q \in Q$ **do**
5:     Sample $n$ initial responses from the old policy: $\{y^{(i)}\}_{i=1}^n \sim \pi_{\text{old}}(\cdot|q)$
6:     Score the responses using the reward model $\pi_{RM}$ to obtain CoT critiques:

$$\{c_{\text{CoT}}^{(i)}\}_{i=1}^n \sim \pi_{RM}(\cdot \mid I_c, q, y^{(i)})$$

7:     Translate binary correctness labels in the critiques into scalar reward scores: $\{R^{(i)}\}_{i=1}^n$
8: **end for**
9: **Step 2: Critique-Guided Self-Refinement**
10: **for** each initial response $y^{(i)} \in \{y^{(i)}\}_{i=1}^n$ **do**
11:     Generate refined responses conditioned on $(q, y^{(i)}, c_{\text{CoT}}^{(i)})$ and instruction $I_{\text{refine}}$:

$$y_{\text{refined}}^{(i)} \sim \pi_{\text{old}}(\cdot \mid I_{\text{refine}}, q, y^{(i)}, c_{\text{CoT}}^{(i)})$$

12:     Score the refined responses using the reward model (or rule-based evaluation function)

$$\{R_{\text{refine}}^{(i)}\}_{i=1}^n$$

13: **end for**
14: Sample a subset of $k$ refinements to mitigate distributional shifts: $\{y_{\text{refined}}^{(i')}\}_{i'=1}^k \subset \{y_{\text{refined}}^{(i)}\}_{i=1}^n$
15: Combine the sampled refinements with the initial responses to form a mixed group:

$$\{y^{(i)}\}_{i=1}^n \cup \{y_{\text{refined}}^{(i')}\}_{i'=1}^k$$

16: **Step 3: Online Policy Optimization**
17: Fine-tune the model on the mixed group of responses using scalar rewards with the Critique-GRPO training objective in Equation 3.
18: Compute the advantage for each token in a response:

$$A_t^{(i)/(i')} = R^{(i)/(i')} - \text{mean}(\{R^{(i)}\}_{i=1}^n \cup \{R_{\text{refined}}^{(i')}\}_{i'=1}^k)$$

19: Calculate the probability ratios for initial and refined responses:

$$r_t^{(i)}(\theta) = \frac{\pi_\theta(y_t^{(i)}|q, y_{<t}^{(i)})}{\pi_{\text{old}}(y_t^{(i)}|q, y_{<t}^{(i)})}, \quad f(r_{\text{refined},t}^{(i')}(\theta)) = \frac{\pi_\theta(y_{\text{refined},t}^{(i')}|q, y_{\text{refined},<t}^{(i')})}{\pi_\theta(y_{\text{refined},t}^{(i')}|q, y_{\text{refined},<t}^{(i')}) + \gamma}$$

20: **Output:** Fine-tuned LLM policy $\pi_\theta$

---

# G  METHODOLOGY (DETAILED VERSION)

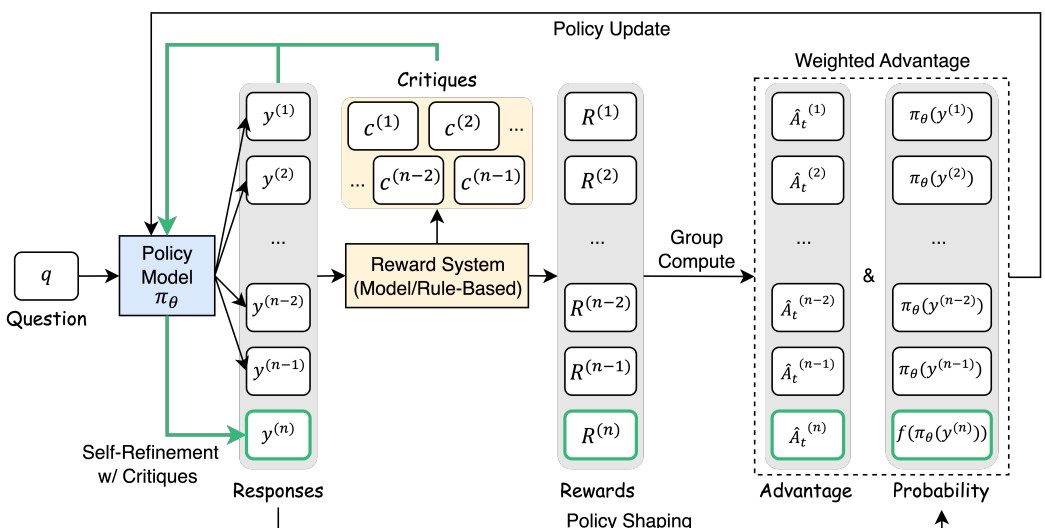

Figure 9: Overview of Critique-GRPO. Given a question, Critique-GRPO samples initial responses and then refines these responses using critiques generated by a reward system (either model-based or rule-based). These refinements are combined with the initial responses to optimize the policy within an online reinforcement learning framework. A weighted advantage function, combined with policy shaping, emphasizes correct refinements while strongly penalizing incorrect ones.

Motivated by the potential of leveraging critiques, particularly CoT critiques, for effective model refinement (Section 3), we introduce Critique-GRPO, an online optimization algorithm that learns from both natural language and numerical feedback. As illustrated in Figure 9, Critique-GRPO facilitates effective online learning and exploration by enabling the model to learn from both its generated responses and its effective refinements by incorporating natural language feedback (specifically, critiques). Before delving into the details of Critique-GRPO (Section G.1), we briefly review Group Relative Policy Optimization (GRPO) (Shao et al., 2024) (Section G.2).

## G.1  FROM GRPO TO CRITIQUE-GRPO

GRPO is an online RL algorithm widely used during the fine-tuning stage of LLMs. It builds on Proximal Policy Optimization (PPO) (Schulman et al., 2017), but eliminates the need for value function approximation by estimating advantages based on the relative performance of groups of actions. In the context of LLM policy optimization, let the model policy be parameterized by $\theta$. For each question $q$ in a given set $Q$, a group of responses $\{y^{(i)}\}_{i=1}^n$ is sampled from the old policy $\pi_{\text{old}}$. A reward model then scores these responses, yielding rewards $\{R^{(i)}\}_{i=1}^n$. The GRPO training objective is formulated as:

$$\mathcal{J}_{\text{GRPO}}(\theta) = \mathbb{E}_{q \sim Q, \{y^{(i)}\}_{i=1}^n \sim \pi_{\text{old}}(\cdot|q)}$$

$$\frac{1}{n}\sum_{i=1}^n \frac{1}{|y^{(i)}|}\sum_{t=1}^{|y^{(i)}|} \left\{ \min\left[ r_t^{(i)}(\theta)\hat{A}_t^{(i)}, \text{clip}(r_t^{(i)}(\theta), 1-\epsilon, 1+\epsilon)\hat{A}_t^{(i)} \right] - \beta D_{\text{KL}}[\pi_\theta || \pi_{\text{ref}}] \right\},$$

$$(8)$$

where $r_t^{(i)}(\theta)$ is the probability ratio, comparing the current policy $\pi_\theta$ to the old policy $\pi_{\text{old}}$ from which the responses were sampled:

$$r_t^{(i)}(\theta) = \frac{\pi_\theta(y_t^{(i)}|q, y_{<t}^{(i)})}{\pi_{\text{old}}(y_t^{(i)}|q, y_{<t}^{(i)})}, \quad \text{where } r_t^{(i)}(\theta_{\text{old}}) = 1,$$

$$(9)$$

Here, $\epsilon$ and $\beta$ are hyperparameters. The term $\epsilon$ controls the range of the clipped probability ratio, enforcing a pessimistic lower bound on policy performance to prevent excessively large policy updates. Meanwhile, $\beta$ regulates the KL divergence penalty, constraining the trained policy from deviating significantly from the reference policy.

The advantage $\hat{A}_t^{(i)}$ for all tokens in a response is calculated by normalizing the rewards $\{R^{(i)}\}_{i=1}^n$ using the group mean and standard deviation:

$$\hat{A}_t^{(i)} = \frac{R^{(i)} - \mathrm{mean}(\{R^{(1)}, \ldots, R^{(n)}\})}{\mathrm{std}(\{R^{(1)}, \ldots, R^{(n)}\})}. \tag{10}$$

Recent work (Liu et al., 2025a) suggests that the token-level normalization and the standard deviation term in the advantage calculation (highlighted in gray) may introduce biased optimization. Following their implementation, we remove these terms to obtain an unbiased optimization objective.

### G.2  Online Learning with Critique-GRPO

We introduce Critique-GRPO, an online policy optimization framework that enables a model to learn simultaneously from its generated responses and their refinements by incorporating critiques generated by a reasoning-based reward model. This approach helps the model discover new solutions and mitigate repeated failures on specific questions, as discussed in Section 3.

Specifically, Critique-GRPO operates in three main steps (Figure 9):

**Step 1: Initial Response Sampling.** Given an LLM and a set of questions $Q = \{q\}$, we sample $n$ initial responses for each question from the old policy $\pi_{\mathrm{old}}$: $\{y^{(i)}\}_{i=1}^n \sim \pi_{\mathrm{old}}(\cdot \mid q)$. These responses are evaluated using a reward system to generate both critiques $\{c^{(i)}\}_{i=1}^n$ and scalar rewards $\{R^{(i)}\}_{i=1}^n$, formulated as:

$$c^{(i)}, R^{(i)} \leftarrow \mathrm{Reward}(q, y^{(i)}), \forall i.$$

We consider two types of reward systems: *model-based* and *rule-based*. (*i*) For the model-based reward system, we use a reasoning-based reward model $\pi_{RM}$ to generate CoT critiques: $c_{\mathrm{CoT}}^{(i)} \sim \pi_{RM}(\cdot \mid I_c, q, y^{(i)})$, where $I_c$ is the critique instruction. These critiques, based on question-response pairs, are described in Appendix C. Binary correctness labels from the critiques are converted into scalar reward scores: $R^{(i)} \leftarrow c_{\mathrm{CoT}}^{(i)}$. (*ii*) For the rule-based reward system, we compare the generated responses against ground-truth answers using a string-matching function to compute scalar rewards: $R^{(i)} = \mathrm{is\_equivalent}(y^{(i)}, y_{\mathrm{GT}})$.[5] From these evaluations, two heuristic-based critiques, *i.e.,* $c_{\mathrm{I}}^{(i)}$ (indicative critique) and $c_{\mathrm{GT}}^{(i)}$ (critique with ground truth) are derived: $c_{\mathrm{I}}^{(i)}, c_{\mathrm{GT}}^{(i)} \leftarrow R^{(i)}$, as detailed in Appendix C.

**Step 2: Critique-Guided Self-Refinement.** Next, we prompt the LLM to generate refined responses conditioned on the question-response-critique triplet $(q, y^{(i)}, c^{(i)})$ and a refinement instruction $I_{\mathrm{refine}}$ (detailed in Appendix S): $y_{\mathrm{refined}}^{(i)} \sim \pi_{\mathrm{old}}(\cdot \mid I_{\mathrm{refine}}, q, y^{(i)}, c^{(i)})$, where $c^{(i)} \in \{c_{\mathrm{CoT}}^{(i)}, c_{\mathrm{GT}}^{(i)}, c_{\mathrm{I}}^{(i)}\}$. The reward model scores these self-generated refinements, producing $\{R_{\mathrm{refine}}^{(i)}\}_{i=1}^n$ (alternatively, the rule-based evaluation function discussed in Appendix C could be used). To mitigate potential distributional shifts induced by the refinements, we randomly sample a subset of $k$ refinements, denoted by $\{y_{\mathrm{refined}}^{(i')}\}_{i'=1}^k$, from the full refinement set $\{y_{\mathrm{refined}}^{(i)}\}_{i=1}^n$. This sampling prioritizes correct refinements; if no correct refinements are generated, incorrect refinements are sampled randomly. We then combine the sampled subset of refinements with the initial responses to form a mixed group of responses.[6]

---

[5]To ensure consistency, we align the results of model- and rule-based evaluations, isolating the effects of incorporating natural language feedback.

[6]Currently, only one refined response is retained. Future work may explore the optimal data ratio.

**Step 3: Online Policy Optimization.** Finally, the model is fine-tuned on a mixed set of initial and refined responses using scalar rewards. The training objective, adapted from GRPO, is given by:

$$
\mathcal{J}_{\text{Critique-GRPO}}(\theta) = \mathbb{E}_{q \sim Q, \{y^{(i)}\}_{i=1}^n \sim \pi_{\text{old}}(\cdot|q), \{y_{\text{refined}}^{(i')}\}_{i'=1}^k \sim \pi_{\text{old}}(\cdot|q)} \Bigg[
$$

$$
\underbrace{\frac{1}{n} \sum_{i=1}^n \sum_{t=1}^{|y^{(i)}|} \min\left[ r_t^{(i)}(\theta)\hat{A}_t^{(i)}, \text{clip}(r_t^{(i)}(\theta), 1-\epsilon, 1+\epsilon)\hat{A}_t^{(i)} \right]}_{\text{Objective for Initial Responses}}
$$

$$
+ \underbrace{\frac{1}{k} \sum_{i'=1}^k \sum_{t=1}^{|y_{\text{refined}}^{(i')}|} \min\left[ r_{\text{refined},t}^{(i')}(\theta)A_t^{(i')}, \text{clip}(r_{\text{refined},t}^{(i')}(\theta), 1-\epsilon, 1+\epsilon)A_t^{(i')} \right]}_{\text{Objective for Refined Responses}} \Bigg].
$$

$$(11)$$

where the advantages $A_t^{(i)}$, $A_t^{(i')}$ for all tokens in a response are defined as:

$$
A_t^{(i)} = R^{(i)} - \text{mean}\big(\{R^{(j)}\}_{j=1}^n \cup \{R_{\text{refined}}^{(j')}\}_{j'=1}^k\big),
$$
$$
A_t^{(i')} = R^{(i')} - \text{mean}\big(\{R^{(j)}\}_{j=1}^n \cup \{R_{\text{refined}}^{(j')}\}_{j'=1}^k\big),
$$

$$(12)$$

$r_t^{(i)}(\theta)$ and $f(r_{\text{refined},t}^{(i')}(\theta))$ represent the token-level probability ratios:

$$
r_t^{(i)}(\theta) = \frac{\pi_\theta(y_t^{(i)}|q, y_{<t}^{(i)})}{\pi_{\text{old}}(y_t^{(i)}|q, y_{<t}^{(i)})}, \quad f(r_{\text{refined},t}^{(i')}(\theta)) = \frac{\pi_\theta(y_{\text{refined},t}^{(i')}|q, y_{\text{refined},<t}^{(i')})}{\pi_\theta(y_{\text{refined},t}^{(i')}|q, y_{\text{refined},<t}^{(i')}) + \gamma}.
$$

$$(13)$$

We adopt a shaping function $f(x) = x/(x + \gamma)$(Yan et al., 2025) $(0 < \gamma < 1)$, depicted in Figure 10 and the lower right corner of Figure 9, to reweight gradients and emphasize low-probability tokens in refined responses. As illustrated in Figure 10, this function is bounded between $(0, 1)$, where $x$ represents the token probability of the policy. When $\gamma$ is small (*i.e.,* 0.1), the function significantly amplifies low probabilities, with this amplification decreasing as $x$ increases. Larger $\gamma$ values (0.9) produce less pronounced scaling effects. The black dashed diagonal line indicates no shaping (*i.e.,* $f(x) = x$).

We set $\gamma = 0.1$ to optimize learning from unfamiliar yet correct refinements while strongly penalizing unfamiliar incorrect ones. Additionally, we remove the KL-divergence penalty term from the original GRPO formulation to reduce constraints on policy updates, enabling more substantial model adjustments and effective learning from refinements. Future work may explore the use of the clip-high strategy (Yu et al., 2025) to reduce update constraints. We summarize Critique-GRPO in Algorithm 2 (Appendix F).

We also experiment with the ratio of initial responses to refinements per prompt, ranging from 1:1 to 7:1. Our results indicate that a 7:1 ratio achieves both stable training and optimal performance. Lower ratios lead to performance degradation due to sudden increases in entropy loss, caused by distribution shifts introduced by the refinements, during later training stages.

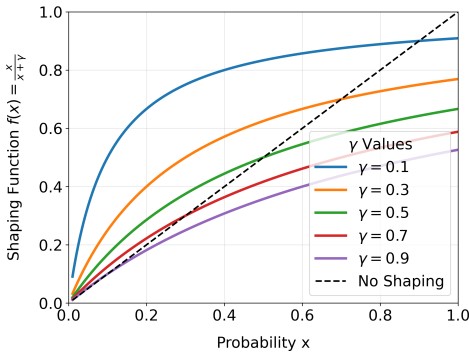

Figure 10: Effect of $\gamma$ on the shaping function $f(x)$. The black dashed diagonal line indicates no shaping, $f(x) = x$.

# H  IMPLEMENTATION DETAILS

Table 8: Default hyperparameters and training configurations used in Critique-GRPO.

| Name | Value (for Qwen2.5-7B-Base or Qwen2.5-Math-7B-Base) | Value (for Qwen3-8B) | Description |
|---|---|---|---|
| **RL Finetuning** | | | |
| num_training_prompts | 4k | 4k | Default number of training prompts (unless specified otherwise). |
| training_steps | 400 | 300 | Total number of training steps. |
| eval_freq | 20 | 25 | Frequency of evaluations (in updates). |
| batch_size | 128 | 128 | Accumulated batch size during training. |
| lr | $1e^{-6}$ | $1e^{-6}$ | Learning rate. |
| max_prompt_length | 1024 | 1024 | Maximum input context length. |
| max_response_length | 8192 | 6144 | Maximum length of generated responses. |
| n_rollouts | 7 | 7 | Number of rollouts per prompt. |
| n_refinements | 1 | 1 | Number of refinements per prompt. |
| rewards | 1 or 0 | 1 or 0 | Scalar rewards for responses. |
| kl_loss_coef | 0.0 | 0.0 | Coefficient for KL divergence loss. |
| $\gamma$ | 0.1 | 0.1 | Hyperparameter in the policy shaping function. |
| train_temp | 1.0 | 1.0 | Sampling temperature during rollouts. |
| val_temp | 0.6 | 0.6 | Sampling temperature during validation. |
| total_epochs | 30 | 30 | Total number of training epochs. |
| **Evaluation** | | | |
| eval_temp | 0.0 | 0.0 | Sampling temperature during evaluation. |
| max_tokens | 8192 | 8192 | Inference token budget during evaluation. |

**Datasets and Evaluation Metrics.** We use randomly sampled subsets of 4k examples from a reorganized 45k subset (Yan et al., 2025) of OpenR1-Math-220k (Bakouch et al., 2025) as the training set (as described in Section 3). For validation, we use the curated validation set provided by (Yan et al., 2025). We evaluate the model on five well-established mathematical reasoning benchmarks: MATH-500 (Hendrycks et al., 2021), Minerva-Math (Lewkowycz et al., 2022), OlympiadBench (He et al., 2024a), AIME 2024 (Li et al., 2024), AIME 2025 (Li et al., 2024), and AMC 2023 (Li et al., 2024). For broader analysis, we assess the model's generalization ability on three scientific and general reasoning tasks: TheoremQA (Math, Physics, EE&CS, and Finance) (Chen et al., 2023), GPQA-Diamond (Physics, Chemistry, and Biology) (Rein et al., 2024), and MMLU-Pro (Business, Computer Science, Law, *etc.*) (Wang et al., 2024). During evaluation, we use greedy decoding (temperature = 0) and report pass@1 over three runs.

**Reward Design.** During RL fine-tuning, we use model-based evaluation to generate critiques and rule-based evaluation to provide binary scalar rewards, as described in Section 3.

**Compared Methods.** We compare Critique-GRPO against the following representative approaches, categorized into supervised learning and reinforcement learning-based finetuning. All differences are considered significant at $p < 0.01$.

*Supervised Learning-based Finetuning:*

(*i*) *Supervised Finetuning (SFT)*: Finetuning the base model on high-quality annotated training data using supervised learning.

(*ii*) *Reward rAnked Finetuning (RAFT)* (Dong et al., 2023): Finetuning on self-generated correct responses, sampled based on rule-based evaluation.

(*iii*) *Refinement Finetuning (Refinement FT)* (Chen et al., 2024): Finetuning on refined correct responses generated conditionally on the question, initial response, and CoT critiques.

(*iv*) *Critique Finetuning (Critique FT)* (Wang et al., 2025): Finetuning on annotated CoT critique data to train the model to critique a given query-response.

(*v*) *Critique-in-the-Loop Finetuning (CITL-FT)* (Xi et al., 2024): Finetuning on mixed data consisting of self-generated correct responses and refined correct responses, conditioned on the question-initial response-CoT critique triplet.

***Reinforcement Learning-based Finetuning:***

(*vi*) *R1-GRPO* (DeepSeek-AI et al., 2025): Finetuning the base model on its own generations using the GRPO algorithm with binary scalar rewards.

(*vii*) *R1-Dr.GRPO* (Liu et al., 2025a): Finetuning the base model on its own generations using the Dr.GRPO algorithm, which removes terms that cause biased optimization, with binary scalar rewards.

(*viii*) *Critique-GRPO (Indicative Critique)*: Fine-tuning the base model with Critique-GRPO by utilizing indicative critiques for refinements, as described in Appendix C.

(*ix*) *Critique-GRPO (Critique with Ground Truth)*: Fine-tuning the base model with Critique-GRPO by leveraging indicative critiques alongside ground-truth answers for refinements, as detailed in Appendix C.

**Implementation Details.** We conduct experiments using Qwen2.5-7B-Base, Qwen2.5-Math-7B-Base, and Qwen3-8B, with GPT-4o (which can be replaced by other reasoning-based reward models) serving as the reward model, as described in Section 3. For supervised finetuning baselines, models are finetuned until convergence, and the best performance is reported. For RL-based approaches, models are finetuned for 400 steps, and the best performance is recorded. To ensure a fair comparison: In R1-GRPO, 8 responses (rollouts) are sampled per training prompt with a temperature of 1. In Critique-GRPO, 7 responses are sampled per prompt, along with one refined response from the refinement sets. we present detailed hyperparameters and training configurations in Table 8.

All experiments are conducted on 40 NVIDIA A800 80G GPUs. To ensure consistency, we use only critiques generated by the reward model that align with rule-based evaluations; otherwise, the reward model is prompted to regenerate the critiques. Following prior works (Liu et al., 2025a), for evaluation, we adopt greedy sampling (temperature set to 0) to generate responses and report pass@1 as the evaluation metric. When reporting pass@k, we uniformly set the temperature to 0.6 and the top-p value to 0.95.

**Compared Methods in the Investigation of Math-Centric Backbone Models.** We evaluate the efficacy of RL fine-tuning with Critique-GRPO on the math-centric backbone model, Qwen2.5-Math-7B-Base. Specifically, we compare its performance against three representative RL fine-tuning approaches based solely on numerical feedback:

(*i*) *SimpleRL-Zero* (Zeng et al., 2025): an open-source reproduction of R1-GRPO.

(*ii*) *PRIME-Zero* (Cui et al., 2025a): fine-tuning the base model using both outcome binary rewards and process binary rewards.

(*iii*) *Oat-Zero* (Liu et al., 2025a): fine-tuning the base model with Dr.GRPO using outcome binary rewards.

# I  SELF-IMPROVEMENT VIA SELF-CRITIQUING (DETAILED VERSION)

Table 9: Potential for self-improvement through RL fine-tuning using Critique-GRPO with self-generated CoT critiques (self-critiquing) on Qwen3-8B.

| Method | w/ External Supervision | Math (ID) | | | | | Science & General (OOD) | | | Avg. |
| --- | --- | --- | --- | --- | --- | --- | --- | --- | --- | --- |
| | | MATH 500 | Minerva MATH | Olympiad Bench | AMC23 | AIME24 | Theorem QA | GPQA Diamond | MMLU Pro | |
| Qwen3-8B (w/ Think) | - | 82.00 | 41.20 | 44.10 | 67.50 | 40.00 | 46.90 | 35.86 | 68.25 | 53.23 |
| + SFT | ✓ | 83.20 | 43.80 | 46.40 | 82.50 | 40.00 | 48.90 | 38.38 | 66.81 | 56.25 |
| + R1-GRPO | ✓ | 91.00 | 52.60 | 65.60 | 82.50 | 50.00 | 57.90 | 40.40 | 70.00 | 63.75 |
| + Critique-GRPO (Ours) (Self-Critique& Self-Evaluation | × | 92.00 | 52.20 | 65.50 | 87.50 | 53.30 | 59.80 | 47.47 | **70.93** | 66.09 |
| + Critique-GRPO (Ours) (Self-Critique) | ✓ | **92.60** | **52.60** | **66.20** | **95.00** | **60.00** | **60.60** | **47.98** | 70.03 | **68.13** |

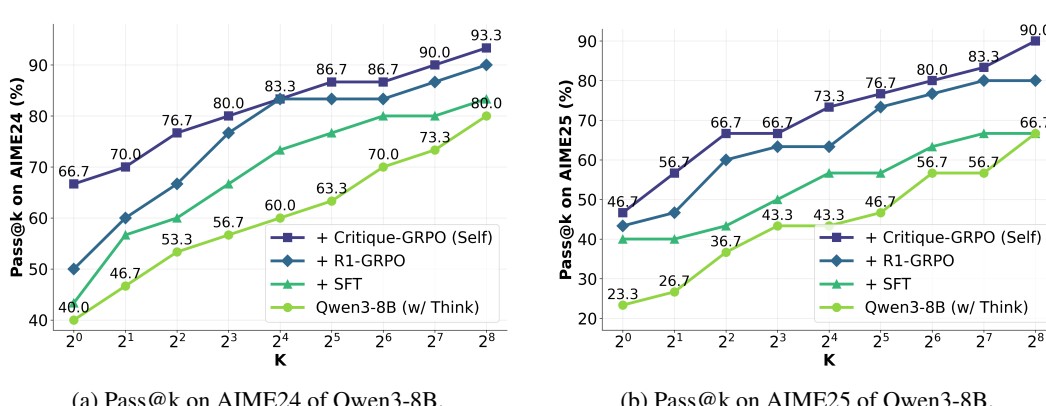

(a) Pass@k on AIME24 of Qwen3-8B.          (b) Pass@k on AIME25 of Qwen3-8B.

Figure 11: Comparison of Pass@k for self-improvement using RL fine-tuning via Critique-GRPO (self-critique), compared to methods relying on external numerical feedback (R1-GRPO), expert demonstrations (SFT), the base model Qwen3-8B.

To explore the potential of Critique-GRPO in enabling an LLM's self-improvement through self-critiquing, we prompt the model itself to serve as a reasoning-based reward model. Specifically, we investigate two types of self-critiquing: (*i*) Self-critique, where the model evaluates the correctness of its own responses using CoT critiques with ground truth answers as reference; and (*ii*) Self-critique & self-evaluation (Zhang et al., 2024), where the model evaluates its responses using CoT critiques *without* any reference. These approaches result in Critique-GRPO (self-critique) and Critique-GRPO (self-critique & self-evaluation), respectively. Details of the prompts are provided in Appendix S. Table 9 shows the evaluation results on Qwen3-8B, and Figure 11 presents pass@k performance changes on AIME24 and AIME25 (Li et al., 2024).

**Critique-GRPO enhances self-improvement through self-critiquing.** Table 9 RL fine-tuning with Critique-GRPO (self-critique) significantly outperforms fine-tuning with GRPO using external numerical feedback (R1-GRPO) and supervised fine-tuning with expert demonstrations (SFT). On average, Critique-GRPO (self-critique) improves pass@1 by +4.5% and +12.0% compared to R1-GRPO and SFT, respectively. Additionally, the unsupervised approach—Critique-GRPO (self-critique & self-evaluation)—achieves an average pass@1 improvement of 2.3% over R1-GRPO, highlighting the potential of leveraging self-critique for self-improvement *without any external supervision*.

**Self-critiquing aids effective exploration.** Figure 11 highlights the consistently superior performance of Critique-GRPO (self-critique) across pass@k metrics, with $k$ ranging from 1 to 256, indicating genuine improvements. Notably, Critique-GRPO (self-critique) achieves remarkable gains over R1-GRPO for pass@k with $k = 1$ to 4, yielding improvements of 10-16.7% on AIME24 (Figure 11a).

## J  EXPLORATION OF WEAK-TO-STRONG GENERALIZATION

Table 10: Exploration of RL fine-tuning with Critique-GRPO for weak-to-strong generalization on Qwen3-8B. Refinements, termed as "weaker refinement," are generated by a weaker model.

| Method | Math (ID) | | | | | Science & General (OOD) | | | Avg. |
|---|---|---|---|---|---|---|---|---|---|
| | MATH 500 | Minerva MATH | Olympiad Bench | AMC23 | AIME24 | Theorem QA | GPQA Diamond | MMLU Pro | |
| Qwen3-8B (w/ Think) | 82.00 | 41.20 | 44.10 | 67.50 | 40.00 | 46.90 | 35.86 | 68.25 | 53.23 |
| + R1-GRPO | **91.00** | **52.60** | **65.60** | 82.50 | 50.00 | 57.90 | 40.40 | 70.00 | 63.75 |
| + Critique-GRPO (Ours) (Weaker Refinement via Critique w/ GT) | 90.40 | 50.70 | 64.90 | **85.00** | **60.00** | **59.00** | **43.43** | **70.94** | **65.55** |

We investigate the potential of weak-to-strong generalization (Burns et al., 2023) using Critique-GRPO, where a strong model learns from refinements generated by a weaker teacher model. Specifically, we use Qwen3-8B-Base (Yang et al., 2025a) as the weaker teacher to generate refinements based on indicative critiques with the ground truth answers, guiding the improvement of Qwen3-8B.

**Critique-GRPO enables effective weak-to-strong generalization.** As shown in Table 10, Critique-GRPO (weaker refinement via critique with ground truth) achieves a +12.3% average pass@1 improvement over Qwen3-8B and outperforms R1-GRPO (65.55% *v.s.* 63.75%). This demonstrates that refinements from a weaker model can significantly enhance the performance of a stronger model.

## K    INVESTIGATION OF POLICY EXPLORATION DURING RL FINETUNING (DETAILED VERSION)

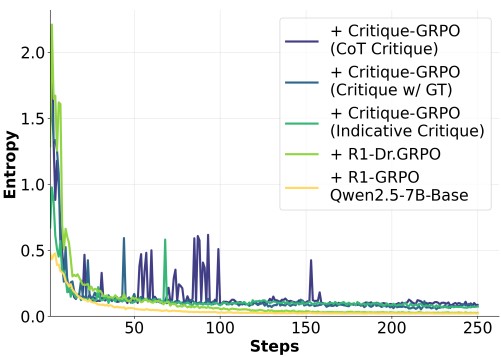
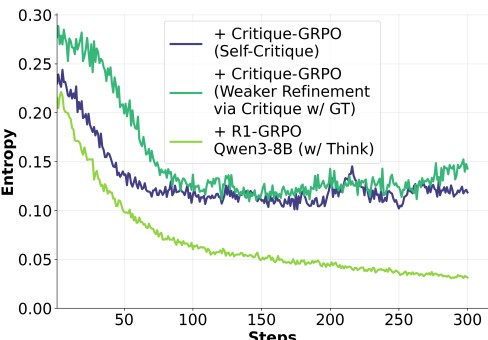

(a) Entropy dynamics for RL-based finetuning approaches over training steps on Qwen2.5-7B-Base.

(b) Entropy dynamics of self-improvement through self-critiquing and weak-to-strong generalization across training steps on Qwen3-8B.

Figure 12: Entropy dynamics during RL finetuning. A comparison of RL-based finetuning approaches for self-improvement on Qwen2.5-7B-Base (left) and self-improvement through self-critiquing with Critique-GRPO on Qwen3-8B (right).

To investigate policy exploration, we analyze two primary aspects of our RL-finetuned models: (*i*) entropy dynamics during RL fine-tuning for self-improvement using compared RL-based finetuning approaches on Qwen2.5-7B-Base and Critique-GRPO (self-critique) on Qwen3-8B (Figure 12), and (*ii*) changes in response length during fine-tuning (Figure 13).

**Learning from natural language feedback helps sustain exploration.** As shown in Figure 12a, the policy entropy of Critique-GRPO generally remains higher than that of R1-GRPO and R1-Dr.GRPO, suggesting more consistent exploration. The peaks in Critique-GRPO's entropy dynamics (before step 200) likely occur when its self-generated refinements deviate significantly from the initial sampled responses, leading to increased entropy and potentially beneficial distributional shifts. The subsequent decrease in entropy indicates that the model quickly internalizes these refinements, reducing the distributional deviation. This dynamic aligns with the observation that rare actions with high advantage can increase policy entropy (*i.e.,* unfamiliar but correct responses with high rewards promote *effective exploration*), whereas high-probability actions with high advantage tend to reduce entropy (Cui et al., 2025b). In contrast, R1-GRPO exhibits *entropy collapse*, where policy entropy drops sharply at the start of training and continues to decline monotonically to near zero. R1-Dr.GRPO initially exhibits higher entropy (before step 50) but rapidly drops to comparable near-zero values with R1-GRPO after step 150. Combined with the results in Table 2, the superior performance of Critique-GRPO over R1-Dr.GRPO and R1-GRPO highlights the importance of maintaining a certain level of entropy for better performance.

**Learning through self-critiquing facilitates policy exploration.** Figure 12b shows that Critique-GRPO (self-critique) avoids entropy collapse and maintains higher entropy than R1-GRPO. This finding aligns with the observation that increased exploration improves performance.

**Higher entropy does not always guarantee effective exploration.** Unexpectedly, as shown in Figure 12b, Critique-GRPO (weaker refinement via critique with ground truth), shown in green, achieves higher entropy than Critique-GRPO (self-critique), shown in dark blue, yet performs worse (average pass@1: 65.55% *v.s.* 68.13%). This discrepancy may be due to refinements from weaker models causing larger distributional shifts compared to self-refinements, while also being of lower quality. This suggests that the *quality* of exploration signals is more critical than the *extent* of exploration (as reflected solely by entropy).

**Critique-GRPO facilitates concise reasoning.** In Figure 13, Critique-GRPO achieves superior performance (Table 2) while minimally increasing response length on Qwen2.5-7B-Base (Figure 13a). This efficiency likely stems from its critique mechanism, which enables precise error identification and refinement, reducing the need for verbose reasoning. Additionally, Critique-GRPO tends to

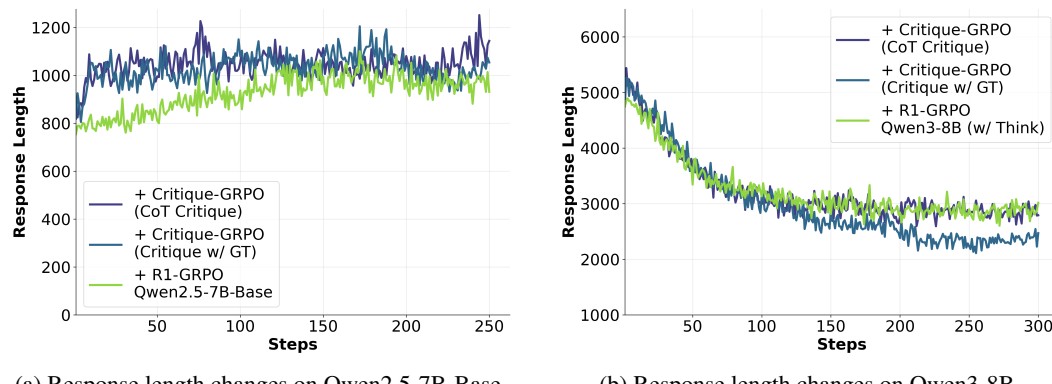

(a) Response length changes on Qwen2.5-7B-Base.

(b) Response length changes on Qwen3-8B.

Figure 13: Comparison of response length changes during RL finetuning on Qwen2.5-7B-Base (left) and Qwen3-8B (right).

reduce response length on Qwen3-8B (Figure 13b). This trend can be attributed to the correction of Qwen3-8B's tendency toward redundant and ineffective self-reflection, as discussed in Appendix P.

## L    IMPACT OF CRITIQUE LENGTH ON CRITIQUE-GRPO

Table 11: Evaluation of Critique-GRPO with varying critique length.

| Method | Critique Length | Math (ID) | | | | | Science & General (OOD) | | | Avg. |
|---|---|---|---|---|---|---|---|---|---|---|
| | | MATH 500 | Minerva MATH | Olympiad Bench | AMC23 | AIME24 | Theorem QA | GPQA Diamond | MMLU Pro | |
| Qwen2.5-7B-Base | - | 60.80 | 20.20 | 30.40 | 35.00 | 13.30 | 21.60 | 28.79 | 46.24 | 32.04 |
| + R1-GRPO | - | 74.00 | 32.00 | 38.50 | 42.50 | 16.70 | 40.60 | 33.33 | 51.81 | 41.18 |
| + Critique-GRPO (Indicative Critique) | 7 tokens | 76.00 | 36.00 | 41.00 | 55.00 | 13.30 | 41.80 | 37.88 | 55.97 | 44.62 |
| + Critique-GRPO (Critique w/ GT) | 15 tokens | 76.80 | 35.70 | 39.60 | **62.50** | 10.00 | **44.00** | **38.89** | 54.88 | 45.30 |
| + Critique-GRPO (CoT Critique) | 683 tokens | 77.80 | 36.80 | **42.40** | **62.50** | 20.00 | **44.00** | 37.88 | 55.28 | **47.08** |
| + Critique-GRPO (CoT Critique) | 2079 tokens | **77.80** | **37.90** | 38.80 | **62.50** | 16.70 | 42.80 | 35.35 | **56.12** | 46.00 |

We investigate the impact of critique token length on the performance of Critique-GRPO by varying the level of detail in generated critiques. As shown in Table 11, shorter CoT critiques (approximately 683 tokens) generally achieve strong performance. However, increasing the token length to around 2079 tokens by prompting the model to produce highly detailed critiques results in a slight performance drop (from 47.08 to 46.00). This decline is likely due to the increased difficulty of extracting critical information from longer contexts. Despite this, all results using Critique-GRPO consistently surpass those of GRPO, demonstrating the robustness of Critique-GRPO.

## M    ANALYSIS OF TRAINING COMPUTATIONAL COST

Table 12: Analysis of Training Computational Cost.

| Model | Method | Training Time (GPU-Hours) | Average Accuracy (%) |
|---|---|---|---|
| Qwen2.5-7B-Base | R1-GRPO | 240 | 41.18 |
| | Critique-GRPO (CoT-Critique) | 245 | 47.08 |
| Qwen3-8B | R1-GRPO | 296 | 63.75 |
| | Critique-GRPO (CoT-Critique) | 315 | 68.26 |

We utilize asynchronous rollouts to minimize the additional computational cost introduced by incorporating critiques and self-refinement in Critique-GRPO. Table 12 presents the training time required for 300 steps on a single NVIDIA A800 GPU.

The results show that the additional cost is limited to only 5 and 19 GPU-hours for Qwen2.5-7B-Base and Qwen3-8B, respectively, representing approximately a 2% and 6.5% increase over GRPO. This overhead is marginal compared to the significant performance improvement, with average gains of +6 and +4.5 percentage points across 8 tasks, achieved using only 4K prompts. These findings highlight the substantial gains in data efficiency and model performance, demonstrating that the additional computational cost is highly justified.

## N  FINE-GRAINED ABLATION STUDIES ON CRITIQUE-GRPO TRAINING OBJECTIVE MODIFICATIONS

Table 13: Fine-grained ablation analyses on objective modifications of Critique-GRPO.

| Method | Modifications | | | | Math (ID) | | | | | Science & General (OOD) | | | Avg. |
|---|---|---|---|---|---|---|---|---|---|---|---|---|---|
| | KL Loss Removal | Language Feedback | Refinement Sampling | Policy Shaping | MATH 500 | Minerva MATH | Olympiad Bench | AMC23 | AIME24 | Theorem QA | GPQA Diamond | MMLU Pro | |
| Qwen2.5-7B-Base | - | - | - | - | 60.80 | 20.20 | 30.40 | 35.00 | 13.30 | 21.60 | 28.79 | 46.24 | 32.04 |
| + R1-GRPO | × | × | × | × | 74.00 | 32.00 | 38.50 | 42.50 | 16.70 | 40.60 | 33.33 | 51.81 | 41.18 |
| | ✓ | × | × | × | **78.40** | 34.90 | 39.90 | 40.00 | 13.30 | 43.10 | **38.89** | 52.83 | 42.66 |
| + Critique-GRPO | ✓ | ✓ | × | × | 77.00 | 30.90 | 42.40 | 52.50 | 13.30 | 41.80 | 33.33 | 54.88 | 43.26 |
| (CoT Critique) | ✓ | ✓ | ✓ | × | 77.40 | **41.00** | 39.70 | 45.00 | 16.70 | 42.60 | 34.34 | 54.88 | 43.95 |
| | ✓ | ✓ | ✓ | ✓ | 77.80 | 36.80 | **42.40** | 62.50 | 20.00 | **44.00** | 37.88 | **55.28** | **47.08** |

We conduct fine-grained ablation analyses on the training objective of Critique-GRPO to isolate the impact of each modification. Since the training objective of Critique-GRPO is derived from GRPO, we sequentially evaluate the effects of the following modifications: (*i*) removing the KL loss; (*ii*) introducing natural language feedback (fine-tuning on initial generations and one randomly sampled refinement); (*iii*) selectively sampling a higher-quality refinement; and (*iv*) applying policy shaping based on token-level probability ratios of the generated refinement.

**Looser optimization and language feedback enable effective learning.** Table 13 shows cumulative improvements from each component. Removing KL regularization alone yields +1.5% average gain by allowing more aggressive policy updates. Natural language feedback adds +0.6% (42.66% → 43.26%), while quality-based refinement selection contributes +0.7% (43.26% → 43.95%). Policy shaping provides the largest individual gain of +3.1%, reaching 47.08% final performance. This progression reveals that RL training predominantly captures common reasoning patterns in sampled data. The substantial improvement from policy shaping indicates that amplifying rare but valuable reasoning patterns in refinements—those with low initial probability but high success rates—is crucial for maximizing performance gains during RL fine-tuning.

## O  IMPACT OF POLICY SHAPING ON RL FINETUNING

Table 14: Impact of policy shaping on the token-level probability ratios of generated refinements (as introduced in Section 4) during RL finetuning of Qwen2.5-7B-Base.

| Method | Policy Shaping | % Failed Questions (Pass@4=0) | Math (ID) | | | | | Science & General (OOD) | | | Avg. |
|---|---|---|---|---|---|---|---|---|---|---|---|
| | | | MATH 500 | Minerva MATH | Olympiad Bench | AMC23 | AIME24 | Theorem QA | GPQA Diamond | MMLU Pro | |
| Qwen2.5-7B-Base | - | - | 60.80 | 20.20 | 30.40 | 35.00 | 13.30 | 21.60 | 28.79 | 46.24 | 32.04 |
| + R1-GRPO | - | 25.85 | 74.00 | 32.00 | 38.50 | 42.50 | 16.70 | 40.60 | 33.33 | 51.81 | 41.18 |
| + Critique-GRPO (Ours) (CoT Critique) | w/o | 16.83 | 77.40 | **41.00** | 39.70 | 45.00 | 16.70 | 42.60 | 34.34 | 54.88 | 43.95 |
| + Critique-GRPO (Ours) (CoT Critique) | w/ | 32.63 | **77.80** | 36.80 | **42.40** | **62.50** | **20.00** | **44.00** | **37.88** | **55.28** | **47.08** |

To clarify the impact of policy shaping on the generated refinements during RL finetuning, we present the results of removing policy shaping during the RL finetuning of Qwen2.5-7B-Base in Table 14.

**Policy shaping enhances learning from refinements during online RL finetuning.** Critique-GRPO with policy shaping applied to the token-level probability ratios of generated refinements consistently outperforms the variant without policy shaping across nearly all tasks, improving average pass@1 scores by 3.1%.

**High training performance does not guarantee superior performance on general downstream tasks.** Compared to R1-GRPO, Critique-GRPO (w/o policy shaping) significantly reduces the percentage of persistently failed training questions by leveraging natural language feedback (25.85% → 16.83%). However, while policy shaping improves overall performance on downstream tasks, it unexpectedly leads to a higher percentage of persistently failed training questions compared to R1-GRPO (32.63% *vs.* 25.85%). We hypothesize that this occurs because policy shaping increases the probabilities of unfamiliar tokens in refinements, encouraging exploration but inadvertently degrading the ultimate performance on training problems.

## P    QUALITATIVE ANALYSIS

**Fine-Grained Analysis.** We conduct a fine-grained analysis of 100 generated responses on the Minerva-MATH dataset across four key dimensions: factuality, conciseness, correctness, and logicality, using the prompt in Appendix S. Figure 14 shows that fine-tuning with Critique-GRPO on Qwen2.5-7B-Base achieves the best performance across all four dimensions as well as in average performance. Additionally, the superior performance of RL fine-tuning with Critique-GRPO over R1-GRPO in terms of factuality and correctness indicates that CoT critiques help the model effectively identify errors and improve valid exploration. The inferior performance of SFT in logicality and conciseness may be attributed to the presence of redundant and sometimes illogical self-reflective reasoning behaviors in expert demonstrations, which could negatively impact user experience.

**Case Study.** We present a qualitative comparison between the responses generated by the base Qwen3-8B model and the RL-finetuned Qwen3-8B model using Critique-GRPO on the MATH-500 dataset. The given question is: `Evaluate` $\sin(\arcsin 0.4 + \arcsin 0.5), \sin(\arcsin 0.5 - \arcsin 0.4)$.

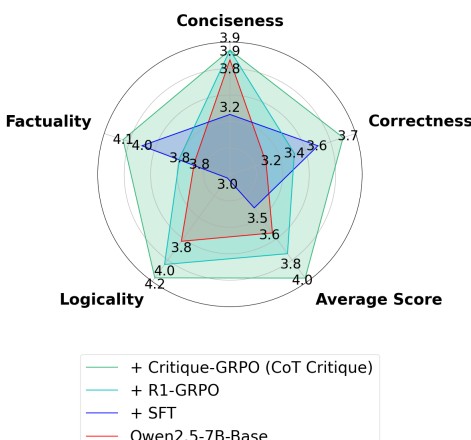

The base Qwen3-8B model produces an incorrect response due to an incorrect formulation of the expression as $\sin(A + B) \cdot {\color{red}\sin(A - B)}$, as shown in the red square. In contrast, the RL-finetuned Qwen3-8B model using Critique-GRPO generates a correct response by correctly formulating the expression as $\sin(\alpha + \beta) \cdot {\color{green}\sin(\beta - \alpha)}$, as shown in the green square. Furthermore, the base Qwen3-8B model exhibits numerous redundant and ineffective self-reflection attempts (*e.g.,* "Wait, ..." highlighted in blue), which fail to help the model identify the actual errors and answer the question correctly. This results in an excessively long response (over 6000 tokens). In contrast, the RL-finetuned Qwen3-8B model using Critique-GRPO demonstrates concise and effective reasoning. It remains on the correct path to solve the problem,

Figure 14: Qualitative analysis of generated responses on Minerva-Math across four dimensions: logicality, factuality, conciseness, correctness, and average performance.

exhibits valid self-reflection to validate the answer (*e.g.,* "Let me check with approximate values. ..."), and ultimately generates the correct answer. Detailed responses are provided in Appendix U.

**Response Generated by Base Qwen3-8B (Abridged)**

Okay, so I need to evaluate this expression: $\sin(\arcsin 0.4 + \arcsin 0.5)$ multiplied by $\sin(\arcsin 0.5 - \arcsin 0.4)$. Hmm, that looks a bit complicated with all those inverse sine functions. Let me try to break it down step by step. First, let me recall some trigonometric identities that might help here. I remember that there are formulas for $\sin(A+B)$ and $\sin(A-B)$. Maybe I can use those? Let me write them down: $\sin(A + B) = \sin A \cos B + \cos A \sin B$, $\sin(A - B) = \sin A \cos B - \cos A \sin B$

So if I let $A = \arcsin 0.4$ and $B = \arcsin 0.5$, then the expression becomes:

$$\sin(A + B) \cdot \sin(A - B)$$

Wait, so maybe I can compute $\sin(A + B)$ and $\sin(A - B)$ separately using those identities and then multiply them together? But before that, maybe I should figure out what $A$ and $B$ are? Well, $A$ is $\arcsin 0.4$, which means that $\sin A = 0.4$, and similarly $\sin B = 0.5$. But I need $\cos A$ and $\cos B$ to use the identities. Since $\sin^2 \theta + \cos^2 \theta = 1$, I can find $\cos A$ and $\cos B$. Let me compute $\cos A$ first. For angle $A$, $\sin A = 0.4$, so

$$\cos A = \sqrt{1 - \sin^2 A} = \sqrt{1 - 0.16} = \sqrt{0.84}.$$

Wait, $\sqrt{0.84}$... So $\cos B = \frac{\sqrt{3}}{2}$.

...

Okay, so now I can compute $\sin(A + B)$ and $\sin(A - B)$ using the identities.

...

So

$$\sin(A + B) = 0.2\sqrt{3} + \frac{\sqrt{21}}{10}$$

...

Okay, now moving on to $\sin(A - B)$:

$$\sin(A - B) = \sin A \cos B - \cos A \sin B$$

...

So

$$\sin(A - B) = \frac{2\sqrt{3} - \sqrt{21}}{10}$$

Therefore, the original expression is:

$$\sin(A + B) \cdot \sin(A - B) = \left(\frac{2\sqrt{3} + \sqrt{21}}{10}\right) \cdot \left(\frac{2\sqrt{3} - \sqrt{21}}{10}\right)$$

...

That seems correct. But let me verify with another approach. Maybe using another identity? Alternatively, I remember that $\sin(A + B) \cdot \sin(A - B)$ can be expressed as $\sin^2 A - \sin^2 B$. Wait, is that correct? Wait, let me recall the identity:

$$\sin(A + B) \cdot \sin(A - B) = \sin^2 A - \sin^2 B$$

Let me check that. Yes, because:

$$\sin(A + B) \cdot \sin(A - B) = [\sin A \cos B + \cos A \sin B][\sin A \cos B - \cos A \sin B]$$

$$= (\sin A \cos B)^2 - (\cos A \sin B)^2 = \sin^2 A \cos^2 B - \cos^2 A \sin^2 B$$

...

Let me compute $\sin^2 A - \sin^2 B$:

$$\sin^2 A - \sin^2 B = (0.4)^2 - (0.5)^2 = 0.16 - 0.25 = -0.09 = \frac{-9}{100}$$

...

**Response Generated by RL-Finetuned Qwen3-8B using Critique-GRPO (Abridged)**

Okay, so I need to evaluate this expression: $\sin(\arcsin 0.4 + \arcsin 0.5)$ multiplied by $\sin(\arcsin 0.5 - \arcsin 0.4)$. Hmm, let me think.

First, I remember that `arcsin` is the inverse sine function, so $\arcsin 0.4$ is an angle whose sine is $0.4$, and similarly for $\arcsin 0.5$. Let me denote some variables to make this easier. Let me set $\alpha = \arcsin 0.4$ and $\beta = \arcsin 0.5$. Then the expression becomes $\sin(\alpha + \beta) * \sin(\beta - \alpha)$.

Alright, so I need to compute $\sin(\alpha+\beta)$ and $\sin(\beta-\alpha)$ separately and then multiply them. Maybe I can use sine–addition formulas here. Let me recall that $\sin(A+B) = \sin A \cos B + \cos A \sin B$, and $\sin(A - B) = \sin A \cos B - \cos A \sin B$.

Since $\alpha$ and $\beta$ are `arcsin` values, I know that $\sin \alpha = 0.4$ and $\sin \beta = 0.5$. Therefore, I need to find $\cos \alpha$ and $\cos \beta$ to use in the formulas.

For $\cos \alpha$, since $\sin^2 \alpha + \cos^2 \alpha = 1$, we have

$$\cos \alpha = \sqrt{1 - \sin^2 \alpha} = \sqrt{1 - 0.16} = \sqrt{0.84}.$$

Note that $0.84 = \frac{84}{100} = \frac{21}{25}$, so $\cos \alpha = \sqrt{\frac{21}{25}} = \frac{\sqrt{21}}{5}$.

Similarly,

$$\cos \beta = \sqrt{1 - \sin^2 \beta} = \sqrt{1 - 0.25} = \sqrt{0.75} = \sqrt{\frac{3}{4}} = \frac{\sqrt{3}}{2}.$$

Now, let me compute $\sin(\alpha + \beta)$. Using the formula,

$$\sin(\alpha + \beta) = \sin \alpha \cos \beta + \cos \alpha \sin \beta = 0.4\left(\frac{\sqrt{3}}{2}\right) + \left(\frac{\sqrt{21}}{5}\right)(0.5).$$

Compute each term:

$$0.4 \cdot \frac{\sqrt{3}}{2} = \frac{2}{5} \cdot \frac{\sqrt{3}}{2} = \frac{\sqrt{3}}{5}, \quad \frac{\sqrt{21}}{5} \cdot 0.5 = \frac{\sqrt{21}}{10}.$$

Therefore

$$\sin(\alpha + \beta) = \frac{\sqrt{3}}{5} + \frac{\sqrt{21}}{10} = \frac{2\sqrt{3} + \sqrt{21}}{10}.$$

Similarly, compute $\sin(\beta - \alpha)$:

$$\sin(\beta - \alpha) = \sin \beta \cos \alpha - \cos \beta \sin \alpha = 0.5\left(\frac{\sqrt{21}}{5}\right) - \frac{\sqrt{3}}{2}(0.4).$$

Thus

$$\sin(\beta - \alpha) = \frac{\sqrt{21}}{10} - \frac{2\sqrt{3}}{10} = \frac{\sqrt{21} - 2\sqrt{3}}{10}.$$

Therefore, the original expression is

$$\left(\frac{2\sqrt{3} + \sqrt{21}}{10}\right)\left(\frac{\sqrt{21} - 2\sqrt{3}}{10}\right).$$

Let $A = 2\sqrt{3} + \sqrt{21}$ and $B = \sqrt{21} - 2\sqrt{3}$; then the product is $\frac{AB}{100}$. Note that $AB = (\sqrt{21})^2 - (2\sqrt{3})^2 = 21 - 12 = 9$, so the value is $\frac{9}{100}$.

...

## Q    LIMITATIONS

While Critique-GRPO establishes a promising foundation for leveraging both natural language and numerical feedback, notable limitations remain.

**Performance limitations due to failed refinements.** Policy models sometimes fail to follow CoT critiques to refine their responses. We attribute this to the lack of deliberate training for self-refinement. An example of a failed refinement is provided in Appendix V. Future work could focus on improving the model's refinement capabilities or training a specialized model dedicated to refinement tasks.

**The role of critique detail in refinement quality.** We currently utilize three types of critiques (see Section 3), with CoT critiques demonstrating the greatest benefits for refinement. This advantage likely stems from their detailed step-by-step evaluations and concise improvement suggestions, which help models identify and correct errors in initial responses. It follows that more detailed critiques could result in higher-quality refinements. For simplicity, we use GPT-4o as the reasoning-based reward model, *not for expert knowledge distillation*. Consequently, the generated CoT critiques do not include expert demonstrations. Future work may explore alternative reasoning-based reward models. One might assume that directly incorporating expert demonstrations into critiques would significantly improve performance. However, our experiments reveal otherwise. Upon analyzing the generated refinements, we observe that both pre-trained models (*e.g.,* Qwen2.5-7B-Base) and alignment-tuned models (*e.g.,* Qwen3-8B) tend to produce conclusive sentences and correct answers as refinements, rather than detailed step-by-step reasoning to derive the correct answer. This behavior limits the effectiveness of expert demonstrations as critiques.

Future work could investigate, in greater depth, which types of critiques provide the most significant benefits for refinement, particularly in reasoning-intensive tasks.

# R ANALYSIS OF COGNITIVE BEHAVIORS

To systematically investigate this question, we characterize six key cognitive behaviors that contribute to self-improving reasoning during RL fine-tuning, as follows:

- **Subgoal Setting**: Breaking down complex problems into smaller, manageable steps or subtasks. For example, "Step 1... Step 2..."

- **Summarization**: Summarizing the current state by identifying completed subtasks and determining what remains to be done. This helps guide the next steps in reasoning. For example, "Now we have obtained..., next, we need to..."

- **Verification**: Systematically checking intermediate results or computations to ensure correctness. For example, "Let's verify this result by..."

- **Backtracking**: Identifying errors or dead-ends in reasoning and explicitly revising previous methods or approaches. For example, "This approach won't work because..., let's try another method..."

- **Backward Chaining**: Reasoning from desired outcomes back to initial inputs or steps required to achieve the result. This is particularly applicable to multiple-choice questions where answer options are provided. For example, "To get 24, I could do $24 \div 2 = 12$..." (Gandhi et al., 2025)

- **Anticipation**: Anticipating potential inaccuracies or exhaustively considering multiple possibilities to solve a problem. For example, "Alternatively, this problem can be solved by..."

We analyze the reasoning (cognitive) behaviors using the prompts shown below.

When assessing the contributions of reasoning behaviors in Section 3 to successful problem-solving in RL fine-tuned models, we count each behavior appearing in the generated responses *only once*. For example, if the model produces multiple subgoals in a single response, the occurrence of "subgoal setting" is counted as one.

**Prompts for Analyzing Reasoning Behaviors**

**System:** You are a helpful assistant.
**User:** The following is a chain-of-thought produced by a language model in response to a math & science problem:
**Question:** <Question Content>
**Reasoning:** <Model Reasoning>
**Ground Truth:** <Ground Truth Content>
**Task 1: Answer Verification**
Determine whether the reasoning includes any *explicit or implicit answer verification steps* — moments where the model checks intermediate computations or final results for correctness. Example: "Let's verify this result by..."
- Report the number of distinct answer verification steps using: `<count>n</count>`. If none are found, return `<count>0</count>`. - If such behavior is present and the final answer matches the ground truth, indicate whether the behavior contributed to the correct answer using the format: `contribution: yes/no`.
**Task 2: Backtracking Behavior**
Determine whether the reasoning demonstrates *backtracking* — where the model identifies an error or dead end and switches to a different approach. Example: "This approach won't work because..., let's try another method..."
- Report the number of distinct backtracking instances using: `<count>n</count>`. If none are found, return `<count>0</count>`. - If such behavior is present and the final answer matches the ground truth, indicate whether the behavior contributed to the correct answer using the format: `contribution: yes/no`.
**Task 3: Subgoal Setting**
Determine whether the reasoning includes any *explicit subgoals* — intermediate steps that break the problem into smaller, manageable parts. Example: "First, I'll try to..., then I'll..."
- Report the number of clearly defined subgoals using: `<count>n</count>`. If none are found, return `<count>0</count>`. - If such behavior is present and the final answer matches the ground truth, indicate whether the behavior contributed to the correct answer using the format: `contribution: yes/no`.
**Task 4: Backward Chaining**
Determine whether the reasoning includes *backward chaining* — starting from the target result and reasoning backward to infer inputs or steps. Example: "To get 24, I could do 24 ÷ 2 = 12..."
- Report the number of distinct backward chaining attempts using: `<count>n</count>`. If none are found, return `<count>0</count>`. - If such behavior is present and the final answer matches the ground truth, indicate whether the behavior contributed to the correct answer using the format: `contribution: yes/no`.
**Task 5: Anticipation**
Determine whether the reasoning includes *enumeration* or *anticipation and re-proposal* — suggesting alternative approaches or revising prior methods. Examples: "Alternatively, this problem can be solved by...", "Let's try a different approach..."
- Report the number of such instances using: `<count>n</count>`. If none are found, return `<count>0</count>`. - If such behavior is present and the final answer matches the ground truth, indicate whether the behavior contributed to the correct answer using the format: `contribution: yes/no`.
**Task 6: Summarization**
Determine whether the reasoning includes *summarization* — identifying completed subtasks, summarizing progress, and determining the next steps. Example: "Now we have obtained..., next, we need to..."
- Report the number of summarization instances using: `<count>n</count>`. If none are found, return `<count>0</count>`. - If such behavior is present and the final answer matches the ground truth, indicate whether the behavior contributed to the correct answer using the format: `contribution: yes/no`.

## S    PROMPTS

**Training Prompt.**    The following training prompt is used during all RL fine-tuning experiments:

---

**Training Prompt**

**System:** You are a helpful assistant.
**User:** `<Question Content>`
Please reason step by step and place your final answer within `\boxed`.

---

**Prompt for Generating Chain-of-Thought Critique.**    We adopt a prompt inspired by (Wang et al., 2025) to enable GPT-4o (Hurst et al., 2024) to generate CoT critiques. For quality control, we retained only those model-generated critiques whose evaluative conclusions (correct/incorrect) aligned with rule-based verification. When inconsistencies occurred, we prompted the critique model to regenerate the critiques.

---

**Prompt for Generating Chain-of-Thought Critique**

**System:** You are a science expert. A student is trying to solve a question. Please explain briefly (step-by-step) whether the student's solution is correct or not. Finally, conclude your judgment with:
"Conclusion: `correct/incorrect` [END]."
**User:**
Question: `<Question Content>`
Ground Truth Answer: `<Ground Truth>`
Student's Solution: `<Generated Solution>`
Critique:

---

**Prompt for Generating Chain-of-Thought Critique with Internal Knowledge**    The following prompt is designed to enable an LLM to leverage its internal knowledge and evaluate the correctness of its own generated responses through step-by-step CoT critiques.

---

**Prompt for Generating Chain-of-Thought Critique with Internal Knowledge**

**System:** You are a science expert. A student is trying to solve a question. Please explain briefly (step-by-step) whether the student's solution is correct or not. Finally, conclude your judgment with:
"Conclusion: `correct/incorrect` [END]."
**User:**
Question: `<Question Content>`
Student's Solution: `<Generated Solution>`
Critique:

---

**Refinement Prompt.**    The following refinement prompt is used to guide the model in improving its response by incorporating the critique.

> **Refinement Prompt**
>
> **System:** You are a helpful assistant.
> **User:** Given the following inputs:
> Question: `<Question Content>`
> Previous Solution: `<Generated Solution>`
> Critique: `<Critique Content>`
> Please re-answer by:
> - Correcting potential errors identified in the critique, if they exist.
> - Providing clear, step-by-step reasoning.
> - Placing your final answer within `\boxed`.
> Ensure the revised solution addresses all issues raised in the critique.

Future work could explore designing prompts (Zhang et al., 2023) to enable LLMs to generate high-quality CoT critiques.

**Prompt for Qualitative Analysis.**   We employ the following prompt to conduct qualitative analysis of the generated responses using GPT-4o.

> **Prompt for Qualitative Analysis**
>
> **System:** You are a science expert. You are provided with a question, the correct ground truth answer, and a student's solution. Please conduct a fine-grained qualitative analysis of the student's solution based on the following four aspects, rated on a scale of 1-5:
> 1. Correctness of the Final Answer: Is the final numerical answer correct based on your calculations or the ground truth? If incorrect, provide the correct answer and explain the discrepancy.
> 2. Verbosity: Is the reasoning path too verbose, too concise, or appropriate? Identify areas for condensation or expansion to improve clarity.
> 3. Factual Accuracy: Are all formulas, conversions, and physical principles factually accurate? Highlight any errors or misleading statements.
> 4. Logical Coherence: Does the reasoning flow logically from one step to the next? Identify gaps in logic, missing steps, or irrelevant details that detract from the solution.
> End your analysis with:
> "Conclusion:" Provide ratings (1-5 scale) for each aspect.
> **User:**
> Question: `<Question Content>`
> Ground Truth Answer: `<Ground Truth>`
> Student's Answer: `<Answer>`
> Conclusion:

# T  AN EXAMPLE OF SUCCESSFUL REFINEMENT USING A COT CRITIQUE

The example below demonstrates a successful refinement using a CoT critique. This output was generated by the best-performing RL-finetuned Qwen3-8B model using GRPO algorithm in Section 3, utilizing the refinement prompt detailed in Appendix T.

---

**Question**

2. A sphere is inscribed in a cone, such that the area of the sphere and the area of the base of the cone are equal. Calculate the cosine of the angle of the axial section of the cone at the vertex which is also the apex of the cone.

---

**CoT Critique**

Let's analyze the student's solution step-by-step and identify any errors:

### Step 1: Geometry Understanding The student correctly identifies the geometry of the problem, noting that the sphere is tangent to the base and sides of the cone, and that the axial section forms an isosceles triangle.

### Step 2: Variable Definitions The student defines $r$ as the radius of the sphere, $R$ as the radius of the base of the cone, and $h$ as the height of the cone. These definitions are appropriate.

### Step 3: Relating Sphere and Cone The student attempts to relate $r$, $R$, and $h$ using similar triangles. However, the equation derived:

$$r = \frac{Rh}{h + R}$$

is not directly applicable to the problem as it does not consider the specific condition that the areas are equal.

### Step 4: Equating Areas The student correctly equates the areas of the sphere and the base of the cone:

$$4\pi r^2 = \pi R^2 \implies R = 2r$$

This step is correct.

### Step 5: Substitution and Error The student substitutes $R = 2r$ into the earlier equation, leading to:

$$r = \frac{(2r)h}{h + 2r}$$

This substitution is unnecessary and results in incorrect simplification. The correct approach would involve using the inradius formula for the triangle formed by the axial section of the cone.

### Step 6: Angle Calculation The student calculates $\tan\theta = 1$ and concludes $\theta = 45°$, leading to $\cos(90°) = 0$. This is incorrect because the angle calculation should be based on the relationship between $h$ and $R$, derived from the inradius formula.

### Step 7: Precise Calculation The student attempts to verify the angle calculation but repeats the same incorrect logic, leading to the same incorrect conclusion.

### Correct Solution Approach The correct approach involves using the inradius formula for the triangle formed by the axial section of the cone: - Calculate the inradius using the area and semi-perimeter of the triangle. - Substitute $R = 2r$ and solve for $h$ in terms of $r$. - Relate $h$ and $\theta$ using trigonometric identities. - Calculate $\cos(2\theta)$ using the double angle identity.

The correct final answer is:

$$\cos(2\theta) = \frac{7}{25}.$$

Conclusion: incorrect [END]

**Incorrect Initial Response**

To solve the problem, let's break it down step by step.

### Step 1: Geometry Understanding - **Sphere Inscribed in a Cone**: The sphere is tangent to the base and the sides of the cone. - **Axial Section**: The cross-section along the cone's central axis forms an isosceles triangle. - **Vertex Angle**: Let the vertex angle of the cone be $2\theta$, so each base angle is $\theta$.

### Step 2: Variable Definitions - $r$: Radius of the sphere. - $R$: Radius of the base of the cone. - $h$: Height of the cone.

### Step 3: Relating the Sphere and the Cone The sphere is inscribed in the cone, so: - The vertical distance from the center of the sphere to the base is $h - r$. - The horizontal distance from the center of the sphere to the cone's side is $r$.

Using similar triangles, we relate $r$, $R$, and $h$:

$$\frac{r}{R} = \frac{h - r}{h} \implies rh = R(h - r) \implies r(h + R) = Rh \implies r = \frac{Rh}{h + R}.$$

### Step 4: Equating Areas The area of the sphere is $4\pi r^2$, and the area of the base of the cone is $\pi R^2$. From the problem:

$$4\pi r^2 = \pi R^2 \implies 4r^2 = R^2 \implies R = 2r.$$

### Step 5: Substituting $R = 2r$ Substituting $R = 2r$ into the earlier equation:

$$r = \frac{(2r)h}{h + 2r} \implies r(h + 2r) = 2rh \implies h + 2r = 2h \implies h = 2r.$$

### Step 6: Finding the Angle $\theta$ In the axial section of the cone, we have a right triangle with: - Height $h = 2r$. - Base radius $R = 2r$.

The tangent of $\theta$ is:

$$\tan\theta = \frac{R}{h} = \frac{2r}{2r} = 1 \implies \theta = 45°.$$

The cosine of the vertex angle $2\theta$ is:

$$\cos(2\theta) = \cos(90°) = 0.$$

### Step 7: Verifying with Precise Calculations For the axial section, the cone forms an isosceles triangle with: - Two equal slant heights $l$. - Base $2R = 4r$. - Height $h = 2r$.

Using the Pythagorean theorem, the slant height $l$ is:

$$l = \sqrt{h^2 + R^2} = \sqrt{(2r)^2 + (2r)^2} = \sqrt{8r^2} = 2r\sqrt{2}.$$

The tangent of the base angle $\theta$ is:

$$\tan\theta = \frac{R}{h} = \frac{2r}{2r} = 1 \implies \theta = 45°.$$

Thus, the vertex angle is $2\theta = 90°$, and:

$$\cos(90°) = 0.$$

### Final Answer

$$\boxed{0}$$

**Correct Refined Response (1/2)**

Let's revise the solution by addressing the issues identified in the critique and providing a clear, step-by-step reasoning.

### Step 1: Geometry Understanding The problem involves a cone with a sphere inscribed in it, such that the sphere is tangent to both the base and the sides of the cone. Additionally, the area of the sphere equals the area of the base of the cone.

### Step 2: Variable Definitions - $r$: Radius of the sphere. - $R$: Radius of the base of the cone. - $h$: Height of the cone.

### Step 3: Relating the Sphere and the Cone The sphere is inscribed in the cone, so: - The vertical distance from the center of the sphere to the base is $h - r$. - The horizontal distance from the center of the sphere to the cone's side is $r$.

Using similar triangles, we relate $r$, $R$, and $h$:

$$\frac{r}{R} = \frac{h-r}{h} \implies r(h+R) = Rh \implies r = \frac{Rh}{h+R}.$$

### Step 4: Equating Areas The area of the sphere is $4\pi r^2$, and the area of the base of the cone is $\pi R^2$. From the problem:

$$4\pi r^2 = \pi R^2 \implies 4r^2 = R^2 \implies R = 2r.$$

### Step 5: Substituting $R = 2r$ Substitute $R = 2r$ into the earlier equation:

$$r = \frac{(2r)h}{h+2r} \implies r(h+2r) = 2rh \implies h + 2r = 2h \implies h = 2r.$$

### Step 6: Finding the Angle $\theta$ In the axial section of the cone: - The height is $h = 2r$. - The base radius is $R = 2r$.

The tangent of $\theta$ is:

$$\tan\theta = \frac{R}{h} = \frac{2r}{2r} = 1 \implies \theta = 45°.$$

The cosine of the vertex angle $2\theta$ is:

$$\cos(2\theta) = \cos(90°) = 0.$$

However, this leads to an incorrect conclusion. Let's apply the **inradius formula** for a more accurate approach.

### Correct Approach: Using the Inradius Formula The axial section of the cone is an isosceles triangle with: - Height $h$. - Base $2R$. - Slant height $l = \sqrt{h^2 + R^2}$.

The inradius $r$ of this triangle is given by:

$$r = \frac{A}{s},$$

where $A$ is the area, and $s$ is the semi-perimeter.

### Area and Semi-Perimeter: The area $A$ of the triangle is:

$$A = \frac{1}{2} \times 2R \times h = Rh.$$

**Correct Refined Response (2/2)**

The semi-perimeter $s$ is:

$$s = \frac{2l + 2R}{2} = l + R = \sqrt{h^2 + R^2} + R.$$

### Inradius: Substitute $A$ and $s$:

$$r = \frac{Rh}{\sqrt{h^2 + R^2} + R}.$$

Given $R = 2r$ and $h = 2r$, substitute into the equation:

$$r = \frac{(2r)(2r)}{\sqrt{(2r)^2 + (2r)^2} + 2r} = \frac{4r^2}{\sqrt{8r^2} + 2r} = \frac{4r^2}{2r\sqrt{2} + 2r}.$$

Simplify:

$$r = \frac{4r}{2(\sqrt{2} + 1)} = \frac{2r}{\sqrt{2} + 1}.$$

### Rationalizing the Denominator:

$$r = \frac{2r}{\sqrt{2} + 1} \cdot \frac{\sqrt{2} - 1}{\sqrt{2} - 1} = \frac{2r(\sqrt{2} - 1)}{2 - 1} = 2r(\sqrt{2} - 1).$$

### Step 7: Find the Final Angle Given $h = 2r$ and $R = 2r$, the slant height $l$ is:

$$l = \sqrt{h^2 + R^2} = \sqrt{(2r)^2 + (2r)^2} = \sqrt{8r^2} = 2r\sqrt{2}.$$

The tangent of $\theta$ is:

$$\tan\theta = \frac{R}{h} = \frac{2r}{2r} = 1 \implies \theta = 45°.$$

The cosine of the vertex angle $2\theta$ is:

$$\cos(2\theta) = \frac{1 - \tan^2\theta}{1 + \tan^2\theta} = \frac{1 - \left(\frac{3}{4}\right)^2}{1 + \left(\frac{3}{4}\right)^2} = \frac{\frac{7}{16}}{\frac{25}{16}} = \frac{7}{25}.$$

### Final Answer:

$$\boxed{\frac{7}{25}}$$

## U  RESPONSES UTILIZED IN QUALITATIVE ANALYSIS

> **Question**
> Evaluate
> $$\sin(\arcsin 0.4 + \arcsin 0.5) \cdot \sin(\arcsin 0.5 - \arcsin 0.4).$$

> **Response Generated by Qwen3-8B (1/4)**
> Okay, so I need to evaluate this expression: $\sin(\arcsin 0.4 + \arcsin 0.5)$ multiplied by $\sin(\arcsin 0.5 - \arcsin 0.4)$. Hmm, that looks a bit complicated with all those inverse sine functions. Let me try to break it down step by step. First, let me recall some trigonometric identities that might help here. I remember that there are formulas for $\sin(A+B)$ and $\sin(A-B)$. Maybe I can use those? Let me write them down:
> $$\sin(A + B) = \sin A \cos B + \cos A \sin B$$
> $$\sin(A - B) = \sin A \cos B - \cos A \sin B$$
> So if I let $A = \arcsin 0.4$ and $B = \arcsin 0.5$, then the expression becomes:
> $$\sin(A + B) \cdot \sin(A - B)$$
> Wait, so maybe I can compute $\sin(A + B)$ and $\sin(A - B)$ separately using those identities and then multiply them together? But before that, maybe I should figure out what $A$ and $B$ are? Well, $A$ is $\arcsin 0.4$, which means that $\sin A = 0.4$, and similarly $\sin B = 0.5$. But I need $\cos A$ and $\cos B$ to use the identities. Since $\sin^2 \theta + \cos^2 \theta = 1$, I can find $\cos A$ and $\cos B$. Let me compute $\cos A$ first. For angle $A$, $\sin A = 0.4$, so
> $$\cos A = \sqrt{1 - \sin^2 A} = \sqrt{1 - 0.16} = \sqrt{0.84}.$$
> Wait, $\sqrt{0.84}$... Let me calculate that. $0.84$ is $\frac{84}{100}$, so $\sqrt{0.84} = \frac{\sqrt{84}}{10}$. Simplify $\sqrt{84}$... $\sqrt{84} = \sqrt{4 \cdot 21} = 2 \cdot \sqrt{21}$. So
> $$\sqrt{0.84} = \frac{\sqrt{84}}{10} = \frac{2 \cdot \sqrt{21}}{10} = \frac{\sqrt{21}}{5}.$$
> Wait, let me check that again. $\sqrt{84}$ is $\sqrt{4 \cdot 21}$ which is $2 \cdot \sqrt{21}$, so yes, $\sqrt{84}/10 = 2 \cdot \sqrt{21}/10 = \sqrt{21}/5$. So $\cos A = \sqrt{21}/5$. Wait, but $\sqrt{21}$ is approximately $4.583$, so $\sqrt{21}/5$ is about $0.9166$. Let me check if that's correct. Since $\sin A$ is $0.4$, then $\cos A$ should be $\sqrt{1 - 0.16} = \sqrt{0.84} \approx 0.9165$, which matches. So that's correct. Similarly, for angle $B$, $\sin B = 0.5$, so $\cos B = \sqrt{1 - 0.25} = \sqrt{0.75} = \frac{\sqrt{3}}{2} \approx 0.8660$. That's right because $\sin\left(\frac{\pi}{6}\right) = 0.5$, so $\cos\left(\frac{\pi}{6}\right) = \frac{\sqrt{3}}{2}$. So $\cos B = \frac{\sqrt{3}}{2}$. Okay, so now I can compute $\sin(A + B)$ and $\sin(A - B)$ using the identities. Starting with $\sin(A + B)$:
> $$\sin(A + B) = \sin A \cos B + \cos A \sin B$$
> Plugging in the values:
> $$\sin A = 0.4, \quad \cos B = \frac{\sqrt{3}}{2}, \quad \cos A = \frac{\sqrt{21}}{5}, \quad \sin B = 0.5$$
> So:
> $$\sin(A + B) = 0.4 \left(\frac{\sqrt{3}}{2}\right) + \left(\frac{\sqrt{21}}{5}\right)(0.5)$$

**Response Generated by Qwen3-8B (2/4)**

Let me compute each term: First term:

$$0.4\left(\frac{\sqrt{3}}{2}\right) = \left(\frac{0.4}{2}\right)\sqrt{3} = 0.2\sqrt{3}$$

Second term:

$$\left(\frac{\sqrt{21}}{5}\right)(0.5) = \left(\frac{\sqrt{21}}{5}\right)\left(\frac{1}{2}\right) = \frac{\sqrt{21}}{10}$$

So

$$\sin(A+B) = 0.2\sqrt{3} + \frac{\sqrt{21}}{10}$$

Hmm, maybe I can write 0.2 as $\frac{1}{5}$? Let me check: 0.2 is $\frac{1}{5}$, so

$$\frac{1}{5}\sqrt{3} + \frac{\sqrt{21}}{10}.$$

To combine these terms, maybe get a common denominator? Yes, common denominator is 10:

$$\frac{2\sqrt{3}}{10} + \frac{\sqrt{21}}{10} = \frac{2\sqrt{3} + \sqrt{21}}{10}$$

So

$$\sin(A+B) = \frac{2\sqrt{3} + \sqrt{21}}{10}$$

Okay, now moving on to $\sin(A-B)$:

$$\sin(A-B) = \sin A \cos B - \cos A \sin B$$

Which is similar to the previous one, but with a minus sign. So plugging in the same values:

$$\sin A \cos B - \cos A \sin B = 0.4\left(\frac{\sqrt{3}}{2}\right) - \left(\frac{\sqrt{21}}{5}\right)(0.5)$$

Which is the same as: First term:

$$0.2\sqrt{3} - \frac{\sqrt{21}}{10}$$

Again, converting 0.2 to $\frac{1}{5}$:

$$\frac{1}{5}\sqrt{3} - \frac{\sqrt{21}}{10} = \frac{2\sqrt{3} - \sqrt{21}}{10}$$

So

$$\sin(A-B) = \frac{2\sqrt{3} - \sqrt{21}}{10}$$

Therefore, the original expression is:

$$\sin(A+B) \cdot \sin(A-B) = \left(\frac{2\sqrt{3} + \sqrt{21}}{10}\right) \cdot \left(\frac{2\sqrt{3} - \sqrt{21}}{10}\right)$$

**Response Generated by Qwen3-8B (3/4)**

Hmm, this looks like a difference of squares. Because $(a + b)(a - b) = a^2 - b^2$. Let me check: Let me denote $a = 2\sqrt{3}$ and $b = \sqrt{21}$. Then the product is $(a+b)(a-b) = a^2 - b^2$. So compute $a^2 - b^2$:

$$a^2 = (2\sqrt{3})^2 = 4 \cdot 3 = 12$$

$$b^2 = (\sqrt{21})^2 = 21$$

Therefore, $a^2 - b^2 = 12 - 21 = -9$

$$\text{Therefore, the product is } \frac{-9}{10 \cdot 10} = \frac{-9}{100}$$

Wait, so the entire expression simplifies to $\frac{-9}{100}$? But let me check my steps again to make sure I didn't make a mistake. First, $\sin(A + B)$ and $\sin(A - B)$ were calculated as $\frac{2\sqrt{3}+\sqrt{21}}{10}$ and $\frac{2\sqrt{3}-\sqrt{21}}{10}$. Multiplying them gives:

$$\frac{(2\sqrt{3})^2 - (\sqrt{21})^2}{100} = \frac{12 - 21}{100} = \frac{-9}{100}$$

That seems correct. But let me verify with another approach. Maybe using another identity? Alternatively, I remember that $\sin(A + B) \cdot \sin(A - B)$ can be expressed as $\sin^2 A - \sin^2 B$. Wait, is that correct? Wait, let me recall the identity:

$$\sin(A + B) \cdot \sin(A - B) = \sin^2 A - \sin^2 B$$

Let me check that. Yes, because:

$$\sin(A + B) \cdot \sin(A - B) = [\sin A \cos B + \cos A \sin B][\sin A \cos B - \cos A \sin B]$$

$$= (\sin A \cos B)^2 - (\cos A \sin B)^2 = \sin^2 A \cos^2 B - \cos^2 A \sin^2 B$$

But that's not exactly $\sin^2 A - \sin^2 B$. Wait, unless there's another identity. Alternatively, maybe using product-to-sum formulas? Alternatively, let me compute $\sin^2 A - \sin^2 B$. Let me see:

$$\sin^2 A - \sin^2 B = (\sin A - \sin B)(\sin A + \sin B). \text{ Not sure if that helps.}$$

But let me check if $\sin(A + B) \cdot \sin(A - B)$ equals $\sin^2 A - \sin^2 B$. Let me compute $\sin^2 A - \sin^2 B$:

$$\sin^2 A - \sin^2 B = (0.4)^2 - (0.5)^2 = 0.16 - 0.25 = -0.09 = \frac{-9}{100}$$

Which is exactly the same as before! So that's a shortcut. Therefore:

$$\sin(A + B) \cdot \sin(A - B) = \sin^2 A - \sin^2 B = \frac{-9}{100}$$

Therefore, the answer is $\frac{-9}{100}$. So that confirms the previous result. Therefore, the value of the original expression is $\frac{-9}{100}$. But let me check once more with actual numbers to make sure. Let me compute $\arcsin(0.4)$ and $\arcsin(0.5)$ numerically, then compute the sines of their sum and difference, multiply them, and see if it's approximately $-0.09$. First, $\arcsin(0.5)$ is $\pi/6 \approx 0.5236$ radians. $\arcsin(0.4)$... Let me calculate that. Let me recall that $\sin(0.4115 \text{ radians}) \approx 0.4$. Let me check with calculator: Well, $\sin(0.4115) \approx \sin(0.4115) \approx 0.4$.

Let me check with calculator steps: But since I don't have a calculator here, maybe approximate. Alternatively, use known values. For example, $\sin(\pi/6) = 0.5$, $\sin(\pi/12) \approx 0.2588$, so 0.4 is between $\pi/12$ and $\pi/6$. Let me approximate $\arcsin(0.4)$.

**Response Generated by Qwen3-8B (4/4)**

Let me use Taylor series or something? Maybe not. Alternatively, just take that $\arcsin(0.4) \approx 0.4115$ radians (as I thought before). Let me take that as approximate value. So $A \approx 0.4115$ radians, $B = \pi/6 \approx 0.5236$ radians. Compute $A + B \approx 0.4115 + 0.5236 \approx 0.9351$ radians. Compute $\sin(0.9351)$. Let me recall that $\sin(\pi/3) \approx 0.8660$, $\pi/3 \approx 1.0472$ radians. So 0.9351 is less than $\pi/3$. Let me compute $\sin(0.9351)$. Let me use calculator-like approximation.

Alternatively, use the value we found earlier: $\frac{2\sqrt{3}+\sqrt{21}}{10}$. Let me compute that numerically. First, $\sqrt{3} \approx 1.732$, $\sqrt{21} \approx 4.5837$. So $2\sqrt{3} \approx 3.464$, $\sqrt{21} \approx 4.5837$. Then $2\sqrt{3} + \sqrt{21} \approx 3.464 + 4.5837 \approx 8.0477$. Divide by 10: $\approx 0.80477$. So $\sin(A + B) \approx 0.8048$. Similarly, $\sin(A - B) = \frac{2\sqrt{3}-\sqrt{21}}{10} \approx \frac{3.464-4.5837}{10} \approx \frac{-1.1197}{10} \approx -0.11197$. Therefore, multiplying them: $0.8048 \cdot (-0.11197) \approx -0.090$. Which is $-0.09$, which is $-\frac{9}{100}$. So that matches. Therefore, the exact value is $-\frac{9}{100}$, and the approximate decimal is $-0.09$. So that seems correct. Therefore, the answer is $-\frac{9}{100}$.

But let me check once more if there's any mistake in the sign. Wait, when I calculated $\sin(A - B)$, I had $\frac{2\sqrt{3}-\sqrt{21}}{10}$. Since $\sqrt{21} \approx 4.583$ and $2\sqrt{3} \approx 3.464$, so $2\sqrt{3} - \sqrt{21}$ is negative, so $\sin(A - B)$ is negative. Then $\sin(A + B)$ is positive, so multiplying them gives negative. Which matches with $-\frac{9}{100}$. So that's correct. Alternatively, if I think about angles: $A$ is $\arcsin(0.4)$, which is in the first quadrant, $B$ is $\arcsin(0.5)$ which is $\pi/6$, also first quadrant. So $A + B$ is the sum of two first quadrant angles. Depending on their measures, $A + B$ could be in the first or second quadrant. Similarly, $A - B$ could be positive or negative.

Let me check if $A - B$ is positive or negative. Since $\arcsin(0.4) \approx 0.4115$ radians, $\arcsin(0.5) \approx 0.5236$ radians. So $A - B \approx 0.4115 - 0.5236 \approx -0.1121$ radians. So a negative angle, which is equivalent to a positive angle by adding $2\pi$, but sine is an odd function, so $\sin(-x) = -\sin x$. Therefore, $\sin(A - B) = -\sin(B - A)$. So the negative value makes sense. Therefore, multiplying $\sin(A + B)$ which is positive with $\sin(A - B)$ which is negative gives a negative result. So $-\frac{9}{100}$ is correct. Therefore, I think the answer is $-\frac{9}{100}$. Let me check once more with another identity.

Alternatively, using complex numbers or other methods? Maybe not necessary. Since two different approaches gave me the same answer, and numerical approximation also matches, I think it's safe. But just to make sure, let me compute $\sin^2 A - \sin^2 B$:

$$\sin^2 A = (0.4)^2 = 0.16$$

**Response Generated by RL-Finetuned Qwen3-8B using Critique-GRPO (1/2)**

Okay, so I need to evaluate this expression: $\sin(\arcsin 0.4 + \arcsin 0.5)$ multiplied by $\sin(\arcsin 0.5 - \arcsin 0.4)$. Hmm, let me think.

First, I remember that `arcsin` is the inverse sine function, so $\arcsin 0.4$ is an angle whose sine is $0.4$, and similarly for $\arcsin 0.5$. Let me denote some variables to make this easier. Let me set $\alpha = \arcsin 0.4$ and $\beta = \arcsin 0.5$. Then the expression becomes $\sin(\alpha + \beta) * \sin(\beta - \alpha)$.

Alright, so I need to compute $\sin(\alpha+\beta)$ and $\sin(\beta-\alpha)$ separately and then multiply them. Maybe I can use sine–addition formulas here. Let me recall that $\sin(A+B) = \sin A \, \cos B + \cos A \, \sin B$, and $\sin(A - B) = \sin A \, \cos B - \cos A \, \sin B$.

Since $\alpha$ and $\beta$ are `arcsin` values, I know that $\sin \alpha = 0.4$ and $\sin \beta = 0.5$. Therefore, I need to find $\cos \alpha$ and $\cos \beta$ to use in the formulas.

For $\cos \alpha$, since $\sin^2 \alpha + \cos^2 \alpha = 1$, we have

$$\cos \alpha = \sqrt{1 - \sin^2 \alpha} = \sqrt{1 - 0.16} = \sqrt{0.84}.$$

Note that $0.84 = \frac{84}{100} = \frac{21}{25}$, so $\cos \alpha = \sqrt{\frac{21}{25}} = \frac{\sqrt{21}}{5}$.

Similarly,

$$\cos \beta = \sqrt{1 - \sin^2 \beta} = \sqrt{1 - 0.25} = \sqrt{0.75} = \sqrt{\tfrac{3}{4}} = \frac{\sqrt{3}}{2}.$$

Now, let me compute $\sin(\alpha + \beta)$. Using the formula,

$$\sin(\alpha + \beta) = \sin \alpha \, \cos \beta + \cos \alpha \, \sin \beta = 0.4\left(\frac{\sqrt{3}}{2}\right) + \left(\frac{\sqrt{21}}{5}\right)(0.5).$$

Compute each term:

$$0.4 \cdot \frac{\sqrt{3}}{2} = \frac{2}{5} \cdot \frac{\sqrt{3}}{2} = \frac{\sqrt{3}}{5}, \quad \frac{\sqrt{21}}{5} \cdot 0.5 = \frac{\sqrt{21}}{10}.$$

Therefore

$$\sin(\alpha + \beta) = \frac{\sqrt{3}}{5} + \frac{\sqrt{21}}{10} = \frac{2\sqrt{3} + \sqrt{21}}{10}.$$

Similarly, compute $\sin(\beta - \alpha)$:

$$\sin(\beta - \alpha) = \sin \beta \, \cos \alpha - \cos \beta \, \sin \alpha = 0.5\left(\frac{\sqrt{21}}{5}\right) - \frac{\sqrt{3}}{2}(0.4).$$

Thus

$$\sin(\beta - \alpha) = \frac{\sqrt{21}}{10} - \frac{2\sqrt{3}}{10} = \frac{\sqrt{21} - 2\sqrt{3}}{10}.$$

Therefore, the original expression is

$$\left(\frac{2\sqrt{3} + \sqrt{21}}{10}\right)\left(\frac{\sqrt{21} - 2\sqrt{3}}{10}\right).$$

Let $A = 2\sqrt{3} + \sqrt{21}$ and $B = \sqrt{21} - 2\sqrt{3}$; then the product is $\frac{A\,B}{100}$. Note that $A\,B = (\sqrt{21})^2 - (2\sqrt{3})^2 = 21 - 12 = 9$, so the value is $\frac{9}{100}$.

**Response Generated by RL-Finetuned Qwen3-8B using Critique-GRPO (2/2)**

Therefore, the value of the original expression is $\dfrac{9}{100}$, which is 0.09.

Let me check with approximate values. Take $\arcsin 0.5 = \frac{\pi}{6} \approx 0.5236$ rad and $\arcsin 0.4 \approx 0.4115$ rad. Then $\alpha + \beta \approx 0.9351$ rad, so $\sin(\alpha + \beta) \approx 0.805$; also $\beta - \alpha \approx 0.1121$ rad, so $\sin(\beta - \alpha) \approx 0.1118$. Their product $0.805 \times 0.1118 \approx 0.09$, confirming $\dfrac{9}{100}$.

**Final Answer**

$$\boxed{\dfrac{9}{100}}$$

To evaluate the expression

$$\sin(\arcsin 0.4 + \arcsin 0.5) \cdot \sin(\arcsin 0.5 - \arcsin 0.4),$$

we begin by introducing a substitution to simplify the notation. Let

$$\alpha = \arcsin 0.4, \qquad \beta = \arcsin 0.5,$$

so that $\sin\alpha = 0.4$ and $\sin\beta = 0.5$.

We rewrite the expression as $\sin(\alpha + \beta)\,\sin(\beta - \alpha)$.

**Step 1: Use the Sine Addition and Subtraction Formulas**

Recall

$$\sin(\alpha + \beta) = \sin\alpha\,\cos\beta + \cos\alpha\,\sin\beta, \qquad \sin(\beta - \alpha) = \sin\beta\,\cos\alpha - \cos\beta\,\sin\alpha.$$

Using

$$\cos\alpha = \frac{\sqrt{21}}{5}, \qquad \cos\beta = \frac{\sqrt{3}}{2},$$

we compute

$$\sin(\alpha + \beta) = \frac{2\sqrt{3} + \sqrt{21}}{10}, \quad \sin(\beta - \alpha) = \frac{\sqrt{21} - 2\sqrt{3}}{10}.$$

**Step 2: Multiply the Two Sine Values**

$$\left(\frac{2\sqrt{3} + \sqrt{21}}{10}\right)\left(\frac{\sqrt{21} - 2\sqrt{3}}{10}\right) = \frac{(\sqrt{21})^2 - (2\sqrt{3})^2}{100} = \frac{9}{100}.$$

**Final Answer**

$$\boxed{\dfrac{9}{100}}$$

## V    AN EXAMPLE OF FAILED REFINEMENT

The following example demonstrates a failed refinement using a CoT critique, where RL-finetuned Qwen3-8B using Critique-GRPO fails to follow the critique to refine its responses.

---

**Question**

Example 15: Let $a, b, c$ be positive real numbers, and $abc + a + c = b$, find the maximum value of

$$p = \frac{2}{a^2 + 1} - \frac{2}{b^2 + 1} + \frac{3}{c^2 + 1}.$$

---

**CoT Critique**

The student's solution contains several errors and misunderstandings. Let's go through the mistakes step-by-step:

1. **Misinterpretation of the Constraint:** - The student correctly rearranges the constraint $abc + a + c = b$ to express $b$ in terms of $a$ and $c$, but then incorrectly assumes $b = \tan(\alpha + \gamma)$ without proper justification. The trigonometric substitution should be used more carefully, ensuring that the constraint is satisfied throughout.

2. **Incorrect Use of Trigonometric Identities:** - The student attempts to use trigonometric identities to simplify the expression for $p$, but the application is incorrect. The identities used do not lead to a valid simplification of the expression.

3. **Incorrect Derivative Calculation:** - The student attempts to find the maximum by taking derivatives, but the calculations are incorrect. The derivative should be taken with respect to the correct variables, and the critical points should be found accurately.

4. **Incorrect Evaluation of $p$:** - The student evaluates $p$ at specific points, but the calculations are inconsistent and do not match the constraint. The values chosen for $a$, $b$, and $c$ do not satisfy the constraint $abc + a + c = b$.

5. **Incorrect Final Answer:** - The student concludes with an incorrect maximum value of $p = \frac{105}{32}$, which is not supported by the calculations. The correct maximum value, as derived from a proper analysis, should be $\frac{10}{3}$.

**Conclusion:** wrong [END]

**Incorrect Initial Response (1/2)**

Okay, so I need to find the maximum value of this expression:

$$p = \frac{2}{a^2 + 1} - \frac{2}{b^2 + 1} + \frac{3}{c^2 + 1},$$

given that $a, b, c$ are positive real numbers satisfying $abc + a + c = b$. Hmm, let me start by understanding the constraint equation better.

The equation is:

$$abc + a + c = b.$$

Maybe I can rearrange this to express one variable in terms of the others. Let me try to solve for $b$. Let's see:

$$abc + a + c = b \implies b - abc = a + c \implies b(1 - ac) = a + c \implies b = \frac{a + c}{1 - ac}.$$

Wait, but since $a, b, c$ are positive real numbers, the denominator $1 - ac$ must be positive because the numerator $a + c$ is positive. Therefore:

$$1 - ac > 0 \implies ac < 1.$$

So, the product of $a$ and $c$ must be less than 1. That's an important condition.

—

Now, the expression for $p$ involves $a, b, c$. Since $b$ is expressed in terms of $a$ and $c$, perhaps I can substitute that into $p$ and then try to maximize with respect to $a$ and $c$. Let me try that. Substituting $b = \frac{a+c}{1-ac}$ into $p$, we get:

$$p = \frac{2}{a^2 + 1} - \frac{2}{\left(\frac{a+c}{1-ac}\right)^2 + 1} + \frac{3}{c^2 + 1}.$$

This looks complicated. Maybe there's a substitution or some symmetry here. Let me check if there's a way to simplify the expression for $b$. Alternatively, maybe using substitution variables.

—

### Trigonometric Substitution

Let's consider substituting variables to simplify the constraint. For example, sometimes in equations with products, substitutions like $x = a, y = c$ are helpful, but I'm not sure in this case. Alternatively, maybe trigonometric substitution? Since we have terms like $a^2 + 1$, which reminds me of $\tan \theta$ substitutions. For example:
- Let $a = \tan \alpha$, $c = \tan \gamma$, where $\alpha, \gamma \in (0, \pi/2)$. - Then:

$$a^2 + 1 = \sec^2 \alpha \implies \frac{1}{a^2 + 1} = \cos^2 \alpha,$$

and similarly for $c$.
Let me try that. Substituting $a = \tan \alpha$ and $c = \tan \gamma$, and using the constraint $abc + a + c = b$, we get:

$$\tan \alpha \cdot b \cdot \tan \gamma + \tan \alpha + \tan \gamma = b.$$

Solving for $b$:

$$b(\tan \alpha \tan \gamma - 1) = \tan \alpha + \tan \gamma \implies b = \frac{\tan \alpha + \tan \gamma}{\tan \alpha \tan \gamma - 1}.$$

Wait, but earlier we had:

$$b = \frac{a + c}{1 - ac}.$$

Since $ac = \tan \alpha \tan \gamma$, so $1 - ac = 1 - \tan \alpha \tan \gamma$. Therefore:

$$b = \frac{\tan \alpha + \tan \gamma}{1 - \tan \alpha \tan \gamma}.$$

—

**Incorrect Initial Response (2/2)**

### Simplifying Further

Notice that:

$$\frac{\tan\alpha + \tan\gamma}{1 - \tan\alpha\tan\gamma} = \tan(\alpha+\gamma).$$

Thus:

$$b = \tan(\alpha+\gamma).$$

Therefore, since $b > 0$, $\tan(\alpha+\gamma) > 0$. Since $\alpha, \gamma \in (0, \pi/2)$, $\alpha + \gamma \in (0, \pi/2)$. This gives:

$$\alpha + \gamma < \frac{\pi}{2}.$$

—

### Substituting Back into $p$

Using the substitution:

$$p = \frac{2}{a^2+1} - \frac{2}{b^2+1} + \frac{3}{c^2+1}.$$

Since $a = \tan\alpha$, we have:

$$\frac{1}{a^2+1} = \cos^2\alpha, \quad \frac{1}{c^2+1} = \cos^2\gamma, \quad \frac{1}{b^2+1} = \cos^2(\alpha+\gamma).$$

Thus:

$$p = 2\cos^2\alpha - 2\cos^2(\alpha+\gamma) + 3\cos^2\gamma.$$

—

### Reducing the Problem

The problem now reduces to maximizing:

$$p = 2\cos^2\alpha - 2\cos^2(\alpha+\gamma) + 3\cos^2\gamma,$$

subject to $\alpha, \gamma > 0$ and $\alpha + \gamma < \pi/2$.
Let $\theta = \alpha + \gamma$. Then:

$$p = 2\cos^2\alpha - 2\cos^2\theta + 3\cos^2(\theta - \alpha).$$

—

### Symmetry Assumption

Assume $\alpha = \gamma$, which implies $\theta = 2\alpha$. Then:

$$p = 2\cos^2\alpha - 2\cos^2(2\alpha) + 3\cos^2\alpha = 5\cos^2\alpha - 2\cos^2(2\alpha).$$

Using the identity $\cos(2\alpha) = 2\cos^2\alpha - 1$, we get:

$$\cos^2(2\alpha) = (2\cos^2\alpha - 1)^2.$$

Let $x = \cos^2\alpha$, so $x \in \left(\frac{1}{2}, 1\right)$. Substituting:

$$p = 5x - 2(2x-1)^2 = 5x - 2(4x^2 - 4x + 1) = -8x^2 + 13x - 2.$$

This is a quadratic equation, and its maximum occurs at:

$$x = \frac{-b}{2a} = \frac{13}{16}.$$

Substituting $x = \frac{13}{16}$ back into $p$:

$$p = -8\left(\frac{13}{16}\right)^2 + 13\left(\frac{13}{16}\right) - 2 = \frac{105}{32}.$$

—

### Final Answer

$$\boxed{\frac{105}{32}}$$

**Incorrect Refined Response (1/2)**

The given problem asks us to find the **maximum value** of the expression:

$$p = \frac{2}{a^2 + 1} - \frac{2}{b^2 + 1} + \frac{3}{c^2 + 1}$$

subject to the constraint:

$$abc + a + c = b$$

where $a, b, c$ are **positive real numbers**.

—

### Step 1: Analyze the Constraint

From the constraint:

$$abc + a + c = b$$

we can rearrange to isolate $b$:

$$b(1 - ac) = a + c \quad \Rightarrow \quad b = \frac{a + c}{1 - ac}.$$

Since $a, b, c > 0$, the denominator $1 - ac$ must be positive, so:

$$ac < 1$$

This condition will be important in limiting the domain of possible values for $a$ and $c$.

—

### Step 2: Trigonometric Substitution

To simplify the expression for $p$, we use the following substitutions:
- Let $a = \tan \alpha$ - Let $c = \tan \gamma$
Then, since $a^2 + 1 = \sec^2 \alpha$, we have:

$$\frac{1}{a^2 + 1} = \cos^2 \alpha, \quad \frac{1}{c^2 + 1} = \cos^2 \gamma.$$

Also, from the tangent addition formula:

$$b = \frac{a + c}{1 - ac} = \tan(\alpha + \gamma).$$

Therefore:

$$\frac{1}{b^2 + 1} = \cos^2(\alpha + \gamma).$$

Substituting into $p$, we get:

$$p = 2\cos^2 \alpha - 2\cos^2(\alpha + \gamma) + 3\cos^2 \gamma.$$

Let $\theta = \alpha + \gamma$, and express $\gamma = \theta - \alpha$. Then:

$$p = 2\cos^2 \alpha - 2\cos^2 \theta + 3\cos^2(\theta - \alpha).$$

—

**Incorrect Refined Response (2/2)**
### Step 3: Assume Symmetry for Simplification
To simplify further, assume $\alpha = \gamma$, which implies $\theta = 2\alpha$. Then:

$$p = 2\cos^2\alpha - 2\cos^2(2\alpha) + 3\cos^2\alpha = 5\cos^2\alpha - 2\cos^2(2\alpha).$$

Using the double-angle identity $\cos(2\alpha) = 2\cos^2\alpha - 1$, we get:

$$\cos^2(2\alpha) = (2\cos^2\alpha - 1)^2.$$

Let $x = \cos^2\alpha$, so $x \in \left(\frac{1}{2}, 1\right)$. Then:

$$p = 5x - 2(2x - 1)^2 = -8x^2 + 13x - 2.$$

This is a quadratic in $x$, opening downward, so its maximum occurs at:

$$x = \frac{-b}{2a} = \frac{13}{16}.$$

Substituting back:

$$p = -8\left(\frac{13}{16}\right)^2 + 13\left(\frac{13}{16}\right) - 2 = \frac{105}{32}.$$

—

### Final Answer

$$\boxed{\dfrac{105}{32}}$$

