# OpenReview forum: "Advancing LLM Reasoning with Natural Language and Numerical Feedback"
_ICLR.cc/2026/Conference — Submitted to ICLR 2026_

### Official Review · Reviewer_StXP · 2025-10-24

**Soundness:** 4
**Presentation:** 3
**Contribution:** 2
**Rating:** 4
**Confidence:** 4

**Summary:**

This work proposes a method called Critique-GRPO, which extends the GRPO framework by integrating steps of critiques and self-refinement, and shows its effectiveness over the GRPO baseline.

**Strengths:**

+ The effectiveness is supported by experiments on eight reasoning tasks across three Qwen base models.
+ Reinforcing the self-refinement abilities during RL fine-tuning may increase the test-time performance.
+ The method design is a reasonable and well-justified extension.

**Weaknesses:**

+ While the components (natural language feedback, self-refinement) are not new, the paper's novelty lies in their integration into RL with verifiable rewards. However, this integration could be seen as an incremental step.
+ The introduction of critiques and self-refinement increases the total training time. The efficiency should be reported for comparisons of the additional cost.
+ The method relies on a much larger LLM such as GPT-4o as the critique model. This makes the comparison between the proposed method and GRPO baseline relatively unfair due to new supervision signals. The authors should also consider other distillation method for comparisons. Though in section 5.4, self-critique experiments are reported, the effectiveness of the results in the main table still relies on a powerful critique model.
+ The experiments are conducted on Qwen models. Please consider one beyond this family for supporting the generalization.
+ The fundamental learning signal is still based on sparse, outcome-level rewards (e.g., +1 or 0 for the entire answer). The method uses NL critiques to generate a better trajectory to learn from, but it doesn't solve the core credit assignment problem. This weak, outcome-level supervision may still lead to performance plateaus, as the model cannot easily pinpoint which specific reasoning steps are responsible for a failure.

**Questions:**

N/A

---

> ### Author Response · Authors · 2025-11-13
> **Response to Reviewer StXP (1/2)**
>
> Thank you for your feedback. We provide the following responses to address your concerns:
>
> **Response to Weakness 1:**
>
> We appreciate the opportunity to clarify our contributions. Our work introduces a fundamentally new paradigm for online RL that integrates natural language critiques with numerical rewards for real-time learning, representing a meaningful advance beyond prior work.
>
> 1. Novel Integration of Natural Language and Numerical Feedback
>
> Our work is the first to successfully combine natural language critiques with numerical rewards in an online RL setting. This integration addresses unique challenges including performance plateaus faced by solely relying on numerical rewards (Section 3, 5). Natural language critiques provide informative, actionable feedback that enables more effective navigation of complex reasoning tasks compared to scalar rewards alone. [1]
>
> 2. Superior Performance Through Online Policy Optimization
>
> While prior work has explored natural language critiques in offline settings (Critique-FT, CITL-FT), these methods rely on static datasets of pre-collected critiques or refinements. These offline SFT methods suffer from two major flaws: they cannot adapt to a policy's evolving mistakes in real-time, and they tend to mimic training data, leading to poor generalization, e.g., reduced performance on out-of-distribution (OOD) tasks in Table 2. Our framework solves these issues by integrating language critiques with numerical rewards in an online loop. This allows the model to learn from its own on-policy generations and refinements, fostering active exploration and adaptation. Empirical results in Table 2 confirm that this online approach significantly outperforms offline baselines, particularly in OOD generalization where prior methods struggle.
>
> 3. Algorithmic Innovations Enabling Hybrid Feedback
>
> Our key algorithmic contributions include: (1) Dual Optimization: Verbally solves the credit assignment problem by jointly optimizing initial and refined responses, enabling models to internalize critiques and eventually generate correct initial responses independently. (2) Response-Refinement Ratio Balancing & Reward Shaping: Address distributional shifts unique to critique-augmented online RL, ensuring stable training across diverse environments.
>
> 4. Adaptation to Diverse, Real-Time Feedback
>
> Unlike prior methods requiring high-quality static critiques, our framework handles diverse, real-time feedback and automatically penalizes incorrect refinements. This enables continuous adaptation in open-ended environments, significantly improving robustness and generalization.
>
> This proposed approach aligns with emerging efforts to scale RL beyond numerical rewards (https://x.com/karpathy/status/1944435412489171119) and lays the foundation for further research in online RL direction.
>
> **Response to Weakness 2:**
>
> Thank you for raising this important point. We acknowledge that incorporating critiques and self-refinement introduces additional computational overhead. Transparent reporting of this cost is indeed essential. To clarify:
>
> | Model         | Method                         | Training Time (GPU-Hours) | Average Accuracy (%) |
> |---------------|--------------------------------|---------------------------|-----------------------|
> | Qwen2.5-7B-Base    | R1-GRPO                       | 240                       | 41.18                |
> |               | Critique-GRPO (CoT-Critique)  | 245                       | 47.08                |
> | Qwen3-8B    | R1-GRPO                       | 296                       | 63.75                |
> |               | Critique-GRPO (CoT-Critique)  | 315                       | 68.26                |
>
> By implementing *asynchronous rollouts* on Critique-GRPO, we limited the additional cost to only 5 and 19 GPU-hours, representing a ~2% and ~6.5% increase over the GRPO baselines on A800 GPUs. This **acceptable overhead** is strongly justified by the +6 and +4.5 average point gains across 8 tasks, achieved using just 4K prompts enriched with natural language and numerical feedback. These results demonstrate significant improvements in both data efficiency and model performance, making the added cost highly worthwhile. We will include these detailed efficiency comparisons in Appendix M.
>
> Additionally, our preliminary results suggest that randomly sampling one failed initial response for refinement generation during training time yields comparable performance. Future researchers may consider this as a cost-effective alternative for real-world applications.
>
> [1] Provably Learning from Language Feedback

---

> ### Author Response · Authors · 2025-11-13
> **Response to Reviewer StXP (2/2)**
>
> **Response to Weakness 3:**
>
> Thank you for raising this concern regarding the reliance on a larger critique model and the fairness of comparisons with the GRPO baseline. We would like to clarify that the effectiveness of Critique-GRPO **does not depend on advanced models** and is compatible with various feedback sources, as validated in the paper.
>
> Specifically, Table 2 shows that Critique-GRPO with *indicative feedback (``The proposed solution is correct/incorrect.'')* derived from a rule-based reward system yields an average improvement of *+3.5 to 4 points* over the GRPO baseline. This highlights a fair and meaningful comparison that does not rely on any external models.
>
> Crucially, as shown in Table 2 and Table 4, Critique-GRPO with self-critiques (68.13) is virtually on par (<0.1% difference) with using GPT-4o-generated CoT critiques (68.26).  Additionally, we evaluate Critique-GRPO with alternative critique models, such as Llama3.1-405B and the fine-tuned DeepCritic-7B-RL1.5-PRM800K. The results below highlight the adaptability of Critique-GRPO to both proprietary and fine-tuned critique models:
>
> | Method             | Critique Model         | MATH 500 | Minerva MATH | Olympiad Bench | AMC23  | AIME24 | Theorem QA | GPQA Diamond | MMLU Pro | Avg.   |
> |---------------------|------------------------|----------|--------------|-----------------|--------|--------|------------|--------------|----------|--------|
> | Qwen2.5-7B-Base    | -                      | 60.80    | 20.20        | 30.40           | 35.00  | 13.30  | 21.60      | 28.79        | 46.24    | 32.04  |
> | R1-GRPO            | -                      | 74.00    | 32.00        | 38.50           | 42.50  | 16.70  | 40.60      | 33.33        | 51.81    | 41.18  |
> | Critique-GRPO      | Llama3.1-405B          | 78.40    | 37.90        | 41.00           | 62.50  | 16.70  | 42.80      | 38.89        | 56.12    | 46.79  |
> | Critique-GRPO      | GPT-4o                 | 77.80    | 36.80        | 42.40           | 62.50  | 20.00  | 44.00      | 37.88        | 55.28    | 47.08  |
> | Critique-GRPO      | DeepCritic-7B          | 79.20    | 40.10        | 44.00           | 60.00  | 23.30  | 43.10      | 36.87        | 57.28    | 47.98  |
>
> These findings demonstrate that Critique-GRPO is effective across a range of critique models and does not rely exclusively on powerful external models like GPT-4o. We will include these results in Section 5.6.
>
> **Response to Weakness 4:**
>
> Thank you for raising this point. We conducted additional **experiments using LLaMA3.2-Instruct-3B**. The results, summarized below, show that Critique-GRPO consistently outperforms the GRPO baseline, achieving an improvement of more than 5 points. This supports the robustness and generalizability of our method across different model architectures. We will include these results in Section 5.6.
>
> | Method               | MATH-500 | Minerva-MATH | Olympiad-Bench | AMC23 | AIME24 | TheoremQA | GPQA-Diamond | MMLU-Pro | Avg.  |
> |----------------------|----------|--------------|----------------|-------|--------|-----------|--------------|----------|-------|
> | Llama3.2-Instruct-3B | 46.60    | 17.30        | 15.30          | 27.50 | 13.30  | 21.80     | 22.22        | 38.16    | 22.52 |
> | +R1-GRPO             | 53.60    | 19.50        | 18.80          | 27.50 | 13.30  | 24.60     | 29.29        | 40.72    | 24.79 |
> | +Critique-GRPO       | 58.80    | 25.70        | 24.10          | 35.00 | 20.00  | 31.10     | 36.87        | 43.79    | 29.86 |
>
> **Response to Weakness 5:**
>
> Thank you for this insightful comment. We agree that outcome-level rewards present a credit assignment challenge. Our approach addresses this not by providing step-level numerical rewards, but by leveraging natural language critiques to guide an internal refinement process.
>
> Textual critiques—provided without expert demonstrations—enable the model to diagnose errors and generate corrected reasoning chains. This process, detailed in Section 3.2 and Appendix C, allows for verbal step-level credit assignment, helping the model identify and correct flawed steps. As shown in Tables 1, 5, and 6, this mechanism enables the model to overcome persistent failures and plateaus seen with numerical rewards alone (Appendix R). Moreover, the superiority of learning from textual feedback over scalar rewards has been well established in recent theoretical work [1].
>
> Additionally, the superior performance of Critique-GRPO compared to prior approaches using both outcome- and process- rewards (e.g., PRIME in Table 3) highlights the effectiveness of our method. We also recognize that integrating explicit fine-grained scalar rewards could be a complementary and promising future direction, one for which our framework is naturally compatible.
>
> We appreciate your valuable feedback and will incorporate these clarifications and additional results in our revised manuscript.
>
> [1] Provably Learning from Language Feedback

---

> ### Author Response · Authors · 2025-11-26
> **A Gentle Reminder**
>
> Dear Reviewer,
>
> Thank you once again for your valuable comments! As the discussion phase is approaching its end, we would like to kindly confirm whether we have sufficiently addressed all of your concerns (or at least some of them). Should there be any remaining questions requiring further clarification, please do not hesitate to let us know. If you are satisfied with our responses, we would greatly appreciate your consideration in adjusting the evaluation scores accordingly. We sincerely look forward to your feedback.

---

### Official Review · Reviewer_cPJg · 2025-10-31

**Soundness:** 3
**Presentation:** 3
**Contribution:** 3
**Rating:** 6
**Confidence:** 4

**Summary:**

The authors introduce **Critique-GRPO** as an online RL algorithm that blends plain numeric rewards with natural-language critiques—indicative, with ground truth, and CoT. The loop is simple: get an initial answer, ask for a critique, refine, and learn from both passes inside a PPO-style update. Compared to vanilla GRPO they drop clipping and the KL penalty, and instead add a policy-shaping term $\(f(x)=\frac{x}{x+\gamma}\)$ with $\(\gamma=0.1\)$ to boost low-probability tokens in the refined response. Training-wise, they keep a small handful of refinements per prompt and say **7:1 initial:refined** works best.

On results, they report gains across **Qwen2.5-7B-Base**, **Qwen2.5-Math-7B-Base**, and **Qwen3-8B** over several benchmarks, with the biggest bump when the critique is **CoT** rather than just a binary signal. The takeaway from the paper is that the natural-language feedback helps break RL plateaus, keeps entropy from collapsing, and nudges the better reasoning and thereby better performance on the benchmarks.

**Strengths:**

1.  **Straightforward and reproducible.** The method augments GRPO with critique-guided refinements and a simple shaping term. The pseudocode and stepwise description are clear, and the hyperparameter table supports replication.

2.  **Well-motivated.** The paper documents plateaus and limited gains under numeric-only RL, then shows that natural-language critiques—especially CoT—convert a subset of failures into solves. The causal narrative is plausible and supported by analyses.

3.  **Robust empirical gains.** Consistent improvements are reported across multiple math and OOD benchmarks, including on a non-math base and a reasoning-focused backbone. Gains persist under greedy (temperature=0) evaluation.

4.  **Training-dynamics transparency.** The inclusion of entropy and response-length curves, along with a rationale for the 7:1 initial:refined sampling ratio, provides useful visibility beyond end metrics.

**Weaknesses:**

1. The results are predominantly Qwen based. After reading some papers recently which claim RL w/ random rewards work really well with Qwen (https://www.interconnects.ai/p/reinforcement-learning-with-random), I am skeptical about papers which show gains with qwen models. It would be nice if the authors can replicate the same experiments with a different base model
2. If ground truth is leaked (even partially) in case of Critique w/ GT, the policy can potentially learn to pattern-match GT traces rather than reason. I am curious to know what authors think regarding this.
3. The method is computationally very expensive than doing RL/SFT since you generate initial responses and refinements, plus the critique tokens and longer contexts for second-pass decoding.
4. You modify the objective (drop GRPO clipping/KL) and add critiques and add refinement sampling. Would be curious to know where the gains are actually coming from -- are wins from looser optimization or language feedback? Right now this is confounded.

**Questions:**

1. How did you get the numbers for Figure 2b? Did you use judge to get these numbers? If yes, what kind of judge?
2. Is there any reason why SFT is not done for RL experiments? Do we get the same gains if we do RL on the SFTed model as a base?

---

> ### Author Response · Authors · 2025-11-13
> **Response to Reviewer cPJg (1/2)**
>
> Thank you for your thoughtful feedback. We address each concern below in a clear and comprehensive manner:
>
> **Response to Weakness 1:**
> We appreciate this important point. To assess the generalizability of our approach beyond the Qwen model family, we conducted **additional experiments using LLaMA3.2-Instruct-3B**. As shown in our updated results, Critique-GRPO consistently outperforms the GRPO baseline, achieving an average improvement of approximately +5 points. These findings support the robustness and cross-architecture effectiveness of our method. We will include these results in Section 5.6.
>
> | Method               | MATH-500 | Minerva-MATH | Olympiad-Bench | AMC23 | AIME24 | TheoremQA | GPQA-Diamond | MMLU-Pro | Avg.  |
> |----------------------|----------|--------------|----------------|-------|--------|-----------|--------------|----------|-------|
> | Llama3.2-Instruct-3B | 46.60    | 17.30        | 15.30          | 27.50 | 13.30  | 21.80     | 22.22        | 38.16    | 22.52 |
> | +R1-GRPO             | 53.60    | 19.50        | 18.80          | 27.50 | 13.30  | 24.60     | 29.29        | 40.72    | 24.79 |
> | +Critique-GRPO       | 58.80    | 25.70        | 24.10          | 35.00 | 20.00  | 31.10     | 36.87        | 43.79    | 29.86 |
>
> **Response to Weakness 2:**
>
> Thank you for raising this concern. We address it with the following points:
>
> 1. As noted in Section 3.2, the Critique w/ GT setting *exposes only the final ground-truth answer*, without revealing any intermediate reasoning steps. Thus, the model must still *reason step-by-step* to derive the correct answer.
>
> 2. As shown in Tables 1, 5, and 6, performance differences between critiques with and without GT are *negligible*, indicating that the model is not exploiting leaked traces.
>
> 3. In Table 2, models trained with *Critique w/ GT* generalize better than GRPO baselines across *three out-of-distribution tasks*—a result unlikely if the model were merely pattern-matching.
>
> 4. Upon manual inspection of model outputs, we found *no evidence of trace-level pattern-matching*. This is likely due to the *difficulty of the dataset (OpenR1-Math-220k)*, which requires multi-step reasoning.
>
> In summary, both qualitative and quantitative analyses support that our model is learning to *reason*, not mimic.
>
> **Response to Weakness 3:**
>
> Thank you for raising this important point. We acknowledge that incorporating critiques and self-refinement introduces additional computational overhead. Transparent reporting of this cost is indeed essential. To clarify:
>
> | Model         | Method                         | Training Time (GPU-Hours) | Average Accuracy (%) |
> |---------------|--------------------------------|---------------------------|-----------------------|
> | Qwen2.5-7B-Base    | R1-GRPO                       | 240                       | 41.18                |
> |               | Critique-GRPO (CoT-Critique)  | 245                       | 47.08                |
> | Qwen3-8B    | R1-GRPO                       | 296                       | 63.75                |
> |               | Critique-GRPO (CoT-Critique)  | 315                       | 68.26                |
>
> By implementing *asynchronous rollouts* on Critique-GRPO, we limited the additional cost to only 5 and 19 GPU-hours, representing a ~2% and ~6.5% increase over the GRPO baselines on A800 GPUs. This **acceptable overhead** is strongly justified by the +6 and +4.5 average point gains across 8 tasks, achieved using just 4K prompts enriched with natural language and numerical feedback. These results demonstrate significant improvements in both data efficiency and model performance, making the added cost highly worthwhile. We will include these detailed efficiency comparisons in Appendix M.
>
> Additionally, our preliminary results suggest that randomly sampling one failed initial response for refinement generation during training time yields comparable performance. Future researchers may consider this as a cost-effective alternative for real-world applications.

---

> ### Author Response · Authors · 2025-11-13
> **Response to Reviewer cPJg (2/2)**
>
> **Response to Weakness 4:**
>
> Thank you for highlighting this. In early experiments, we removed GRPO clipping but later re-introduced it for training stability. Unfortunately, we did not update Section 4.2 accordingly—this will be corrected in the revised version. In addition, directly incorporating refinements without *sampling one only* often lead to performance degradation. Although early training steps may show temporary gains, this strategy often results in *entropy explosion*, caused by distribution shifts introduced by natural language feedback.
>
> We conducted fine-grained ablation analyses on the training objective of Critique-GRPO to isolate the impact of each modification. Since the Critique-GRPO objective is derived from GRPO, we sequentially evaluated the effects of the following modifications: 1. removing KL loss, 2. introducing natural language feedback, 3. refinement sampling, 4. policy shaping. The results, summarized below, demonstrate that both looser optimization and natural language feedback positively contribute to policy learning. We will present the detailed discussion in Appendix N.
>
> | Method | Modifications |||| Math (ID) ||||| Science & General (OOD) ||| Avg. |
> |--------|---------|---------|---------|---------|---------|---------|---------|---------|---------|---------|---------|---------|---------|
> | | KL Loss Removal | Language Feedback | Refinement Sampling | Policy Shaping | MATH 500 | Minerva MATH | Olympiad Bench | AMC23 | AIME24 | Theorem QA | GPQA Diamond | MMLU Pro | |
> | Qwen-base | - | - | - | - | 60.80 | 20.20 | 30.40 | 35.00 | 13.30 | 21.60 | 28.79 | 46.24 | 32.04 |
> | + R1-GRPO | × | × | × | × | 74.00 | 32.00 | 38.50 | 42.50 | 16.70 | 40.60 | 33.33 | 51.81 | 41.18 |
> | + Critique-GRPO (CoT Critique) | ✓ | × | × | × | 78.40 | 34.90 | 39.90 | 40.00 | 13.30 | 43.10 | 38.89 | 52.83 | 42.66 |
> | | ✓ | ✓ | × | × | 77.00 | 30.90 | 42.40 | 52.50 | 13.30 | 41.80 | 33.33 | 54.88 | 43.26 |
> | | ✓ | ✓ | ✓ | × | 77.40 | 41.00 | 39.70 | 45.00 | 16.70 | 42.60 | 34.34 | 54.88 | 43.95 |
> | | ✓ | ✓ | ✓ | ✓ | 77.80 | 36.80 | 42.40 | 62.50 | 20.00 | 44.00 | 37.88 | 55.28 | 47.08 |
>
> **Response to Question 1:**
> Thank you for raising this question. Due to space constraints in the main paper, we provide a detailed description of our methodology in Appendix C. Figure 2b aims to analyze the contribution of six key cognitive behaviors to self-improving reasoning during RL fine-tuning. Specifically, we examine how these behaviors contribute to solving problems that the base model previously failed to solve.
>
> We identify 87 previously unsolved problems from Minerva-Math dataset and 37 previously unsolved problems from GPQA-Diamond dataset on Qwen2.5-7B-Base. To detect the presence of each cognitive behavior in the model's refined responses, we used GPT-4o as an automatic judge, guided by manually crafted prompts (see Appendix P). Each behavior is scored 1 if present in a response, and 0 otherwise. Finally, the *average success contribution* reported in Figure 2b is computed as:  (Total Number of behavior occurrences) / (Total number of analyzed responses). We will add more details in Section 3.1 and Appendix C to ensure clarity.
>
> **Response to Question 2:**
>
> Thank you for raising this question. Our experiments target the *R1-zero training paradigm* for flexible RL scaling, but *our method is compatible with SFT*. For example, Table 2 includes results for Qwen3-8B, a model that has undergone extensive supervised fine-tuning. Our Critique-GRPO approach consistently outperforms the GRPO baseline, confirming that our method remains effective even when built upon SFTed models.
>
> We appreciate your valuable feedback and will incorporate these clarifications and additional results in our revised manuscript to ensure greater transparency and rigor. We hope these responses address your concerns. We look forward to continuing this discussion and welcome further feedback to resolve any remaining questions.

---

> ### Author Response · Authors · 2025-11-26
> **A Gentle Reminder**
>
> Dear Reviewer,
>
> Thank you once again for your valuable comments! As the discussion phase is approaching its end, we would like to kindly confirm whether we have sufficiently addressed all of your concerns (or at least some of them). Should there be any remaining questions requiring further clarification, please do not hesitate to let us know. We sincerely look forward to your feedback.

---

> ### Comment · Reviewer_cPJg · 2025-11-26
>
> Dear authors,
>
> Thank you for the detailed rebuttal and for running the additional experiments. The new LLaMA3.2-Instruct-3B results help address my concern about generality beyond Qwen (Weakness 1). Your analysis of the Critique w/ GT setting (similar performance with/without GT, OOD generalization, and manual inspection) alleviates most of my worries about superficial reasoning, and the clarified compute breakdown and training-objective ablations address my questions about overhead and stability (Weaknesses 3–4).
>
> Overall, I consider my main concerns addressed and will keep my current score unchanged.
>
> Best regards,

---

> ### Author Response · Authors · 2025-11-28
> **Response to Reviewer cPJg**
>
> We appreciate the follow-up and are glad to know we have resolved most of your concerns. Regarding the potential for superficial reasoning, we would like to offer two key clarifications:
>
> 1. Training Dynamics: During training, the model learns primarily from initial responses rather than solely relying on critique-guided refinements. This training objective naturally prioritizes deep reasoning. In addition, the tasks are challenging and require accurate, step-by-step derivation to reach the correct answer, .
>
> 2. Empirical Evidence (OOD): Crucially, at test time, no critique guidance is provided; the model must independently generate the reasoning chain. If the model had learned superficial patterns, we would expect a performance drop, particularly on Out-of-Distribution (OOD) tasks. However, Critique-GRPO demonstrates consistent improvements on both in-distribution and, notably, OOD benchmarks. This generalization capability confirms that the model has improved its intrinsic reasoning abilities rather than relying on superficial cues.
>
>
> We hope these responses address your concerns. Should there be any remaining questions requiring further clarification, please do not hesitate to let us know. If you are satisfied with our responses, we would greatly appreciate your consideration in adjusting the evaluation scores accordingly. We sincerely look forward to your feedback.

---

### Official Review · Reviewer_tB8n · 2025-11-01

**Soundness:** 3
**Presentation:** 3
**Contribution:** 2
**Rating:** 4
**Confidence:** 3

**Summary:**

This paper introduces Critique-GRPO, an online RL framework that combines natural-language feedback and numerical feedback to enhance LLM reasoning. Building on the GRPO algorithm, Critique-GRPO allows a model to learn jointly from initial responses and critique-guided refinements, where critiques may be rule-based or model-generated. A shaping function further prioritizes correct and unfamiliar refinements. Experiments on three Qwen-series backbones across eight reasoning benchmarks show consistent gains over both supervised and RL baselines, with meaningful improvements and strong self-critiquing capabilities.

**Strengths:**

Strengths
- The integration of textual feedback into online RL is a neat idea. Unlike prior works that either transform critiques into scalar rewards or use them offline, Critique-GRPO integrates them within the RL loop, enabling simultaneous exploration and refinement
- Comprehensive evaluation and consistent empirical gains. Results across Qwen2.5-7B-Base, Qwen2.5-Math-7B-Base, and Qwen3-8B on eight reasoning datasets demonstrate broad effectiveness, including out-of-distribution generalization
- Analysis section provide qualitative and quantitative insights on entropy dynamics, response-length efficiency, and self-critique exploration, which helps interpret why NLF sustains exploration while improving precision

**Weaknesses:**

Weaknesses
- Limited novelty beyond combination. Although integrating natural-language feedback with GRPO is useful, the framework mainly reuses known components like textual critique generation (introduced in Critique FT) on top of GRPO. The algorithmic novelty (e.g., removing clipping/KL + adding a shaping weight) is incremental compared to existing works (e.g, Dr. GRPO, DAPO, LUFFY, etc.)
- Heavy reliance on Qwen models and numerical comparisons. All experiments are performed on Qwen-series backbones. It remains unclear whether Critique-GRPO generalizes to other architectures (e.g., Llama-3). Including one non-Qwen model would strengthen claims of generality, as in recent RLVR works[1, 2].
- Ablations and control analyses appears incomplete. The paper compares variants with different critique types, but lacks fine-grained ablations isolating (i) the shaping function, (ii) removal of clipping/KL terms, and (iii) critique quality (noisy vs. clean). Such ablations would clarify which component drives most of the gain.

## References
[1]. Spurious Rewards: Rethinking Training Signals in RLVR. arXiv:2506.10947

[2]. The Surprising Effectiveness of Negative Reinforcement in LLM Reasoning. arXiv:2506.01347

**Questions:**

- Since the textual feedback is generated by the reward model, how sensitive is Critique-GRPO to the quality and length of CoT critiques? For example, how does the method perform when critiques are partially wrong or adversarial?
- Could the authors provide more details on training cost? Specifically, how is the training efficiency (e.g., wall-time/GPU hours) of Critique-GRPO compared to standard GRPO, and how long does the critique generation take?

---

> ### Author Response · Authors · 2025-11-14
> **Response to Reviewer tB8n (1/3)**
>
> Thank you for your feedback. We provide the following responses to address your concerns:
>
> **Response to Weakness 1:**
>
> We appreciate the opportunity to clarify our contributions. Our work introduces a fundamentally new paradigm for online RL that integrates natural language critiques with numerical rewards for real-time learning, representing a meaningful advance beyond prior work.
>
> 1. Novel Integration of Natural Language and Numerical Feedback
>
> Our work is the first to successfully combine natural language critiques with numerical rewards in an online RL setting. This integration addresses unique challenges including performance plateaus faced by solely relying on numerical rewards (Section 3, 5). Natural language critiques provide informative, actionable feedback that enables more effective navigation of complex reasoning tasks compared to scalar rewards alone. [1]
>
> 2. Superior Performance Through Online Policy Optimization
>
> While prior work has explored natural language critiques in offline settings (Critique-FT, CITL-FT), these methods rely on static datasets of pre-collected critiques or refinements. These offline SFT methods suffer from two major flaws: they cannot adapt to a policy's evolving mistakes in real-time, and they tend to mimic training data, leading to poor generalization, e.g., reduced performance on out-of-distribution (OOD) tasks in Table 2. Our framework solves these issues by integrating language critiques with numerical rewards in an online loop. This allows the model to learn from its own on-policy generations and refinements, fostering active exploration and adaptation. Empirical results in Table 2 confirm that this online approach significantly outperforms offline baselines, particularly in OOD generalization where prior methods struggle.
>
> 3. Algorithmic Innovations Enabling Hybrid Feedback
>
> Our key algorithmic contributions include: (1) Dual Optimization: Verbally solves the credit assignment problem by jointly optimizing initial and refined responses, enabling models to internalize critiques and eventually generate correct initial responses independently. (2) Response-Refinement Ratio Balancing & Reward Shaping: Address distributional shifts unique to critique-augmented online RL, ensuring stable training across diverse environments.
>
> 4. Adaptation to Diverse, Real-Time Feedback
>
> Unlike prior methods requiring high-quality static critiques, our framework handles diverse, real-time feedback and automatically penalizes incorrect refinements. This enables continuous adaptation in open-ended environments, significantly improving robustness and generalization.
>
> This proposed approach aligns with emerging efforts to scale RL beyond numerical rewards (https://x.com/karpathy/status/1944435412489171119) and lays the foundation for further research in online RL direction.
>
> **Response to Weakness 2:**
> We appreciate this important point. To assess the generalizability of our approach beyond the Qwen model family, we conducted **additional experiments using LLaMA3.2-Instruct-3B**. As shown in our updated results, Critique-GRPO consistently outperforms the GRPO baseline, achieving an average improvement of approximately +5 points. We will include these results in Section 5.6.
>
> | Method               | MATH-500 | Minerva-MATH | Olympiad-Bench | AMC23 | AIME24 | TheoremQA | GPQA-Diamond | MMLU-Pro | Avg.  |
> |----------------------|----------|--------------|----------------|-------|--------|-----------|--------------|----------|-------|
> | Llama3.2-Instruct-3B | 46.60    | 17.30        | 15.30          | 27.50 | 13.30  | 21.80     | 22.22        | 38.16    | 22.52 |
> | +R1-GRPO             | 53.60    | 19.50        | 18.80          | 27.50 | 13.30  | 24.60     | 29.29        | 40.72    | 24.79 |
> | +Critique-GRPO       | 58.80    | 25.70        | 24.10          | 35.00 | 20.00  | 31.10     | 36.87        | 43.79    | 29.86 |
>
> [1] Provably Learning from Language Feedback

---

> ### Author Response · Authors · 2025-11-14
> **Response to Reviewer tB8n (2/3)**
>
> **Response to Weakness 3:**
>
> Thank you for raising this important point. We provide the following clarifications and additional context:
>
> 1. Impact of Policy Shaping: As discussed in Appendix I, policy shaping significantly enhances learning from natural language refinements during online RL fine-tuning.
>
> 2. GRPO Clipping for Stability: In early experiments, we removed GRPO clipping but later reintroduced it to ensure training stability. However, we did not update Section 4.2 accordingly—this oversight will be corrected in the revised version.
>
> 3. Ablation Analyses:
>    To isolate the effects of each modification in Critique-GRPO, we conducted fine-grained ablation analyses on the training objective. Since the Critique-GRPO objective is derived from GRPO, we sequentially investigated the following modifications:
>    - Removing KL loss.
>    - Introducing natural language feedback.
>    - Refinement sampling.
>    - Policy shaping.
>
> The results, summarized below, confirm that both looser optimization (removing KL loss) and natural language feedback contribute positively to policy learning. Policy shaping provides the largest performance gain by amplifying high-quality refinements. A detailed discussion will be presented in Appendix N.
>
> | Method | Modifications |||| Math (ID) ||||| Science & General (OOD) ||| Avg. |
> |--------|---------|---------|---------|---------|---------|---------|---------|---------|---------|---------|---------|---------|---------|
> | | KL Loss Removal | Language Feedback | Refinement Sampling | Policy Shaping | MATH 500 | Minerva MATH | Olympiad Bench | AMC23 | AIME24 | Theorem QA | GPQA Diamond | MMLU Pro | |
> | Qwen-base | - | - | - | - | 60.80 | 20.20 | 30.40 | 35.00 | 13.30 | 21.60 | 28.79 | 46.24 | 32.04 |
> | + R1-GRPO | × | × | × | × | 74.00 | 32.00 | 38.50 | 42.50 | 16.70 | 40.60 | 33.33 | 51.81 | 41.18 |
> | + Critique-GRPO (CoT Critique) | ✓ | × | × | × | 78.40 | 34.90 | 39.90 | 40.00 | 13.30 | 43.10 | 38.89 | 52.83 | 42.66 |
> | | ✓ | ✓ | × | × | 77.00 | 30.90 | 42.40 | 52.50 | 13.30 | 41.80 | 33.33 | 54.88 | 43.26 |
> | | ✓ | ✓ | ✓ | × | 77.40 | 41.00 | 39.70 | 45.00 | 16.70 | 42.60 | 34.34 | 54.88 | 43.95 |
> | | ✓ | ✓ | ✓ | ✓ | 77.80 | 36.80 | 42.40 | 62.50 | 20.00 | 44.00 | 37.88 | 55.28 | 47.08 |
>
> 4. Impact of critique quality: We tested the impact of noisy critiques by deliberately using a critique model to generate incorrect and misleading critiques. Our findings reveal that the model shows marginal improvement at the start of training but quickly experiences a sharp performance drop below the initial baseline due to entropy explosion. The initial marginal improvement likely stems from the model training on a greater proportion of initial responses compared to refinements. These findings demonstrate that the efficacy of Critique-GRPO cannot be achieved with noisy critiques.
>
> However, Critique-GRPO demonstrates strong robustness to various critique sources. As shown in Table 2 and Table 4, Critique-GRPO with self-critiques (68.13) performs virtually on par (<0.1% difference) with using GPT-4o-generated CoT critiques (68.26). Additionally, we evaluated Critique-GRPO with alternative critique models, including Llama3.1-405B and the fine-tuned DeepCritic-7B-RL1.5-PRM800K.
>
> | Method             | Critique Model         | MATH 500 | Minerva MATH | Olympiad Bench | AMC23  | AIME24 | Theorem QA | GPQA Diamond | MMLU Pro | Avg.   |
> |---------------------|------------------------|----------|--------------|-----------------|--------|--------|------------|--------------|----------|--------|
> | Qwen2.5-7B-Base    | -                      | 60.80    | 20.20        | 30.40           | 35.00  | 13.30  | 21.60      | 28.79        | 46.24    | 32.04  |
> | R1-GRPO            | -                      | 74.00    | 32.00        | 38.50           | 42.50  | 16.70  | 40.60      | 33.33        | 51.81    | 41.18  |
> | Critique-GRPO      | Llama3.1-405B          | 78.40    | 37.90        | 41.00           | 62.50  | 16.70  | 42.80      | 38.89        | 56.12    | 46.79  |
> | Critique-GRPO      | GPT-4o                 | 77.80    | 36.80        | 42.40           | 62.50  | 20.00  | 44.00      | 37.88        | 55.28    | 47.08  |
> | Critique-GRPO      | DeepCritic-7B          | 79.20    | 40.10        | 44.00           | 60.00  | 23.30  | 43.10      | 36.87        | 57.28    | 47.98  |
>
> These findings demonstrate that Critique-GRPO is effective and robust across a range of critique models with varying critique quality, consistently outperforming the GRPO baseline. We will include these results in Section 5.6.

---

> ### Author Response · Authors · 2025-11-16
> **Response to Reviewer tB8n (3/3)**
>
> **Response to Question 1:**
>
> Thank you for raising this question. Below, we provide clarifications and results addressing the impact of critique quality and token length on model performance.
>
> 1. Impact of Critique Quality: Please see the response to Weakness 3 (Part 2).
>
> 2. Impact of Critique Token Length: We examined how the length of critique tokens affects performance by increasing the level of detail in generated critiques. Shorter critiques (approximately 683 tokens) generally perform well. However, increasing the token length to around 2079 tokens by prompting the model to generate highly detailed critiques leads to a slight performance decline (from 47.08 to 46.00). This decline may be attributed to the difficulty of identifying critical information within longer contexts. Nonetheless, all results using Critique-GRPO consistently outperform GRPO, highlighting the robustness of Critique-GRPO. A summary of the results is provided below, more details will be presented in Appendix L.
>
> | Method                                     | Critique Length | MATH-500 | Minerva-MATH | Olympiad-Bench | AMC23 | AIME24 | TheoremQA | GPQA-Diamond | MMLU-Pro | Avg.  |
> |--------------------------------------------|-----------------|----------|--------------|----------------|-------|--------|-----------|--------------|----------|-------|
> | Qwen2.5-7B-Base                            | -               | 60.80    | 20.20        | 30.40          | 35.00 | 13.30  | 21.60     | 28.79        | 46.24    | 32.04 |
> | +R1-GRPO                                   | -               | 74.00    | 32.00        | 38.50          | 42.50 | 16.70  | 40.60     | 33.33        | 51.81    | 41.18 |
> | +Critique-GRPO (Indicative Critique)       | ~7 tokens       | 76.00    | 36.00        | 41.00          | 55.00 | 13.30  | 41.80     | 37.88        | 55.97    | 44.62 |
> | +Critique-GRPO (Indicative Critique w/ GT) | ~15 tokens      | 76.80    | 35.70        | 39.60          | 65.50 | 10.00  | 44.00     | 38.89        | 54.88    | 45.30 |
> | +Critique-GRPO (CoT critique)              | ~683 tokens     | 77.80    | 36.80        | 42.40          | 62.50 | 20.00  | 44.00     | 37.88        | 55.28    | 47.08 |
> | +Critique-GRPO (CoT critique)              | ~2079 tokens    | 77.80    | 37.90        | 38.80          | 62.50 | 16.70  | 42.80     | 35.35        | 56.12    | 46.00 |
>
> **Response to Question 2:**
>
> Thank you for raising this important point. We acknowledge that incorporating critiques and self-refinement introduces additional computational overhead. Transparent reporting of this cost is indeed essential. To clarify:
>
> | Model         | Method                         | Training Time (GPU-Hours) | Average Accuracy (%) |
> |---------------|--------------------------------|---------------------------|-----------------------|
> | Qwen2.5-7B-Base    | R1-GRPO                       | 240                       | 41.18                |
> |               | Critique-GRPO (CoT-Critique)  | 245                       | 47.08                |
> | Qwen3-8B    | R1-GRPO                       | 296                       | 63.75                |
> |               | Critique-GRPO (CoT-Critique)  | 315                       | 68.26                |
>
> By implementing *asynchronous rollouts* on Critique-GRPO, we limited the additional cost to only 5 and 19 GPU-hours, representing a ~2% and ~6.5% increase over the GRPO baselines on A800 GPUs. This *acceptable overhead* is strongly justified by the +6 and +4.5 average point gains across 8 tasks, achieved using only 4K prompts enriched with natural language and numerical feedback. These results demonstrate significant improvements in both data efficiency and model performance, making the added cost highly worthwhile. Additionally, our approach does not rely exclusively on advanced models like GPT-4. As demonstrated in our response to Weakness 5, Critique-GRPO is compatible with self-critiques and alternative critique models, such as Llama3.1-405B and DeepCritic-7B, offering cost-effective and scalable alternatives. We will include these efficiency comparisons in Appendix M.
>
> Additionally, our preliminary results suggest that randomly sampling one failed initial response for refinement generation during training time yields comparable performance. Future researchers may consider this as a cost-effective alternative for real-world applications.
>
> We appreciate your valuable feedback and will incorporate these clarifications and additional results in our revised manuscript to ensure greater transparency and rigor. We hope these responses address your concerns.

---

> ### Author Response · Authors · 2025-11-26
> **A Gentle Reminder**
>
> Dear Reviewer,
>
> Thank you once again for your valuable comments! As the discussion phase is approaching its end, we would like to kindly confirm whether we have sufficiently addressed all of your concerns (or at least some of them). Should there be any remaining questions requiring further clarification, please do not hesitate to let us know. If you are satisfied with our responses, we would greatly appreciate your consideration in adjusting the evaluation scores accordingly. We sincerely look forward to your feedback.

---

### Official Review · Reviewer_qUVB · 2025-11-01

**Soundness:** 3
**Presentation:** 3
**Contribution:** 3
**Rating:** 4
**Confidence:** 3

**Summary:**

The paper introduces Critique-GRPO, an online RL framework for fine-tuning large language models with both numerical and natural language feedback.
Instead of optimizing only scalar rewards as in GRPO or R1-style methods, the model receives textual critiques that guide a refinement step before policy updating.
The authors test this approach on multiple reasoning benchmarks using Qwen-7B and 8B models, showing consistent improvements over R1-GRPO and other offline critique-based fine-tuning methods.
Ablation studies analyze critique types, shaping functions, and the role of online vs. offline training.

**Strengths:**

The method is well-engineered: integrating natural language feedback into an online RL loop is implemented cleanly and supported by solid ablations.

The study includes careful analyses of critique types, shaping functions, and self-critique variants, which justify each design choice.

Results are consistent, statistically stable, and show meaningful improvements across tasks.

**Weaknesses:**

1) The conceptual novelty is moderate. The idea of critique-guided refinement has been explored in prior offline methods (Critique-FT, CITL-FT). The contribution mainly lies in bringing it into an online GRPO-style RL framework and tuning several design components.

2) Most innovations—such as removing clipping/KL, adding a shaping function, or optimizing both initial and refined responses—are engineering-level adjustments rather than new theoretical ideas.

3) The study is limited to Qwen-7B/8B models and reasoning benchmarks. There is no evidence of cross-model generalization or scaling analysis, so it remains unclear whether the method truly closes the gap with larger models (e.g., Qwen-32B, GPT-4) or only provides small gains under constrained capacity.

4) The paper assumes that textual critiques are inherently superior to scalar rewards but does not compare against graded or structured reward signals (e.g., partial credit, continuous correctness scores). Without testing these intermediate forms, it is uncertain whether the gains come from the linguistic form of feedback or simply from providing richer supervision.

5) The reliance on GPT-4 to generate critiques limits practicality and makes the training process resource-intensive. Self-critique is promising but still weaker in performance.

6) (Minor) The analysis section remains surface-level; there is no deeper explanation of why critique feedback improves optimization dynamics.

**Questions:**

1) Could the authors compare their method to a graded reward baseline, where critique text is converted into a numerical score (e.g., 0–5 or partial correctness)? This would clarify whether textual feedback itself brings additional benefits.

2) Have the authors tried larger or different base models (Llama, Mistral, or Qwen-32B) to test the method’s scalability and generality?

3) How sensitive is the approach to the quality of critiques? For example, if the critiques are noisy or inconsistent, does the policy still improve?

4) Could the authors share more analysis on the optimization behavior, e.g. does the critique-guided refinement change token-level gradient patterns compared to GRPO?

5) Is there a plan to make the critique generation process cheaper or partially automated without GPT-4 dependence?

---

> ### Author Response · Authors · 2025-11-21
> **Response to Reviewer qUVB (1/3)**
>
> Thank you for your feedback. We provide the following responses to address your concerns:
>
> **Response to Weakness 1 & 2:**
>
> We appreciate the opportunity to clarify our contributions. Our work introduces a fundamentally new paradigm for online RL that integrates natural language critiques with numerical rewards for real-time learning, representing a meaningful advance beyond prior work.
>
> 1. Novel Integration of Natural Language and Numerical Feedback
>
> Our work is the first to successfully combine natural language critiques with numerical rewards in an online RL setting. This integration addresses unique challenges including performance plateaus faced by solely relying on numerical rewards (Section 3, 5). Natural language critiques provide informative, actionable feedback that enables more effective navigation of complex reasoning tasks compared to scalar rewards alone. [1]
>
> 2. Superior Performance Through Online Policy Optimization
>
> Prior works (Critique-FT, CITL-FT) rely on static datasets of pre-collected critiques or refinements. These offline SFT methods suffer from two major flaws: they cannot adapt to a policy's evolving mistakes in real-time, and they tend to mimic training data, leading to poor generalization, e.g., reduced performance on out-of-distribution (OOD) tasks in Table 2. Our framework solves these issues by integrating language critiques with numerical rewards in an online loop. This allows the model to learn from its own on-policy generations and refinements, fostering active exploration and adaptation. Empirical results (Table 2) confirm that this online approach significantly outperforms offline baselines, particularly in OOD generalization where prior methods struggle.
>
> 3. Algorithmic Innovations Enabling Hybrid Feedback
>
> Our key algorithmic contributions include: (1) Dual Optimization: Verbally solves the credit assignment problem by jointly optimizing initial and refined responses, enabling models to internalize critiques and eventually generate correct initial responses independently. (2) Response-Refinement Ratio Balancing & Reward Shaping: Address distributional shifts unique to critique-augmented online RL, ensuring stable training across diverse environments.
>
> 4. Adaptation to Diverse, Real-Time Feedback
>
> Unlike prior methods requiring high-quality static critiques, our framework handles diverse, real-time feedback and automatically penalizes incorrect refinements. This enables continuous adaptation in open-ended environments, significantly improving robustness and generalization.
>
> This proposed approach aligns with emerging efforts to scale RL beyond numerical rewards (https://x.com/karpathy/status/1944435412489171119) and lays the foundation for further research in online RL direction.
>
> **Response to Weakness 3:**
>
> Thank you for highlighting this important point. We have addressed this concern as follows:
>
> 1. Cross-Model Generalization and Scaling Analysis
>
> To extend our analysis, we evaluated Critique-GRPO across different model architectures and scales, including Llama3.2-3B-Instruct and Qwen3-32B. The decoding budget is set to 8192 tokens for Llama3.2-3B-Instruct and 16384 tokens for Qwen3-32B. The results, presented below, demonstrate that Critique-GRPO consistently outperforms GRPO by significant margins (+5.1 points on Llama3.2-3B-Instruct and +4.0 points on Qwen3-32B). Additional details will be provided in Section 5.6.
>
>    | Method                | MATH 500 | Minerva MATH | Olympiad Bench | AMC23  | AIME24 | Theorem QA | GPQA Diamond | MMLU Pro | Avg.   |
>    |-----------------------|----------|--------------|----------------|--------|--------|------------|--------------|----------|--------|
>    | Evaluation with varying policy model architectures |
>    | Llama3.2-3B-Instruct | 46.60    | 17.30        | 15.30          | 27.50  | 13.30  | 21.80      | 22.22        | 38.16    | 22.52  |
>    | R1-GRPO              | 53.60    | 19.50        | 18.80          | 27.50  | 13.30  | 24.60      | 29.29        | 40.72    | 24.79  |
>    | Critique-GRPO        | 58.80    | 25.70        | 24.10          | 35.00  | 20.00  | 31.10      | 36.87        | 43.79    | 29.86  |
>    | Evaluation with varying policy model scales |
>    | Qwen3-32B            | 91.80    | 53.70        | 62.70          | 92.50  | 63.30  | 62.40      | 45.96        | 77.06    | 68.68  |
>    | R1-GRPO              | 93.00    | 54.00        | 66.40          | 97.50  | 80.00  | 64.00      | 56.57        | 77.97    | 73.68  |
>    | Critique-GRPO        | 96.40    | 59.60        | 69.30          | 100.00 | 90.00  | 66.40      | 60.61        | 78.49    | 77.60  |
>
> 2. Closing the Gap with Larger Models
>
> We observe that, under the same decoding budget (8192 tokens), the average performance of Critique-GRPO on Qwen3-8B (68.26) surpasses Qwen3-32B (60.15) by +8.1 points. This indicates the Critique-GRPO not only enables concise reasoning but also helps close the gap with larger models.
>
> [1] Provably Learning from Language Feedback

---

> ### Author Response · Authors · 2025-11-21
> **Response to Reviewer qUVB (2/3)**
>
> **Response to Weakness 4:**
>
> Thank you for raising this important point. We would like to clarify that our goal is to validate the efficacy of combining natural language feedback with numerical feedback, compared to numerical feedback alone. We do not assume that textual critiques are inherently superior to scalar rewards. However, the superiority of learning from textual feedback over scalar rewards has been established in recent theoretical work [1].
>
> Furthermore, the superior performance of Critique-GRPO compared to prior methods that utilize both outcome- and process-oriented rewards (e.g., PRIME, as shown in Table 3) highlights the effectiveness of our approach. Textual critiques, provided without expert demonstrations, enable the model to diagnose errors and generate corrected reasoning chains. This process, detailed in Section 3.2 and Appendix C, supports verbal step-level credit assignment, allowing the model to identify and rectify flawed reasoning steps.
>
> As demonstrated in Tables 1, 5, and 6, Critique-GRPO helps the model overcome persistent failures and performance plateaus that are commonly seen when relying solely on numerical rewards. Appendix R provides additional discussion of these results. This evidence suggests that the improvements are stem from the unique advantages of linguistic feedback in guiding reasoning and corrective behaviors.
>
> **Response to Weakness 5:**
>
> Thank you for raising this concern. We would like to clarify that the effectiveness of Critique-GRPO **does not depend on advanced models** and is compatible with various feedback sources, as validated in the paper.
>
> Specifically, Table 2 shows that Critique-GRPO with *indicative feedback (``The proposed solution is correct/incorrect.'')* derived from a rule-based reward system yields an average improvement of *+3.5 to 4 points* over the GRPO baseline. This highlights a fair and meaningful comparison that does not rely on any external models.
>
> Crucially, as shown in Table 2 and Table 4, Critique-GRPO with self-critiques (68.13) is virtually on par (<0.1% difference) with using GPT-4o-generated CoT critiques (68.26).  Additionally, we evaluate Critique-GRPO with alternative critique models, such as Llama3.1-405B and the fine-tuned DeepCritic-7B-RL1.5-PRM800K. The results below highlight the adaptability of Critique-GRPO to both proprietary and fine-tuned critique models:
>
> | Method             | Critique Model         | MATH 500 | Minerva MATH | Olympiad Bench | AMC23  | AIME24 | Theorem QA | GPQA Diamond | MMLU Pro | Avg.   |
> |---------------------|------------------------|----------|--------------|-----------------|--------|--------|------------|--------------|----------|--------|
> | Qwen2.5-7B-Base    | -                      | 60.80    | 20.20        | 30.40           | 35.00  | 13.30  | 21.60      | 28.79        | 46.24    | 32.04  |
> | R1-GRPO            | -                      | 74.00    | 32.00        | 38.50           | 42.50  | 16.70  | 40.60      | 33.33        | 51.81    | 41.18  |
> | Critique-GRPO      | Llama3.1-405B          | 78.40    | 37.90        | 41.00           | 62.50  | 16.70  | 42.80      | 38.89        | 56.12    | 46.79  |
> | Critique-GRPO      | GPT-4o                 | 77.80    | 36.80        | 42.40           | 62.50  | 20.00  | 44.00      | 37.88        | 55.28    | 47.08  |
> | Critique-GRPO      | DeepCritic-7B          | 79.20    | 40.10        | 44.00           | 60.00  | 23.30  | 43.10      | 36.87        | 57.28    | 47.98  |
>
> These findings demonstrate that Critique-GRPO is effective across a range of critique models and does not rely exclusively on powerful external models like GPT-4o. We will include these results in Section 5.6.
>
> **Response to Weakness 6:**
>
> Thank you for raising this point. This work leverages textual critiques to generate valid refinements, enabling the model to learn from both its initial responses and the refinements during policy optimization. Critique-guided refinement acts as an verbal credit assignment process, which differs from fine-grained scalar reward assignment.
>
> As discussed in Section 5.5, textual critiques encourage exploration and facilitate the generation of valid refinements. This prevents fast entropy collapse, thereby improving optimization dynamics. This mechanism provides a deeper explanation of how critique feedback impacts the learning process and is consistent with recent findings in related studies [2]. Additionally, the superiority of learning from textual feedback over scalar rewards has been well established in prior theoretical work [1]. By combining the benefits of both textual feedback and scalar rewards, our approach achieves more effective and robust policy optimization.
>
> [1] Provably Learning from Language Feedback
>
> [2] The Entropy Mechanism of Reinforcement Learning for Reasoning Language Models

---

> ### Author Response · Authors · 2025-11-21
> **Response to Reviewer qUVB (3/3)**
>
> **Response to Question 1:**
>
> Thank you for raising this question. Please refer to our responses to Weakness 4 and Weakness 6 for a detailed discussion of the benefits of textual feedback. Additionally, our proposed approach is inherently compatible with fine-grained rewards. In addition, we want to emphasize that the primary focus of our work is to validate the effectiveness of combining natural language feedback with numerical feedback, compared to numerical feedback alone.
>
> **Response to Question 2:**
>
> Thank you for your question. Please refer to our response to Weakness 3, where we present evaluations on different base models and scales, including Llama3.2-3B-Instruct and Qwen3-32B, to demonstrate the scalability and generality of our method.
>
> **Response to Question 3:**
>
> Thank you for this insightful question. To assess the sensitivity of our approach, we tested the impact of noisy critiques by deliberately using a critique model to generate incorrect and misleading critiques. Our findings showed that while there was marginal improvement at the start of training, the model quickly experienced a sharp performance drop below the initial baseline due to entropy explosion. The initial marginal improvement likely arises because the model trains on a greater proportion of initial responses compared to refinements. These results confirm that Critique-GRPO cannot achieve its effectiveness with noisy critiques. In addition, the findings in our response to Weakness 5 demonstrate that Critique-GRPO is robust across a range of critique models with varying critique quality, consistently outperforming the GRPO baseline.
>
> **Response to Question 4:**
>
> Thank you for raising this question. Please refer to our response to Weakness 6.
>
> **Response to Question 5:**
>
> Thank you for this important question. We acknowledge that incorporating critiques and self-refinement introduces additional computational overhead. Transparent reporting of this cost is essential. To clarify:
>
> | Model           | Method                         | Training Time (GPU-Hours) | Average Accuracy (%) |
> |------------------|--------------------------------|---------------------------|-----------------------|
> | Qwen2.5-7B-Base | R1-GRPO                        | 240                       | 41.18                |
> |                  | Critique-GRPO (CoT-Critique)  | 245                       | 47.08                |
> | Qwen3-8B        | R1-GRPO                        | 296                       | 63.75                |
> |                  | Critique-GRPO (CoT-Critique)  | 315                       | 68.26                |
>
> By implementing *asynchronous rollouts* on Critique-GRPO, we limited the additional cost to only 5 and 19 GPU-hours, representing a ~2% and ~6.5% increase over the GRPO baselines on A800 GPUs. This *acceptable overhead* is strongly justified by the +6 and +4.5 average point gains across 8 tasks, achieved using only 4K prompts enriched with natural language and numerical feedback. These results demonstrate significant improvements in both data efficiency and model performance, making the added cost highly worthwhile. Additionally, our approach does not rely exclusively on advanced models like GPT-4. As demonstrated in our response to Weakness 5, Critique-GRPO is compatible with self-critiques and alternative critique models, such as Llama3.1-405B and DeepCritic-7B, offering cost-effective and scalable alternatives. We will include these efficiency comparisons in Appendix M.
>
> Additionally, our preliminary results suggest that randomly sampling one failed initial response for refinement generation during training time yields comparable performance. Future researchers may consider this as a cost-effective alternative for real-world applications.
>
> We appreciate your valuable feedback and will incorporate these clarifications and additional results in our revised manuscript to ensure greater transparency and rigor. We hope these responses address your concerns.

---

> ### Author Response · Authors · 2025-11-26
> **A Gentle Reminder**
>
> Dear Reviewer,
>
> Thank you once again for your valuable comments! As the discussion phase is approaching its end, we would like to kindly confirm whether we have sufficiently addressed all of your concerns (or at least some of them). Should there be any remaining questions requiring further clarification, please do not hesitate to let us know. If you are satisfied with our responses, we would greatly appreciate your consideration in adjusting the evaluation scores accordingly. We sincerely look forward to your feedback.

---

### Author Response · Authors · 2025-11-28
**General Response**

We sincerely thank all reviewers for their thoughtful and constructive feedback. We greatly appreciate the recognition of:

1. The introduction of an online RL framework that integrates natural‑language and numerical feedback. (qUVB, tB8n, cPJg, StXP)
2. Strong and consistent empirical gains across eight reasoning tasks. (qUVB, tB8n, cPJg, StXP)
3. Comprehensive analysis of training dynamics, entropy, and design choices. (qUVB, tB8n, cPJg)
4. Straightforward and reproducible methodology. (cPJg)

We have carefully addressed all raised concerns, including:

1. Generalization with Llama-3.2-3B-Instruct: We added new experiments to demonstrate generalization. (qUVB, tB8n, cPJg, StXP)
2. Scalability on larger models: We included evaluations with Qwen3-32B. (qUVB)
3. Computational cost analysis: A detailed breakdown shows that our method introduces an acceptable overhead (5–19 GPU hours) compared to standard GRPO. Additionally, our preliminary results suggest that randomly sampling one failed initial response for refinement generation during training time yields comparable performance. Future researchers may consider this as a cost-effective alternative for real-world applications. (qUVB, tB8n, cPJg, StXP)
4. Reduced reliance on proprietary models: We validated that Critique-GRPO performs effectively with Self-Critique, Llama-3.1-405B, and a small critique model, demonstrating it is not dependent on GPT-4. (qUVB, tB8n, StXP)
5. Fine-grained ablations: We isolated the contributions of KL removal, refinement sampling, and policy shaping to clarify the sources of performance gains. (qUVB, tB8n, cPJg)
6. Robustness to critique quality: We analyzed the impact of noisy and adversarial critiques, confirming the method’s reliance on valid feedback signals. (qUVB, tB8n)
7. Optimization dynamics: We clarified entropy behavior and explained why critiques improve exploration. (qUVB, tB8n)
8. Fairness in supervision: We provided results using self-critiques, rule-based indicative critiques. (qUVB, StXP)
9. Clarification of novelty: We introduced a new paradigm for online RL that integrates critiques with numerical rewards for real-time learning. By providing informative feedback, natural language critiques enables the model to navigate complex reasoning paths more effectively than scalar rewards alone [1]. This approach aligns with emerging efforts to scale RL beyond numerical rewards (https://x.com/karpathy/status/1944435412489171119) and lays the foundation for further research in this direction. (qUVB, tB8n, StXP)

All major modifications are highlighted in the revised paper. Once again, we sincerely thank the reviewers for their time and invaluable insights, which have helped us strengthen the paper.

[1] Provably Learning from Language Feedback

---

### Meta-Review · Area_Chair_aRnj · 2025-12-21

**Summary:**

Advantages:
1. It integrated natural‑language and numerical feedback to the online RL framework, clearly implemented.
2. It justified the design choice by analyzing the critique types, shaping functions, and self-critique variants.
3. Experimental results and appended studies showed the performance improvement and generalization over baselines across tasks.

Weaknesses:
1. The contribution was minimal, by combining existing techniques to online RL. "The first" is over-claimed, for the authors should clarify what was the major problem for the online problem, and how they conquer it the first?
2. Missing generalization and scalability experiments with other models than Qwen in the original version, pointed by all the reviewers, although added some in the rebuttal.
3. Raising computational cost compared to standard GRPO. Although the authors carried out simple comparison experiments, they might differ a lot for different benchmarks.

The paper was unprepared in their original version, so most reviewers raised common issues with the same low score. Although adding experiments in the rebuttal, the paper still needs more rigorous improvement by considering all the review comments carefully.

**Reviewer Concerns:**

Thanks for the rebuttal and the added experimental results.

The reviewer cPJg with score 6 replied to keep it. However, others did not reply with unchanged score 4.

**Reviewer Scores:**

It received 4 reviews with the average score 4.5.

The authors made rebuttal, but reviewers may not be persuaded to change the score.

---

### Decision · Program_Chairs · 2026-01-26

Reject